

# A new method to separate precipitation phases
Yulian Liu[a,b]    Guoyu Ren[a,c]    Xiubao Sun[a,c]    Xiufen Li[d,e]
[a] Department of Atmospheric Science, School of Environmental Science, China University of
Geosciences, Wuhan 430074, China;
[b] Heilongjiang Climate Center, Harbin 150030,China;
[c] Laboratory for Climate Studies, National Climate Center, CMA, Beijing100081,China;
[d] Innovation and Opening laboratory of Regional Eco-Meteorology in Northeast, China Meteorological
Administration
[e] Heilongjiang Provincial Institute of Meteorological Sciences, Harbin 150030,China.
Corresponding author: G. Ren (guoyoo@cma.gov.cn)









**Abstract**
Separating the solid precipitation from liquid precipitation in an existing
historical precipitation observation data series is a key problem in the monitoring and
study of climate anomaly and long-term change of extreme precipitation events in
difference phases. In this study, based on the comprehensive analysis of the historical
daily temperature, precipitation data, and weather phenomenon records in the northern
areas of Mainland China (north of 30 N), the threshold temperature of rainfall and
snowfall in historical precipitation data for a complex and diverse geographical and
climatic region were determined. A statistical model was established, and a method of
separating solid precipitation from liquid precipitation was proposed. The main
conclusions include: (1) in northern China, the actual threshold temperature range of
the daily mean temperature of rain and snow determined based on weather
phenomenon records was between -1.2–6.3 ℃, with a difference of 7.5 ℃ among areas,
and a mean threshold value of 2.81 ℃ for the whole region. The actual threshold
temperature in the northern Tibetan Plateau was the highest (generally higher than
4 ℃). The low threshold temperature values appeared in eastern Northeast China,
North China, and northern Xinjiang Autonomous Region, which were less than 2 ℃.
(2) The actual threshold temperature decreased with increase in longitude east of
105 E; meanwhile, it was more dispersed in the areas west of 105 E. The actual
threshold temperature was generally higher and more variable in the low latitude areas,
while it was lower and more concentrated in the high latitude; the threshold
temperature was lower in the low-altitude areas and higher in the high-altitude areas,



and it generally increased with altitude. (3) There was a negative correlation between
the actual threshold temperature and the annual precipitation; the actual threshold
temperature was higher in the areas with less precipitation, and lower in the areas with
more precipitation. The actual threshold temperature was negatively correlated with
the annual average relative humidity, and was generally low in humid areas with
relatively large humidity and vice versa. (4) The multivariate regression fitting model
developed in this paper based on latitude, altitude, and annual precipitation was able
to simulate the actual threshold temperature of the precipitation phase in northern
China well. According to the calculated threshold temperature based on the model, the
relative deviation of snow days and snowfall are smaller, and the stations with less
than 10% of relative deviation reached 95.1% and 90.7%, respectively. The results of
this study can be used for the separation of solid and liquid precipitation events in the
areas without sufficient weather phenomenon records or metadata.
**Key words:** Northern China; Precipitation; Phase; Snowfall; Rainfall; Separation;
Statistical model; Simulation; Regional differences









## 1. Introduction

Precipitation is an important parameter used to characterize climate characteristics and climate change, and it is one of the key components of the Earth's water and energy cycles (Loth et al., 1993). The influence of different phases of precipitation on the surface water and energy cycles is enormous (Vavrus, 2007; Wu et al., 2009), as more than 50% of the global meteorological disasters are closely related to different phases of precipitation (WMO, 2013; Wang et al., 2005). Under the same precipitation condition, the effect of different phases of precipitation on the Earth's surface system and the social and economic system is clearly different, thus it is important to distinguish and understand the characteristics and anomalies of snowfall or sleet and their causes. In addition, when monitoring and studying the long-term changes in extreme precipitation events on sub-continental to global scales, it is also necessary to distinguish rainfall and snowfall events from historical precipitation data.

To date, many studies have been published on the characteristics and multi-decadal variation of snowfall in China (e.g. Jiang et al., 2003; Yang et al., 2005; Qin et al., 2006; Liu et al., 2012, 2013; Zhang et al., 2015). Also, many studies on both the global and Asian regional total precipitation and extreme precipitation events and their long-term change have been reported (Becker et al., 2012; Noake et al., 2012; Polson et al., 2013; Blanchet et al., 2009; O'Hara et al., 2009; Kunkel et al., 2009; Ren, 2007, 2015a, 2015b, 2016; Liu et al., 2011; Fang et al., 2011; Yu et al., 2014; Zhong et al., 2013; Wan et al., 2013; Xiao et al., 2015; Dang et al., 2015). All of these



studies have greatly enriched the understanding of global precipitation and snowfall
climatology and the climate change and variability in different regions and varied
scales. However, less research has been done on global and Asian regional solid
precipitation; this is mainly because there is solid precipitation observation in the
domestic surface observation network, while the current global datasets only contain
the total precipitation amount without type of precipitation phase, and researchers
usually cannot separate liquid and solid precipitation (snowfall). Even in the case of
relatively abundant meteorological observational data in China, some works often
need to use certain methods to separate the different phases of precipitation in
historical precipitation data.
Many scholars have discussed the phase identification of precipitations (Harder,
2013, 2014). Dai (2008) analyses the temperature range of precipitation phase change
on the continent and the ocean, and discusses the relationship between the phase
change temperature and the pressure. Stefan et al. (2008) proposes to use two input
variables (threshold temperature and range) to estimate daily snowfall from
precipitation data. Ye et al. (2013) suggests that application of site-specific critical
values of air temperature and dewpoint to discriminate between solid and liquid
precipitation is needed to improve snow and hydrological modeling at local and
regional scales. Froidurot et al. (2014) points out that surface air temperature and
relative humidity show the greatest explanatory power. Sims and Liu (2015) point out
that atmospheric moisture impacts precipitation phase and that wet-bulb temperature,
rather than ambient air temperature, should be used to separate solid and liquid



precipitation. Harpold et al. (2017) and Keith et al. (2018) all point out that a humidity
phase prediction method had similar accuracy to temperature phase prediction method
in separating snowfall from precipitation data.

After the large-scale freezing rain and snow disaster in Central and South China

in winter of 2008, domestic scholars paid more attention to the studies of the
discrimination and identification of the precipitation phase, in order to meet the
challenge of the disastrous weather forecast (Liu et al., 2013). The discriminant basis
is generally the temperature of the surface and upper air layers. Zhang et al. (2013)
studied the identification criteria of winter precipitation phase in Beijing, and pointed
out that the phase transition in Beijing mainly occurred in March and November. They
found six physical quantities closely related to the conversion of snow and rain (850
hPa temperature, 925 hPa temperature, 1000 hPa temperature, thickness between
1000 hPa and 700 hPa, thickness between 1000 hPa and 850 hPa, and the combination
of surface air temperature and relative humidity). According to these physical
quantities, the objective forecast index of the Beijing winter precipitation phrases was
established, and its accuracy reached 77%. You et al. (2013) also analyzed the
discriminant index of precipitation phases in Beijing, pointing out that precipitation is
considered as rainfall when the surface air temperature is greater than 2 ℃ and the
dew temperature is greater than or equal to 0 ℃, and precipitation is considered as
snowfall when the surface air temperature is less than 1 ℃ and the dew temperature is
less than 0 ℃. It is sleet, or rain and snow, when the surface air temperature is
between 1 ℃ and 3 ℃. The surface air temperature, dew temperature, upper air



temperature, and relative humidity are frequently used in developing methods to
discriminate precipitation phases.
However, in a larger scale study, it is usually difficult to obtain the observational
records in the global dataset. Bourgouin (2000) introduced the area-method in
separating different precipitation phases, which is based on the vertical thermal
structure of the atmosphere, the distribution of condensation nuclei of water vapor,
and the descent velocity to predict the precipitation phase (liquid or solid). The
method, however, also needs data of multiple observational variables in surface and
upper atmosphere, which is difficult to obtain.
Rainfall-induced runoff and snowmelt runoff are completely different
hydrological processes. Therefore, in some hydrological models, the solid-liquid
precipitation separation uses the double threshold temperature method (Wigmosta et
al., 1994; Kang et al., 1999, 2001; Chen et al., 2008) and the single threshold
temperature method (Arnold et al., 1998; Refsgaard et al., 1998; Wang et al., 2004),
or relies on precipitation radar monitoring data (Terry et al., 2012; Edwin et al., 2006).
Han et al. (2010) discussed the difficulty of applying the double threshold temperature
method. They used the data of the national stations of the China Meteorological
Administration (CMA) during 1961–1979 to draw a single threshold temperature
contour map, and combined it with the monthly snowfall ratio method to separate the
precipitation phases by determining occurrence of snowfall and the amount of
snowfall in the watershed. Chen et al. (2013) improved the solid-liquid precipitation
separation procedure for mainland China by supplementing the threshold of daily



mean dew temperature. The data used for the previous studies were observed prior to
1979, and they used the monthly snowfall ratio method as an auxiliary indicator.
When the rainfall and snowfall condition in different regions outside mainland China
is not known, and at the same time there is no dew temperature data in the current
international datasets, the method cannot be applied to the larger scale analysis.

Although humidity phase separating method has a similar suitability with

temperature based method (Arpold et al., 2017; Keith et al., 2018), it is at the same
time difficult to be used in large scale due to the unavailability of humidity data.
Research on the global scales can be only based on the temperature phase separating
method.

China has sub-continental scale characteristics of lands and natural conditions,

and has a diversity of climates and topographic types, and the phase separating
methods developed in mainland China should have a better universality in continents
and the world.

In this work, the precipitation phase separation method was developed by using

the daily observational data of the national stations for years 1961–2013 in mainland
China, and the threshold temperature values of rainfall and snowfall in northern China
(north of 30°N) was analyzed and tested. A statistical model of the threshold
temperature was established to provide a method for use in studies of large-scale
snowfall climatology and climate change, weather forecasting, and hydrological
model parameterization.



## 2. Data and methods


The main purpose of this study was to develop a method for separating solid and
liquid precipitation, so that the objective separation of solid and liquid parts of
precipitation can be achieved without exhaustive reference of observational data.
International exchange data generally only contain the daily temperature and
precipitation, with no other reference data, so we have only used the indicators related
to temperature and precipitation to develop a method of separation.
The data used was obtained from the National Meteorological Information
Center of China Meteorological Administration (CMA). The air temperature,
precipitation and relative humidity data were derived from the "China Land Daily
Climatic Dataset (V3.0)''. The precipitation weather phenomenon was derived from
"China Land Climatic Data Daily Weather Phenomena Dataset''. All the data have
been quality controlled. Collected since January 1951, the "China Land Daily
Climatic Dataset (V3.0)'' contains the daily data of 839 national stations' air pressure,
surface air temperature (daily mean, daily maximum and daily minimum),
precipitation, evaporation, relative humidity, wind speed, sunshine hours, and 0-cm
ground temperature. The "China Land Climatic Data Daily Weather Phenomena
Dataset'' is the daily records encoded by the 752 national stations in mainland China
since 1951. Cross comparison of the two datasets and the examination of station
information was performed, and any incomplete temperature, precipitation, relative
humidity and weather phenomena data were removed. At the same time, the data of
the latitude and longitude of the station were corrected. There are 623 stations



selected for use in the study, all of which meet the demand to have information
integrity, sequential continuity, and records of more than 20 years in climate reference
period (1981–2010). The data may contain inhomogeneities caused by the relocation
and other factors, but they would exert little influence on the analysis results, so the
data are not adjusted for homogeneity.

First, the precipitation caused by fog, dew, and frost as well as the trace

precipitation was removed, and daily precipitation greater than or equal to 1 mm was
taken as the effective precipitation. In this regard, the main consideration is that the
international exchange precipitation observation data only contains greater than or
equal to 1 mm of daily precipitation. The rain and snow separation procedures
developed in China thus can be compared with the corresponding works of other
regions, and the method developed in this paper will be able to be applied to larger
scale research.

In the separation of daily rainfall (pure rain), sleet, snow (pure snow) events,

'pure rain' was registered when the weather phenomenon data indicate that only rain
occurred on that day without snow and sleet; it was registered as 'pure snow' when
only snowfall occurred without rain and sleet, and 'sleet' when there is rain and snow
in the same day, in the records of weather phenomenon data. The daily maximum and
minimum temperature during an occurrence of sleet at each station were recorded as
the reference thresholds for the snow and rain temperature threshold values.

When there is less snowfall at the station in lower latitude zone or more arid

regions, there may be random cases of snowfall. An example is from Lijiang station,





Yunnan, located in 26 N, at which pure snow occurred only six times in the 30 years
from 1981 to 2010. The representation of the threshold temperature would be poor in
these cases. In order to ensure that the snowfall frequency is great enough and the
threshold temperature is representative, we took 324 stations (Fig. 1) in northern
China for use in this study. They are generally located north of the Yangtze River,
approximately consistent with the January mean temperature isotherm of in 3 ℃ or the
30 N parallel. The days with the snowfall records during 1981-2010 were greater than
or equal to 100d. In order to avoid the influence of extreme values on the
determination of threshold temperature, the maximum and minimum daily mean
temperature in each of the precipitation phases were not counted.
For the extreme rain and snow records, comparison was made to ensure that the
minimum and maximum temperature was correct by examining the weather
phenomena, surface air temperature and precipitation on the same day. When sleet
occurred, the range of daily mean temperature was larger. Threshold temperature was
determined only for pure rain and pure snow; The daily mean temperature on a sleet
day was only taken as the reference temperature threshold value.
According to the method of China's physical geographical regionalization,
mainland China is divided into three natural geographical divisions: Eastern Monsoon
Region (I, 231 stations), Northwest Arid Region (II, 67 stations), and Qinghai-Tibetan
Plateau Region (III, 26 stations) (Fig. 1). The representative station of the Eastern
Monsoon Region is Zhaozhou station in Heilongjiang province, which has the lowest
threshold temperature of snowfall and rainfall in the country. The representative



station of the Qinghai-Tibet Plateau Region is Shiquanhe station in Tibet Autonomous
Region, which has the highest threshold temperature of snowfall and rainfall in the
country. There are relatively fewer precipitation events in the Northwest Arid Region,
and Balikun station in Xinjiang Autonomous Region was selected as the
representative station because it observed relatively more precipitation events, and the
rain, sleet, and snow events were evenly distributed. The station is also far from the
two other regions (Table 1).

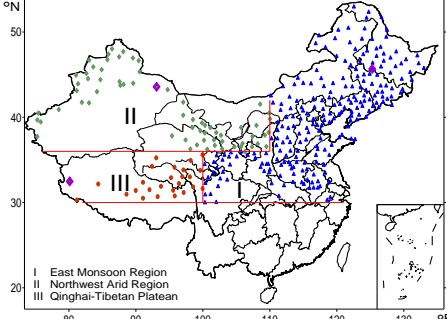

**FIG.1. Regionalization and distribution of 324 national stations north of 30 °N in mainland China**
**(I: East Monsoon Region; II: Northwest Arid Region; III: Qinghai-Tibetan Plateau;**
**Blue triangle: stations in the East Monsoon Region; Green diamond: stations in the Northwest Arid Region; Red**
**circle: stations in the Qinghai-Tibetan Plateau.**
**The purple diamond denotes the representative stations in different regions: Zhaozhou of Region I; Balikun of**
**Region II; Shiquanhe of Region III)**

**Table 1 Information of representative stations**

| Station name | Zhaozhou | Balikun | Shiquanhe |
|---|---|---|---|
| Province | Heilongjiang | Xinjiang | Tibet |
| Climate zone | I | II | III |
| Elevation(m) | 148.7 | 1679.4 | 4278.6 |
| Latitude(N) | 45°42′ | 43°36′ | 32°30′ |
| Longitude(E) | 125°15′ | 93°03′ | 80°05′ |


The relative or percent deviation of snow days (snowfall) was defined as the

percentage (%) of the difference between simulated snow days (snowfall) and actual



snow days (snowfall) to actual snow days (snowfall), which could be used to indicate
the effectiveness of simulated results.

The establishment of model was realized using the stepwise regression analysis

method included with the SPSS Statistics 17.0. The basic idea of stepwise regression
is that the variables are introduced one by one, the condition of introducing the
variable is the square of the partial regression, and the test is significant; at the same
time, after the introduction of each variable, the selected variables are checked
individually and the insignificant variables are eliminated to ensure that all the
variables in the final variable subset are significant. Thus, after a number of steps, we
obtain the "Optimal" variable subset. The advantage of stepwise regression is that the
number of the arguments contained in the regression equation is fewer, it is easy to
apply, the root mean squared error (RMSE) is small, and the model created is more
stable. All the arguments in the equation are guaranteed to be significant because each
step has been tested.

Figure 2 shows a flow diagram of the analysis of this paper. Firstly, the daily

mean temperature of different precipitation phases in northern China was calculated,
the threshold temperature of each station was determined by the method of 'snow-day
mean temperature', and the relationships between threshold temperature and
geographical and climatic factors were analyzed. Then, by using the stepwise
regression analysis method in a module of the SPSS software, the main factors
affecting the threshold temperature were determined, and the threshold temperature
model was established. Finally, the difference of the simulated threshold temperature




and the actual threshold temperature was analyzed. The spatial distribution of the
relative deviation was examined, and the applicability of the model was tested and
evaluated, in the last step.

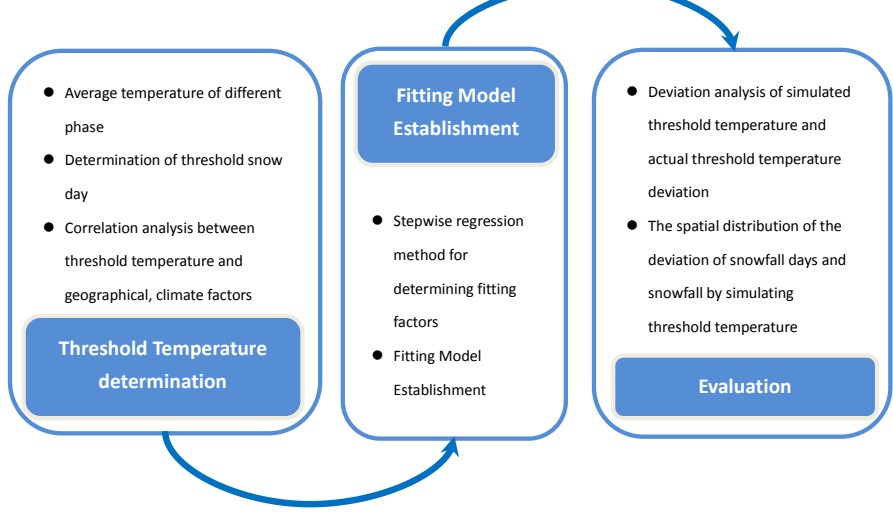

**FIG.2. Technical roadmap**












**3. Threshold temperature**

3.1 Daily mean temperature corresponding to precipitation in different phases

There are three types of precipitation phases in northern China: snowfall, rainfall and sleet. Most of the time, snowfall occurs in winter, rainfall occurs in summer, and snow, rain, and sleet can occur during the autumn and spring. Fig. 3 and Table 2 show phase temperature distribution of precipitation events at the stations. The total precipitation events at 324 stations were included in the statistical calculations, and their corresponding daily mean temperature values (Fig. 3a) were examined: only snowfall occurred when the daily mean temperature was below -12.9 ℃; only rainfall occurred when the daily mean temperature was higher than 22.1℃; and the three phases of snow, rain, and sleet occurred when the temperature was between -12.9℃ and 22.1℃.

In northern China (Fig. 3a) pure snow (snowfall) events occurred when the daily mean temperature was below 8.5 ℃, and 95% of the snowfall events occurred when the daily mean temperature was less than 2.7 ℃ and higher than -16.6 ℃. All pure rain events (rainfall) occurred when the daily mean temperature was higher than -4.9 ℃, and 95% occurred when the temperature was lower than 26.0℃ and higher than 6.4 ℃. All sleet events appeared in the temperature range of -12.9–22.1 ℃, with 95% occurring when the daily mean temperature was lower than 8.3℃ and higher than -1.6 ℃.




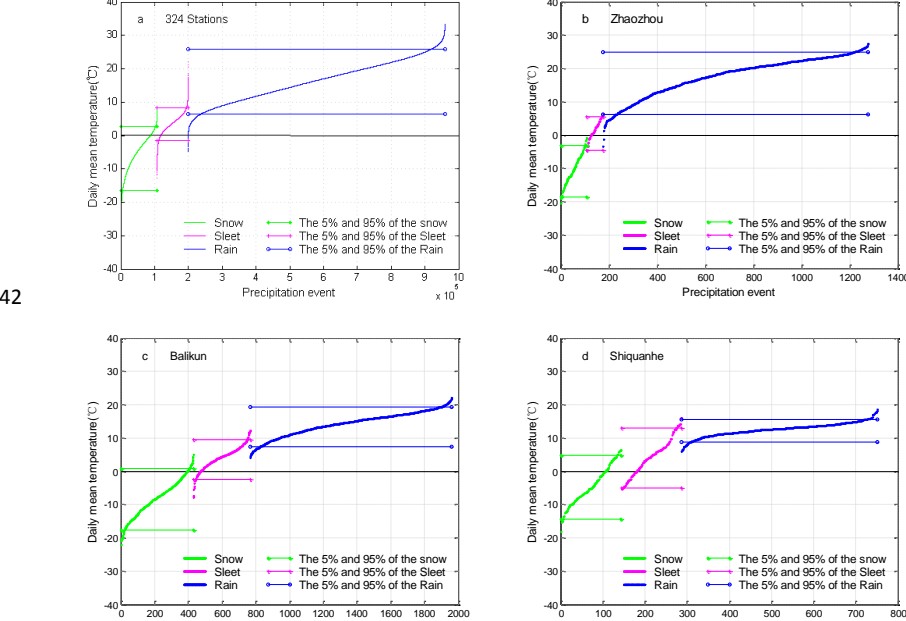


**FIG.3. Precipitation phase temperature distribution of regional average and representative stations (a-324**
**stations; b-Zhaozhou; c-Balikun; d-Shiquanhe)**

At Zhaozhou station (Fig. 3b), the pure snow events all occurred when the daily
mean temperature was lower than -0.9 ℃, pure rainfall occurred when the daily mean
temperature was higher than 3.4 ℃, and sleet occurred in case of -4.5–6.5 ℃.
Zhaozhou station had the lowest threshold temperature of snowfall and rainfall in the
study region. At Balikun station (Fig. 3c), the pure snow events all occurred when the
daily mean temperature was lower than -5.1 ℃, pure rain events occurred when the
daily mean temperature was higher than 4.1 ℃, and sleet occurred within a
temperature range of -7.8–12.3 ℃. At Shiquanhe station (Fig. 3d), the pure snow
events all occurred when the daily average temperature was lower than 6.4 ℃, pure
rainfall occurred when the daily mean temperature was higher than 6.1 ℃, and sleet





occurred when the temperature was from -3.3 ℃ to 16.0 ℃. Shiquanhe station had the
highest threshold temperature of snowfall and rainfall in the whole region.
Pure snowfall occurred when the daily mean temperature was above 0 ℃, and
pure rainfall occurred when it was below 0 ℃. This may be because the daily mean
temperature is higher/lower than instantaneous air temperature when snowfall/rainfall
occurs, or the instantaneous air temperature is below/above 0 ℃ with
warming/cooling after snow/rain. It could also be because the snowflakes are formed
in the upper atmosphere with the lower temperature, the temperature near the surface
cools faster due to the intrusion of extremely cold air, and they are not fully melted
when they fall and still exist in the form of snow. In the lower atmosphere layer
(below 3000 m), there is a lot of super-cooling water, and the air temperature is in the
range of 0 – -15 ℃. With a rich condensation nucleus, an abundance of moisture, and
a lack of a freezing nucleus (the ice nucleation), raindrops can form below 0 ℃,
producing glaze or rime on the ground surface.

**Table 2 The distribution range of daily mean temperature under different phases of precipitation at stations**

| Station | | All | Zhaozhou | Balikun | Shiquanhe |
|---|---|---|---|---|---|
| | Maximum | 8.5 | -0.9 | 5.1 | 6.4 |
| Snow day | Minimum | -35.4 | -20.5 | -22.2 | -18.1 |
| temperature | Average | -5.2 | -10.2 | -8.2 | -4.4 |
| (℃) | 5% value | -16.6 | -18.6 | -17.6 | -14.3 |
| | 95% value | 2.7 | -3.3 | 0.8 | 4.8 |
| | Maximum | 22.1 | 6.5 | 12.3 | 16.0 |
| Sleet day | Minimum | -12.9 | -4.5 | -7.8 | -5.3 |
| temperature | Average | 3.6 | 1.6 | 4.1 | 4.3 |
| (℃) | 5% value | -1.6 | -4.5 | -2.5 | -5.0 |
| | 95% value | 8.3 | 5.5 | 9.5 | 13.1 |
| Rain day | Maximum | 33.3 | 27.5 | 22.1 | 18.7 |



| temperature | Minimum | -4.9 | -3.4 | 4.1 | 6.1 |
|---|---|---|---|---|---|
| (℃) | Average | 16.3 | 17.8 | 14.3 | 12.6 |
| | 5% value | 6.4 | 6.1 | 7.3 | 8.7 |
| | 95% value | 26.0 | 25.0 | 19.4 | 15.7 |


It can be seen from Fig. 3 and Table 2 that there is a larger difference of the
maximum temperature of snowfall (extreme threshold temperature of snowfall) and
the minimum temperature of rainfall (extreme threshold temperature of rainfall)
among the stations.
Statistics on the maximum daily mean temperature of all snowfall at each station
(Tsm) and the minimum daily mean temperature of all rainfall at each station (Trn) is
shown in Fig. 4, with Fig. 4a indicating the spatial distribution of maximum daily
mean temperature of snowfall, Fig. 4b the minimum rainfall daily mean temperature
of rainfall, Fig. 4c the average daily mean temperature of sleet, and Fig. 4d the
difference of the maximum daily mean temperature of snowfall and minimum daily
mean temperature of rainfall (Trm-Trn). There is a common spatial distribution
feature in the maximum daily mean temperature of snow day, minimum daily mean
temperature of rain day, and the average daily mean temperature of sleet day in
northern China, with the high values generally in the Tibetan Plateau and southern
Xinjiang, while the low values mostly in eastern and northern Xinjiang. In the stations
analyzed, most have a relationship of Trn<Tsm, that is, the minimum daily mean
temperature at the time of a rain event is lower than the maximum daily mean
temperature at the time of a snowfall event. Only in a few of places in Northwest Arid
Region, is the maximum daily mean temperature of a snow day lower than the





minimum daily mean temperature of a rain day, that is, pure rain and snow events do
not overlap.

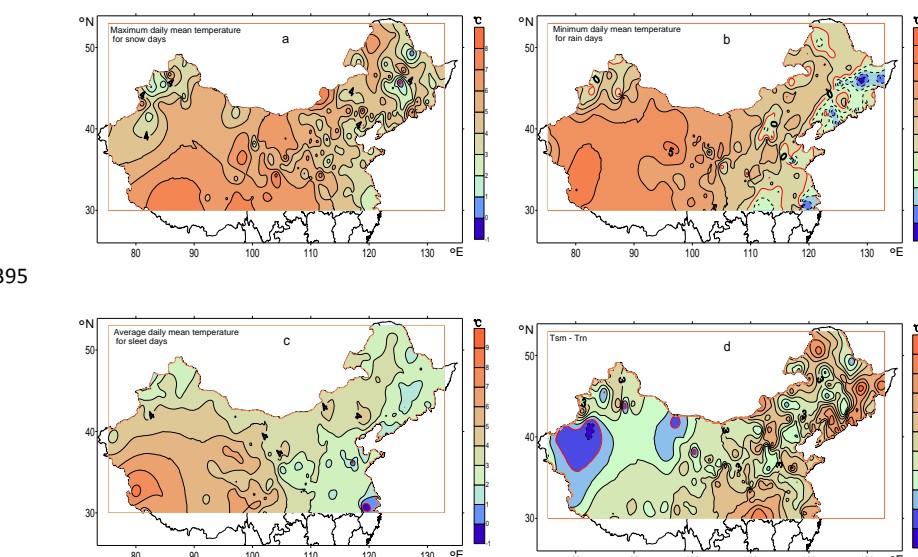


**FIG.4. The distribution of daily mean temperatures when precipitation occur (a. maximum daily mean**
**temperature of snow day; b. minimum daily mean temperature of rain day; c. average daily mean temperature**
**of sleet day; and d. difference snow day maximum daily mean temperature and rain day minimum daily mean**
**temperature) (Red thick line represents 0℃ isotherms)**

3.2 Threshold temperature determination

The threshold temperature is determined directly by daily mean temperature of

various precipitation days, and the calculation steps are as follows: First, the number
of snow days (Sn) and the number of rain days (Rn) between Trn and Tsm is
calculated, and the total number of the rain and snow days (Nsr = Sn + Rn) between
Trn and Tsm is also calculated. Second, the daily mean temperature of Nsr is
calculated and ranked in ascending order. Last, the average of daily mean temperature
of the $Sn^{th}$ day and the $(Sn+1)^{th}$ day is calculated, and it is taken as the threshold



temperature (Tt-d) of the rain and snow days. For the area where pure rain and snow
events do not overlap, the average of the maximum daily mean temperature of snow
day and the minimum daily mean temperature of rain day is taken as the threshold
temperature (Tt-d). The average of Tt-d and the daily mean temperature of sleet day is
taken as the Tt-d when Tt-d is not in the range of sleet day daily mean temperature.
The Tt-ds values in this study are all within the daily mean temperature of sleet day,
however, and this operation is not required.

Figure 5 shows the distribution of the relative deviation of the snow days and

snowfall in northern China, determined by the threshold temperature as mentioned
above, to the actual snow days and snowfall counted by using weather phenomenon
records. The relative deviation of snow day was smaller. This is due to the definition
of threshold temperature being directly determined by snow-day mean temperature.
Since the daily mean temperature of the $Sn^{th}$ day and the $(Sn+1)^{th}$ (or more) day is the
same under this definition, however, there will be a slight positive bias in the
threshold temperature of the same temperature day, with a range of relative deviations

(0, 2.3%).

The spatial distribution of the relative deviation of the snowfall was mainly

positive, which is due to the systematic deviation of the method. Larger deviation
appeared in the Qinghai-Tibetan Plateau and the Yangtze-Huaihe River Basins. These
areas have more precipitation and sufficient water vapor. Under the same water vapor
condition, the observed rainfall was greater than the observed snowfall, and the
amount of snowfall determined by the threshold temperature was slightly large, with





the certain sites even larger. Small values occurred in the southeastern Northeast
China, the border zone between Inner Mongolia and Xinjiang, and western Xinjiang,
with the main reason related to the less precipitation and insufficient water vapor.
Overall, the relative deviation of snowfall is between -5% and 20%. There were 312
stations (more than 96%) whose deviation was less than or equal to 10%, and the
absolute value of the relative deviation was less than 5% in most areas.

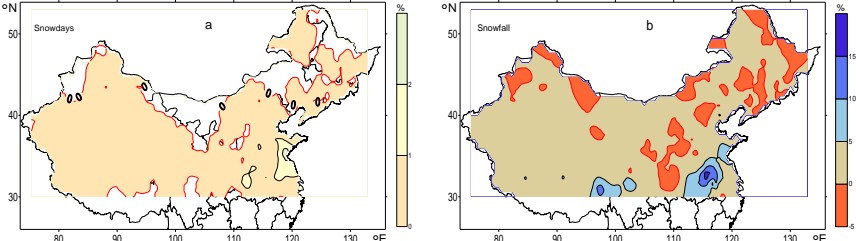


**FIG.5.   The spatial distribution of the relative deviation of the days (a) and amount (b) of snowfall determined**
**by the threshold temperature (Tt-d) in northern China**

The spatial distribution of the threshold temperature (Tt-d) of rain and snow at the
stations north of 30 °N are shown in Fig. 6. The average Tt-d is 2.3 °C for Eastern
Monsoon Region, 3.4 °C for Northwest Arid Region, and 5.2 °C for the
Qinghai-Tibetan Plateau. The highest threshold temperature of the study region is
6.3 °C (Shiquanhe, Fig. 3d), the lowest is -1.2 °C (Zhaozhou, Fig. 3b), the threshold
temperature range was 7.5 °C, and the average threshold temperature for the whole
region was 2.81 °C. The high-value area was in the northern Qinghai-Tibet Plateau,
with a threshold temperature of more than 4 °C, and the low-value areas were
generally in eastern Northeast China, North China, and northern Xinjiang with the
threshold temperature less than 2 °C. The threshold temperature east of 90 °E


decreased from west to east, and it decreased from east to west in areas west of 90 E.
On the whole, the west of 105 E showed an approximately zonal distribution, and the
threshold temperature decreased with the increase of latitude; the east of 105 E had a
meridional distribution, and the threshold temperature decreased with increasing
longitude.

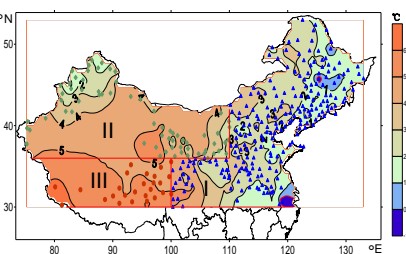

**FIG.6. Spatial distribution of threshold temperature of precipitation phases in northern China (I: East**
**Monsoon Region; II: Northwest Arid Region; III: Qinghai-Tibetan Plateau. Unit: ℃)**

This distribution feature was well consistent with the spatial pattern of the
maximum daily mean temperature of snow days (Fig. 4a), the minimum daily mean
temperature of rain days (Fig. 4b), and the average daily mean temperature of sleet
days (Fig. 4c) previously counted in northern China. It can therefore be considered to
have reflected the actual observations.
3.3 Correlation between threshold temperature and geographical/climatic factors
Because the precipitation records of the major international datasets do not
indicate the precipitation phases, it is necessary to distinguish them outside China by
establishing a statistical model of threshold temperature applicable in the
sub-continental or larger scales.
The spatial distribution of threshold temperature of solid and liquid precipitation



in northern China may be affected by various geographical and climatic factors. Our
analysis found that the threshold temperature (Tt-d) is related to the longitude, latitude,
altitude, annual precipitation, annual mean air temperature, and annual relative
humidity of the observational sites, with a positive correlation with altitude and a
negative correlation with the other factors. All the correlations passed the significant
test at 0.05 level.

Figure 7 shows the changes of the threshold temperature in northern China with

latitude, altitude, annual precipitation, and annual mean relative humidity. In the
lower latitude area, the threshold temperature was generally higher and more disperse,
while in the higher latitude area, it was generally slightly lower and relatively
centralized. The threshold temperature had a clear decreasing trend with increase of
latitude. In lower altitude area, the threshold temperature was lower, while it was
higher in mountains and plateaus, and a highly significant increasing trend of
threshold temperature with altitude can be seen. There was a negative correlation
between the threshold temperature and the annual precipitation, and a more significant
negative correlation with the annual relative humidity.

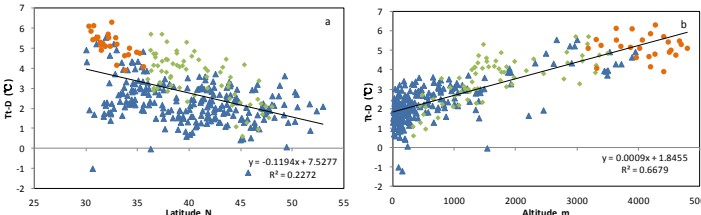




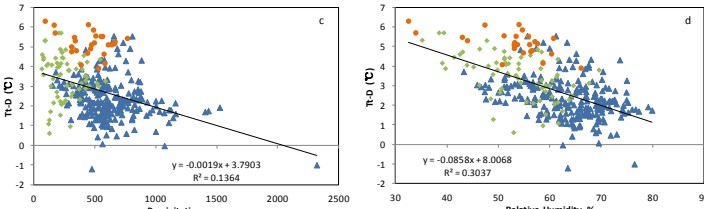

**FIG.7. Relationship of the threshold temperature (Tt-d) with latitude (a), altitude (b), annual precipitation (c) and annual mean relative humidity (d) in northern China**

**(Blue triangle: East Monsoon Region; Green diamond: Northwest Arid Region; Red circle: Qinghai-Tibetan Plateau)**

The threshold temperature decreased with the increase of latitude. This may be mainly related to the occurrences of inversion and the smaller temperature lapse rate in the cold season in high latitudes, which makes the difference between surface air temperature and upper air temperature relatively small, and snowfall is more likely to occur when the surface air temperature is low. In low latitude region or high annual mean temperature area, the cold season inverse temperature phenomenon is scarce, the temperature lapse rate is larger, the temperature difference between surface and upper layer is large, and the surface air temperature is often higher when snowfall occurs.

The threshold temperature was positively correlated with altitude, which may mainly be because the ground surface receives stronger solar radiation, causing the boundary-layer atmosphere to heat rapidly in the high altitude areas during daytime. However, the upper air temperature is low, the temperature lapse rate is larger, the cloud bottom-height is low, and the path of snowflakes is short, so snowfall phenomenon can also be observed when the daytime surface air temperature is high.

The threshold temperature was negatively correlated with annual precipitation in





particular with relative humidity, which may be related to the low latent heat flux and
high sensible heat flux in arid area. When the sensible heat flux is high, the ground
surface air temperature is high, and the temperature lapse rate is large. In the case of
the same condensation height or cloud bottom-height, snowfall is more likely to occur
under the condition of higher surface air temperature.
3.4 Establishment of the threshold temperature model

Considering that the relative humidity data of some areas is difficult to obtain, the

precipitation factor was selected as the independent variable. Using the SPSS software
stepwise regression analysis method, a statistical model of threshold temperature was
established with latitude, altitude, and annual precipitation as influential factors. The
model, which passed the significant test at the 0.05 level, can be expressed as follow:

$T_{t-p} = 6.81576376 + (-.09305) * N + (.000567) * H + (-0.00182) * R$          (1)

where $T_{t-p}$ is the simulated threshold temperature (℃), N is the latitude of the station,
H is the altitude of the station (m), and R is the annual precipitation of the station
(mm).

The correlation coefficient between $T_{t-p}$ and $T_{t-d}$ (threshold temperature

determined by using the synoptic phenomena) is 0.87. The median and standard
deviation of the simulated threshold temperature ($T_{t-p}$) were 2.53 and 1.16, which
were close to the median (2.64) and standard deviation (1.33) of the $T_{t-d}$. The
maximum simulated threshold temperature was 6.05 ℃, minimum was -0.22 ℃,
temperature range was 6.26 ℃, and average simulated threshold temperature was
2.81 ℃ for the whole region. The maximum positive deviation of the $T_{t-p}$ to the $T_{t-d}$



was 3.0 ℃, and the minimum negative deviation was -1.7 ℃. The stations, at which
relative deviation of snow day and snowfall were less than 10%, reached 95% and 91%
of the total, respectively.

In the East Monsoon Region (Region I) and the Northwest Arid Region (Region

II), the simulated threshold temperature was generally lower than the Tt-d (0.005 ℃
lower in Region I on average, and 0.02 ℃ lower in Region II on average). However, it
was higher in the Qinghai-Tibetan Plateau Region (0.097 ℃ higher on average) (Fig.

8).

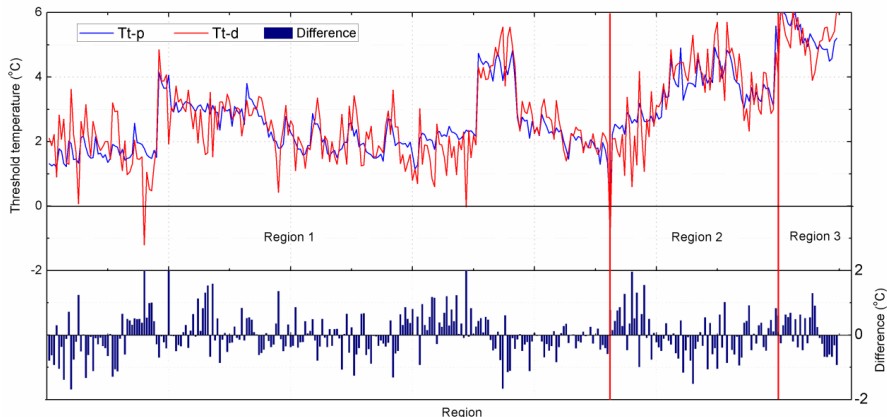

**FIG.8. Simulated threshold temperature (Tt-p), actual threshold temperature (Tt-d) and their difference for**
**observational stations in different regions of northern China (1: East Monsoon Region; 2: Northwest Arid**
**Region; 3: Qinghai-Tibetan Plateau Region)**

The correlation coefficients of the standard deviation and median of the snowfall

days (simulated snowfall days) with those of actual snowfall days at all the stations
were 0.92 and 0.94, respectively. The differences of the standard deviation and
median of the simulated snowfall days and actual snowfall days are smaller overall,
and the differences of the median is slightly larger in the Qinghai-Tibet Plateau where





there was more snowfall. Fig. 9 shows spatial distribution of the relative deviation of
the simulated snow days (Fig. 9a) and snowfall (Fig. 9b) relative to the actual snow
days and snowfall at the stations. The relative deviation range of snowfall days in
northern China was between -21.17% and 18.38%, with an average of -0.12%; the
relative deviation was smaller in mid-southern parts of the study region, and larger in
the coastal areas and the northern Qinghai-Xizang Plateau. In the Qinghai-Tibet
Plateau Region, the medians of the simulated snow days were smaller than those of
the actual snow days, and the relative deviations were larger. This may be related to
the fact that the snowfall days in northern Tibetan Plateau fluctuated greatly, and there
are some years with larger numbers of snowfall days. The relative deviation range of
snowfall in the whole region was between 17.3% and 30.38% with an average of
1.09%, and the spatial distribution was basically the same as that of the relative
deviations of snow days.

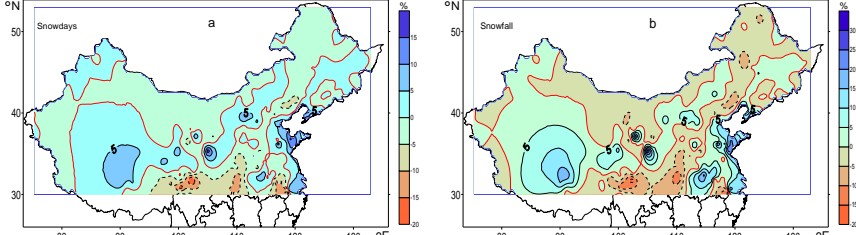


**FIG.9. Deviation distribution of snowfall days (a) and snowfall (b) defined by the simulated threshold**
**temperature**

Affected by the extremely low air temperature and the abnormally deficient water
vapor due to the East Asian winter monsoon, the pure snow days (snowfall) with only
snowfall weather phenomenon were relatively less frequent (low) in northern China;





therefore, it is more likely that the relative deviation is large in the study region.
However, the relative deviation range shown here is acceptable, and the fitting effect
is generally good.
The MSRE of the relative deviation of snow days was 3.9, and the MSRE of the
relative deviation of snowfall was 5.3. The annual snow days and the amount of
snowfall were less in the mid-southern parts of the study region which had negative
relative deviations of the simulated snow events; however, snow days and snowfall
were slightly more numerous in the northern part of the Sichuan Basin. The number
of snow days and snowfall was less in the coastal area which had positive relative
deviations of the simulated snow events, while there were more snow days and
snowfall in the northern Qinghai-Xizang Plateau. The relative deviation of snow days
(snowfall) and the threshold temperature had a correlation coefficient of -0.38 (-0.31);
both passed the significant test at 0.05 level. It can be seen that the relative deviation
in the area with low threshold temperature tends to be positive, and the relative
deviation in area with high threshold temperature is generally negative.









## 4. Comparison with previous works


Previous researches used the insurance probability to obtain the threshold
geophysical parameters of the snow-rain separation (e.g. Han et al., 2010; Sims and
Liu, 2015). Sims and Liu (2015) found that the wet-bulb temperature and low-layer
temperature lapse rate had the most significant influence on the precipitation phase,
with a lapse rate of $6^\circ$ C km$^{-1}$ resulting in an 86% insurance probability of solid
precipitation if the near-surface wet-bulb temperature was around $0^\circ$ C. Surface air
pressure also exerted an influence on precipitation phase in some cases. However, the
climatic parameters are once again less available in the major international historical
climate datasets, though the finding and the method recommended are valuable in
investigating into local and regional precipitation phrases.
For comparison of snow-day mean temperature method and insurance probability
method as reported in Han et al. (2010), the number of snow days (Sn) and rain days
(Rn) between Trn and Tsm was calculated, respectively. The corresponding daily
mean temperature at the insurance probability of the snow and rain days between [Trn,
Tsm], X (x $\in$ (0–99%)) (at 1% intervals), was estimated. For example, the number of
rain days and snow days between Trn and TSM is 100d respectively; when x = 90% is
taken, the rain day temperature Tr90 corresponds to the insurance probability of 90%,
that is, to ensure the minimum daily mean temperature in the event of 90% rain days
between Trn and TSM, while Ts90 is to guarantee that the maximum daily mean
temperature in the event of 90% snowfall days is between Trn and TSM. The
arithmetic mean of each station's Trx and Tsx is defined as the threshold temperature



Tt-x at the station's insurance probability x.

The threshold temperature (Tt-x) was calculated according to the insurance

probability method, and the threshold temperature (Tt-d) was obtained based on the
definition in this paper; the relative deviation comparison is presented in Table 3. For
simplicity, the insurance probability interval in the table was taken as 10%. The
maximum, minimum, and range of the threshold temperature (Tt-x) under different
insurance probability, and of the (Tt-d), in northern China, are given in the table; at
the same time, the maximum, minimum, and range of the relative deviation of the
snow days and snowfall, as well as the number of stations with a relative deviation
less than or equal to 10%, are also given.

**Table 3 Comparison of statistics and the relative deviations resulting from threshold temperature Tt-x and**
**Tt-d**

|  | Threshold temperature (℃) | | | Relative deviation of snow days (%) | | | | Relative deviation of snowfall (%) | | | |
|---|---|---|---|---|---|---|---|---|---|---|---|
|  | max | min | max-min | max | min | max-min% | Stations <10% | max | min | max-min% | Stations <10% |
| Tt-0 | 6.4 | −2.3 | 8.7 | 30.2 | −11.1 | 41.3 | 311 | 36.6 | −15.3 | 51.9 | 280 |
| Tt-10 | 6.4 | −2.3 | 8.7 | 25.1 | −11.1 | 36.3 | 313 | 29 | −11.8 | 40.8 | 284 |
| Tt-20 | 6.5 | −2.3 | 8.8 | 25.1 | −9.5 | 34.6 | 316 | 29 | −9.7 | 38.7 | 287 |
| Tt-30 | 6.5 | −2.2 | 8.7 | 23.6 | −7.1 | 30.7 | 314 | 31.5 | −15.3 | 46.8 | 287 |
| Tt-40 | 6.4 | −2.2 | 8.6 | 23.6 | −5.8 | 29.4 | 316 | 31.5 | −8.4 | 39.9 | 289 |
| Tt-50 | 6.5 | −2 | 8.5 | 21.1 | −5.7 | 26.8 | 312 | 32.2 | −9.7 | 41.9 | 286 |
| Tt-60 | 6.4 | −1.5 | 7.9 | 19.1 | −6.5 | 25.6 | 313 | 32.2 | −9.7 | 41.9 | 289 |
| Tt-70 | 6.4 | −1.4 | 7.8 | 15.6 | −6.5 | 22.1 | 314 | 30.2 | −6.2 | 36.4 | 283 |
| Tt-80 | 6.7 | −1.4 | 8.1 | 18.3 | −5.8 | 24 | 307 | 45.2 | −8.4 | 53.6 | 282 |
| Tt-90 | 6.5 | −1.2 | 7.7 | 23 | −7 | 29.9 | 306 | 33.4 | −9.7 | 43.1 | 276 |
| Tt-d | 6.3 | −1.2 | 7.5 | 2.6 | 0 | 2.6 | 323 | 20.2 | −4.3 | 24.5 | 312 |


Table 3 shows that, using the insurance probability method, the test results of the

threshold temperature (Tt-70), obtained when the insurance probability x = 70% was



taken, represented the best values, as the difference between the minimum and
maximum values of the threshold temperature was small, and the relative errors were
small, with the relative deviation of the snow days at 314 stations ≤10%, and that of
the snowfall at 283 stations ≤ 10%.
The range of threshold temperature Tt-d of snow days determined in this paper
was less than that of the Tt-70. The relative deviation of snow days was obviously
small, and the relative deviation of snowfall was much less than that of the Tt-70,
with more stations having the relative deviations ≤10% for both snow days and
snowfall. Therefore, the method developed in this paper has an advantage over the
insurance probability method developed in the previously works.














## 5. Discussion

China has a vast territory. The study region across the latitude range 30–54°N, and a longitude range of 73–136°E, with various climate types of temperate monsoon zone, continental arid zone and alpine including the highest mountainous system of the Qinghai-Tibetan Plateau. The complex and diverse geophysical and climatic condition makes the region ideal for understanding the transition of precipitation phrases and developing a method to separate the different precipitation phrases.

We made an attempt to develop such a method to separate the precipitation phases by using a high-quality daily observational dataset in this paper. Our study not only determined the threshold temperature with more reliable results, but also tested the statistical model of threshold temperature, provided the results of the model and the relative deviation range for different regions, and confirmed the applicability of the method in the complex geographic area with diverse climate types.

With the method of determining threshold temperature developed in this paper, the relative deviation of snow days and snowfall calculated for most of the stations was very small, and the stations with less than 10% relative deviations accounted for 95.1% and 90.7%, respectively. This method could be used to better determine the snow days than the snowfall, with the relative deviation of snowfall was slightly larger in the Huaihe River basin. This is mainly because, when using the threshold temperature to calculate the amount of snowfall, rain days with a daily mean temperature below the threshold temperature could be identified as the snow day, and also some snow days with a daily mean temperature above the threshold temperature



could be classified as rain days. In the frequent transformation of the precipitation
phases (early spring and early winter), precipitation on a rain day is often greater than
that on a snow day, so the priority to ensure the determination of a snow day, the
estimated relative deviation of snowfall would be a little larger.

In this paper, only the two phases of pure snowfall and pure rainfall were

determined, however, and the sleet was not analyzed. In the case of sleet, the surface
air temperature changed greatly during a day; there was probably sleet, pure rain and
pure snow in the same day, the actual threshold temperature fluctuations were large,
and it would be difficult to accurately determine and simulate. Because the method
used in this paper did not quantify the sleet, when precipitation was separated into
solid and liquid state, the sleets will be classified as snow when the daily mean
temperature is lower than the threshold temperature, and as rain when the daily mean
temperature is higher than the threshold temperature, causing a certain error. However,
for the study of large-scale snowfall climatology, especially for studies of the larger
than subcontinental scale snowfall climate change, the snow and rain separation
method presented in this paper could well meet the needs.










6**. Conclusions**

Based on the analysis of the historical daily temperature, precipitation, and

weather phenomenon observation data in northern China, the threshold temperature
model for determining the phase of rain and snow was established and tested. The
main conclusions are as follows:

(1) The threshold temperature value of rain and snow determined based on

weather phenomenon data is between -1.2–6.3 ℃, with a temperature range of 7.5 ℃
and an average value of 2.81 ℃. The high values were in the northern
Qinghai-Tibetan Plateau, reaching more than 4 ℃, and the low values were found in
Northeast China, North China, and northern Xinjiang Autonomous Region, generally
less than 2 ℃. The west of 105 E showed an approximately zonal distribution, and
the threshold temperature decreased with latitude; the east of 105 E had a meridional
distribution, and the threshold temperature decreased with increasing longitude.

(2) The threshold temperature was more variable in the low latitude areas, while

it was slightly lower and relatively centralized in the high latitudes, with a clear
decreasing trend with increase of latitude. The threshold temperature was lower at low
altitudes, higher in the high altitude areas, and had a trend to increase with altitude.
There was a good negative correlation between the threshold temperature and annual
total precipitation and annual mean relative humidity, with the negative correlation
with relative humidity specially significant.

(3) A statistical model based on latitude, elevation, and annual precipitation can

be used to simulate the threshold temperature of the precipitation phase in northern



China, with less relative deviation in simulated snow days and snowfall. The stations
with relative deviation less than 10% reached 95.1% and 90.7% for the snow days and
snowfall respectively.


**Acknowledgements:** This study is financed by the China Natural Science Foundation
(NSC) (Fund No:41575003) and the Ministry of Science and Technology of China
(Fund No: GYHY201206012).





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
