# Peer review of "A new method to separate precipitation phases"

_Hydrology and Earth System Sciences, 2018_

## Referee Comment (RC1) · Anonymous Referee #1 · 28 Aug 2018

The authors analyzed the range and distribution of the threshold temperature of rain and snow based on the CMA daily data, and developed a statistical model to discriminate rain and snow based on the relationship between threshold temperature and latitude, elevation, and annual precipitation. This study provides a practical method to separate precipitation types. However, the introduction needs to be adjusted, and the relationship between threshold temperature and other variables needs more consideration and discussion.

Major Revisions:

1. The title of the paper is too broad. It is suggested to be more specific.

2. The number of key words is too many. Please revise it according to the requirement of the journal (usually not more than 5-6 key words).

[Figure]

3. L120-L185: (a) These five paragraphs are all about the precipitation type discrimination schemes. The L120-l134 part introduces some precipitation type discrimination methods based on surface variables (air temperature, pressure, dewpoint, relative humidity, and wet-bulb temperature), the following L135-155 part presents the methods based on surface air temperature and upper air temperature, the L156-L162 part introduces the methods based on vertical structure of atmosphere, and the L163-L180 part presents the surface variable based methods again (single threshold temperature and double threshold temperature), and finally analyzes the difficulty of humidity based method (which is also surface variable based method). These parts of content are about various kinds of method and mixed together. The structure of these paragraphs needs to be adjusted.

(b) L120-L185: This part has reviewed the various methods of separating different precipitation types. It is surprising that the recent work about precipitation type discrimination (Ding et al., 2014) has not been cited in the paper, which finds out that the precipitation type is dependent on the temperature, elevation, and relative humidity (i.e., the threshold temperature for rain and snow increases with the increasing of elevation, and the probability of sleet increases when relative humidity increases), and develops a dynamic threshold scheme to discriminate rain, sleet, and snow.

Ding, B., K. Yang, J. Qin, L. Wang, Y. Chen, and X. He, 2014: The dependence of precipitation types on surface elevation and meteorological conditions and its parameterization, J. Hydrol., 513, 154-163, doi:10.1016/j.jhydrol.2014.03.038.

(c) L153-155: This sentence is a little abrupt here. It seems not closely related to the last sentence or paragraph, and also does not have any reference cited for the sentence.

4. Section 2 "Data and methods" could be divided into two sub-sections like "2.1 Data" and "2.2 Methods (or data processing)". Further, the flow diagram of the analysis needs to be explained more clearly.

5. L335, L338, and L339: Actually, these three values are 90 percent instead of "95 percent", since the authors gave the percentages of snowfall, rainfall, and sleet between the 5 percent and 95 percent of the event profile, respectively.

6. (a) In L349, the minimum daily mean temperature of rainfall at Zhaozhou station was 3.4°C; however, it was -3.4°C in Table 2.

(b) The maximum daily mean temperature of snowfall at Balikun station was -5.1°C in L352, but 5.1°C in Table 2.

(c) The minimum daily mean temperature of sleet at Shiquanhe station was -3.3°C in L357, but -5.3°C in Table 2.

Please correct them, and also check the other numbers in the paper carefully.

7. Fig. 4d, if reversing the positive and negative, the spatial distribution of (Trn-Tsn) would be also similar to those of maximum daily mean temperature of snowfall (Fig. 4a), minimum daily mean temperature of rainfall (Fig. 4b), and the average daily mean temperature of sleet (Fig. 4c).

8. Fig.5, about the positive deviation of the snowfall amount by the threshold temperature (Tt-d), is there any possible that because there is occasionally snowfall occurred at high temperature in these areas, which may pull up the threshold temperature by the few snowfall events? If the temperature range to settle the threshold is enlarged (larger than Tsn ∼ Trn), for example, ranking from (Trn-2)°C to (Tsn+2)°C, and settling the threshold temperature from this ranking profile, maybe this deviation could be reduced.

9. (a) L452: the authors stated that "and it decreased from east to west in areas west of 90°E", however, from Fig. 6, it seems the threshold temperature decreased from south to north in the areas west of 90°E. (b) Actually, the L451-L452 sentence has the totally same meaning with the L453-L456 sentence, only with the different demarcations of longitude values (i.e., 90°E and 105°E, respectively).

10. In Section 3.3, the authors stated that the threshold temperature (Tt-d) is related

to longitude, latitude, altitude, annual precipitation, annual mean air temperature, and annual relative humidity. However, from R2 in Figure 7, altitude is most closely related to the threshold temperature (Tt-d), and then the relative humidity is. Although latitude and longitude showed some negative relationship with the threshold temperature, actually, from Fig. 6 "the distribution of the threshold temperature", the relationship between the threshold temperature with longitude and latitude is also related to that with altitude. Since in China, the mean altitude decreases from west to east and from south to north, as well as the decreasing trend of threshold temperature with decrease of altitude, that is why threshold temperature decreases with the increase of the longitude and the increase of the latitude.

11. Also in Section 3.3, I think another explanation is more reasonable for the negative relationship of threshold temperature with the relative humidity. In the arid area, where the relative humidity is lower, when the snow particle is falling from the same condensation height, since the relative humidity of atmosphere is lower, the difference of specific humidity between the particle surface and the air is larger, and therefore the heat absorbed by the snow particle is easier to be released in the form of latent heat flux. In that case, it needs more energy, i.e., higher surface air temperature, to melt the snow particles. That is why the threshold temperature is relative higher in the arid region (i.e., lower relative humidity region).

12. From the above Question 10, the threshold temperature has close relationship with altitude, but not latitude or longitude. Beside, from R2 in Fig. 7c, the relationship between the threshold temperature and the annual precipitation is also relative low. Therefore, the statistical model of threshold temperature derived from latitude, altitude, and annual precipitation (Equation 1) needs more consideration. Actually, from Equation 1, the impact from R (annual precipitation) on Tt-p (threshold) is relative low comparing with N (latitude) and H (altitude). And the impact of N could be also combined to H.

13. L575: What is MSRE short for? Do you mean RMSE?

14. L584-L585: from Figure 9, it seems the deviation is generally positive, no matter the low threshold temperature area or the high threshold temperature area.

Minor Revisions:

1. Please check the references. The reference "Stefan et al. (2008)" (L123) has not been included in the reference list.

2. L146: phrases => phases

3. L193: was = > were

4. L211-L214: daily data of 839 national stations' air pressure, surface air temperature (daily mean, daily maximum and daily minimum), precipitation, evaporation, relative humidity, wind speed, sunshine hours, and 0-cm ground temperature => daily data of air pressure, surface air temperature (daily mean, daily maximum and daily minimum), precipitation, evaporation, relative humidity, wind speed, sunshine hours, and 0-cm ground temperature from 839 stations

5. L228-L229: greater than or equal to 1 mm => no less than 1 mm

6. L233: sleet, snow => sleet, and snow

7. L234: registered => recorded

8. L247: isotherm of in 3°C => isotherm of 3°C

9. L336: less => lower

10. L361: than instantaneous air temperature => than the instantaneous air temperature

11. Fig 4: (a) the symbol "Tsm" and "Trn" is suggested to be added in the figure caption and in the figures to be more clearer; (b) the caption for Figure 4d could be revised as "difference between the maximum daily mean temperature on snowfall day and the minimum daily mean temperature on rainfall day" or "difference between Tsm and Trn";

(c) L397: occur => occurs

12. L384: Trm-Trn => Tsm-Trn

13. There are several different spellings as "Qinghai-Xizang Plateau" (L557), "Qinghai-Tibet Plateau" (L557-L558), and "Qinghai-Tibetan Plateau" (L539). They need to be unified.

14. L660 (two places): phrases => phases
* * *

---

## Referee Comment (RC2) · Anonymous Referee #2 · 29 Aug 2018

**Overview**

The authors present a detailed analysis of rain-snow partitioning over northern China using meteorological measurements and precipitation phase observations from several hundred research stations. They show marked spatial variability in the rain-snow air temperature threshold, which is generally highest near the Tibetan Plateau and lowest in the northeast part of the country. They found that the threshold was correlated with latitude, longitude, relative humidity, and precipitation across their study domain. The authors then used those variables, minus longitude and relative humidity, in a stepwise multiple linear regression to predict the rain-snow air temperature threshold. The regression performed well relative to observations.

**Major issues**

Overall, this work adequately presents spatial variation in rain-snow partitioning using a robust meteorological dataset. However, I would not accept this paper in its current form. First, the paper is titled "A new method to separate precipitation phases." My contention with this is that the method is a multiple linear regression that cannot be transferred to other regions. The authors use latitude as a predictor variable instead of other meaningful physical quantities that might be transferrable in space. Because of this shortcoming, their method could not be used in other geographic regions (Europe, North America), thus limiting its utility. Froidurot et al. (2014) presents regression methods that use more meaningful independent variables.

Additionally, there is not much novelty in the work the authors present. For one, spatial variability in rain-snow partitioning has already been described at local (Wayand et al., 2016), regional (Rajagopal and Harpold, 2016), continental (Ye et al., 2013), hemispherical (Jennings et al., 2018), and global (Dai, 2008) scales. In this context, the finding that rain-snow air temperature thresholds vary over large distances is unsurprising. Secondly, the authors relate this variability to relative humidity and altitude, which has been covered in depth by previous authors. I feel that this work may be considered more of a case study than a significant contribution to hydrologic science.

There is novel research that can be done with the datasets the authors have at their disposal. For example, figure 3 presents interesting differences across the regions in rain-snow partitioning and mixed-phase events. However, most of the work presented is not currently suited to the high standards of HESS. I would therefore either recommend major revisions or rejection depending on the opinions of the Associate Editor and other reviewers.

**Throughout**

Given HESS's large international audience, I would recommend working with a translating and copyediting service to clean up the English for clarity. I have noted in my specific comments below certain sentences and paragraphs that need particular attention, but the writing needs improvement before resubmission, if resubmission is suggested by the Associate Editor.

**Specific comments**

Line 1: I would change the title as multiple linear regression is not a new method and the regression only applies to northern China (i.e., it cannot be used in other geographic regions).

Lines 45–48: The motivation could be clearer (i.e., change from snow to rain).

Lines 48–52: Break into two sentences. Clearly define the rain-snow temperature threshold. Additionally, the study mentions other types of temperature (dew point and wet bulb), so always note when it is air temperature.

Line 49: Based on the rest of the article, I'm assuming that weather phenomenon records are visual observations of precipitation phase as in Ding et al. (2014) and Dai (2008). This line should be changed to reflect this (along with all other mentions of "weather phenomenon").

Lines 52–53: The first and second clauses are describing the same thing.

Line 56: Although the en dash is correct here, it might be mistaken for a negative symbol. Perhaps write out the ranges (i.e.,  $-1.2^{\circ}$ C to  $6.3^{\circ}$ C).

Line 57: Do the temperature data go to two significant figures after the decimal point (2.81)? If not, please correct this and all other instances.

Lines 57-58: What is the "actual threshold?" Use consistent terminology throughout.

Line 59: Low or lowest?

Line 62: Do you mean more variable instead of dispersed?

Line 64: Remove semi-colon and split into two sentences.

Lines 66–69: Annual precipitation does not control precipitation phase partitioning. I would assume this was likely an effect of relative humidity based on the other data presented in the work and the research done by other authors.

Lines 74–76: This seems like an important finding, but I'm unsure of what it is. What is a relative deviation of snowfall days? Is that the number of days misidentified as rain when snow was actually occurring? Please use more precise language to clearly convey the findings.

Lines 76–78: You must note that this is only for northern China as the regression would not apply to other areas.

Lines 91–94: I'm not sure what this means. Blizzards? Heavy rain events? Freezing rain? Please clarify.

Line 95: What is a precipitation condition? Depth?

Lines 102–109: What do these studies actually say? Be specific.

Lines 112–116: Break up into two sentences and rewrite for clarity (i.e., there are few direct observations of precipitation phase at the global scale). Also, researchers can partition

precipitation phase, but there are difficulties in doing so at air temperatures near freezing (Ding et al., 2014; Jennings et al., 2018; Stewart et al., 2015).

Lines 120–134: Please give more information on the studies you cite. The readers need to know the relevant conclusions of the papers, not just what was studied. For example, Harder and Pomeroy (2013) showed the psychrometric energy balance was more effective at predicting precipitation phase than air temperature alone.

Line 123: Stefan et al. (2008) should be Kienzle (2008). Please double-check all citations to match HESS's style, which is to give the author's last name (I know this can be tricky as in most western countries the family name comes last, which is not the case in China).

Lines 132–134: The Jennings et al. (2018) paper showed that including humidity improved the predictive capacity of precipitation phase methods over air temperature alone. This was also shown by other authors (e.g., Ding et al., 2014; Harder and Pomeroy, 2013; Marks et al., 2013).

Lines 135–137: International readers will need more info on this event.

Lines 135–155: This information would be best combined with the previous paragraph in a discussion of how precipitation phase can be best predicted. There are air temperature methods and then those that use other meteorological and physiographic quantities. The conclusions presented in lines 139–147 are much more specific than any other introductory material and seem out of place.

Line 148: What is a discriminant index?

Line 150: Dew temperature should be dew point temperature. Correct this throughout paper.

Lines 156–162: You could add a separate paragraph about precipitation phase methods that use atmospheric (i.e., not just surface) quantities to predict rain and snow. Much of this is covered in detail in two review papers (Feiccabrino et al., 2015; Harpold et al., 2017a).

Lines 163–164: Be specific on how rain and snow have differing effects on the land surface (e.g., rapid runoff from rainfall versus winter storage and spring release for snowfall).

Line 164: "Therefore" is not correct here.

Lines 164–180: As with my previous comments, this section should be shortened and cleaned up. Here the authors list different ways of partitioning precipitation phase, but the paragraph is introduced as if it is providing different information. Double and single thresholds are covered in depth in the aforementioned review papers (Feiccabrino et al., 2015; Harpold et al., 2017a).

Line 177: What is an auxiliary indicator?

Line 179: Many gridded climate datasets provide humidity information that can be used to estimate dew point temperature with reasonable accuracy.

Line 180: There are several mentions of larger-scale analyses and how current precipitation phase methods struggle over broad spatial extents. While this is correct, the method the authors provide in this paper is specific to northern China and can also not be applied to large scales outside of the country.

Lines 181–185: This paragraph provides redundant information. It should be edited and combined with previous information on phase partitioning methods.

Lines 181–182: Again, many authors have shown that humidity improves phase prediction over air temperature only methods.

Line 182: Arpold should be Harpold and Keith should be changed to Jennings. Please doublecheck all citations.

Lines 186–189: Yes, China does have diverse climatic and physiographic characteristics. No, the method the authors introduce cannot be applied to other areas in the world because it uses latitude, which is not physically meaningful in the context of precipitation phase.

Line 191: What type of observational data? Be specific.

Lines 193–196: Again, the method can only be applied to northern China where the regression was developed.

Lines 202–204: Many spatially extensive gridded climate and reanalysis products include surface pressure and humidity information.

Lines 207–209: Data availability should be provided in a section at the end of the manuscript.

Lines 208–209: As noted above, weather phenomenon should be changed to precipitation phase observations if that is what is included in the dataset. It should be clear what the dataset contains.

Lines 211–214: For the quantities besides air temperature, please note whether they are daily averages or totals.

Lines 216–218: Were the stations removed or the data removed? It is unclear.

Lines 218–219: Why and how were the latitude and longitude corrected?

Lines 219–220: Are the meteorological stations in the same location as the precipitation phase observations?

Lines 229–232. Remove and combine the first part of the paragraph with the next paragraph (lines 233–239).

Lines 233–239: Sleet is not technically the same thing as a rain-snow mix. I.e., a rain-snow mix could be sleet but there could also be rain and snow in a day without sleet occurring. Change this terminology to mixed-phase events to be more accurate.

Lines 237–239: What is the reasoning behind this?

Lines 240–251: This paragraph is confusing. Are there 324 stations in the analysis or 623 as previously mentioned? I feel like the authors are trying to stay that only stations with a minimum of 100 snowfall days were analyzed.

Line 241: Random is not correct here.

Line 252: What are extreme rain and snow records?

Lines 255–257: This is very confusing. Are you saying the mean temperature during sleet was considered to be the rain-snow air temperature threshold?

Lines 261–270: The choice of representative stations seems arbitrary unless I am missing something. Why not provide summary statistics for the stations in each region? Or, at least provide reasoning for representative station selection. Additionally, the Monsoon Region (I) is compared to the others throughout the paper despite the fact that it has an order of magnitude more stations than II and III. This needs to be addressed. Finally, throughout the paper, please note in which geographic region each representative station is located (i.e., Zhaozhou becomes Zhaozhou-I).

Lines 272–279: The station colors should be included as a legend.

Lines 282–285: I might be misreading this but the "percent deviation of snow days" seems like a poor way of quantifying method success. For example, let's say Station X has 50 snow days and 50 rain days in a year. I could still get a 0% deviation (100% success rate) if my method predicts 50 days of snow on the rain days and 50 days of rain on the snow days even though my method was completely wrong. Different ways of validating rain-snow methods are provided in previously mentioned literature (Ding et al., 2014; Froidurot et al., 2014; Harpold et al., 2017b; Jennings et al., 2018). Please correct me if I read this section incorrectly.

Lines 286–297: This can be shortened significantly. Stepwise regression is not a new/novel technique.

Lines 300–301: What is the snow day mean temperature method? This methods section should be rewritten to provide much more clarity. It should be obvious how the rain-snow air temperature threshold is calculated.

Line 312: Figure 2 confused me more than it helped me. This information, when clearly and logically presented, should be easy to understand by reading the methods section.

Lines 324–326: Remove.

Lines 343–346 (Fig. 3): These plots are great (they present a lot of useful information), but could be improved slightly. First, I would flip the axes as it is generally customary to have the cumulative distribution on the y-axis and the measured variable on the x-axis (air temperature in this case). Second, I would normalize the precipitation events so the scale goes from 0-1 (fractional) or 0-100 (percentage). Third, you only need one legend for the whole figure, not one for each subplot.

Lines 350–351: What is the threshold (same comment applies to line 358)?

Lines 359–370: This paragraph is mostly discussion material and should be moved. Citations should be provided that support the suggestions made by the authors.

Line 372: This table should be flipped so that the stations are listed in the first column and the metrics are in the following columns. I provide an unformatted example below:

|          | Snow day temperature (°C) |         |
|----------|---------------------------|---------|
| Station  | Maximum                   | Minimum |
| Zhaozhou | -0.9                      | -20.5   |

Lines 374–377: This may be the case, but how much precipitation is falling at these extreme values?

Lines 378–384: This reads like a figure caption and should be removed or combined with the Fig. 4 caption.

Lines 385–386: Use the abbreviation once it is introduced (e.g., Tsm, Trn, etc.).

Line 395 (Fig. 4): How were the spatial interpolations performed from the point data? The same color ramp should be used for each figure in this case and only one legend is needed. Also, there is no need to include the parts of China that were not analyzed (i.e., limit the plot to what is shown in the red box).

Lines 403–416: These are methods and should be moved. Additionally, I do not agree that this is the way the rain-snow threshold should be calculated as it seems unnecessarily confusing and arbitrary. The Dai (2008) method would be a preferable easy-to-understand and well-validated way of calculating the threshold.

Lines 417–441: I would remove these paragraphs and figures for three reasons: 1) A different method for calculating the rain-snow threshold should be used (see my comment above); 2) Relative deviation is not the best method for calculating method error (see my comment on lines 282–285); and 3) The findings are not central to the authors' main story.

Line 452: West of 90°E, the threshold decreased from south to north, not from east to west.

Lines 458–460: Please note how the spatial interpolations were calculated. There are large spatial extents with no station data. It may be misleading to present the threshold information in this way.

Lines 461–465: Remove. The threshold was calculated from the observations, so of course it should reflect them.

Lines 467–472: Remove. This is introductory material and the regression cannot be used outside China as it uses latitude as a predictor variable.

Lines 473–477: Combine with following paragraph and remove redundancies.

Line 480: Change disperse to variable.

Line 482: Change centralized to less variable.

Lines 490–492 (Fig. 7): Add legend for figure colors. Latitude needs a degree symbol. Did you check how the regression in c was affected by the extreme precipitation outlier? Why was longitude not plotted? This was given as one of the significant correlates of the threshold temperature.

Lines 495–515: Remove. This is discussion material. Citations are needed if this material is kept.

Line 517: Again, I do not agree that humidity is difficult to obtain. Yes, it is less common than air temperature and precipitation, but it is available from many meteorological stations and gridded climate products. Additionally, figure 7 shows a very weak relationship between precipitation and the threshold, especially considering the extreme outlier I noted above.

Line 520: As I have mentioned throughout, the use of latitude is a large limitation of this method as it means it can only be applied to northern China.

Lines 526–527: The coefficient of variation  $(r^2)$ , root mean squared error, and mean bias would all be more appropriate error metrics to provide. Additionally, were any of the stations removed from the data when computing the regression? Error statistics cannot be calculated reliably if all stations were used. Ideally, the model output should be cross-validated on stations that were removed before calculating the regression coefficients.

Lines 534–535: This is confusingly written. Please rewrite for clarity.

Lines 541–546 (Fig. 8): What is the x-axis on these figures? The region lines should not be the same color as the data.

Lines 547–549: Again, please use more robust error metrics. In this case, the mean bias would be helpful as it would indicate whether snowfall was being over or underpredicted. (It looks like this is shown in the following lines and Fig. 9, so please make this information more clear to the reader.)

Lines 565–568 (Fig. 9): If this is the mean bias of snowfall days and snowfall, please make that clear.

Lines 569–574: I do not understand what this paragraph is trying to say and how it relates to the previous information.

Lines 575–586: Rewrite for clarity. I think the main point of this paragraph is that the method underpredicts snow in high threshold areas and overpredicts in low threshold areas, but I am not certain.

Line 594–641: There are many, many methods for predicting precipitation phase. I am unclear as to why the authors devote nearly 50 lines for comparing the method to one other (there is too much space given here to the Han method). If the authors wish to include this information in their results, they should include other methods and provide a robust comparison. As it stands, I would remove this section or shorten it and add to discussion along with other phase method comparison papers. Particularly relevant is Ding et al. (2014), who also used China Meteorological Administration data in their work.

Lines 654–691: This is more of a summary of the paper than a discussion and should be rewritten. There were parts of the results section that should be moved here (noted in my previous comments). Additionally, the discussion should clearly note major limitations and assumptions, which this does not except for the last paragraph. Finally, the authors should compare their work to that of other researchers. There is a lot of literature on the subject of rain-snow partitioning, none of which is discussed here.

Lines 708–710: Remove.

Lines 711–713: To reiterate, latitude is not a physically meaningful quantity in terms of precipitation phase partitioning. Figure 6 shows that at  $\sim$ 30°N the threshold temperature decreases from 5°C in the west to 0°C in the east. This pattern appears fairly consistently in the data as one moves northward with the only exception being northwest China, where the threshold is low. The authors are likely seeing the effect of altitude and humidity, which is, in some cases, cross-correlated with latitude. The use of latitude throughout this paper is a major weakness that must be addressed.

Lines 713-714: Rewrite to remove redundancies.

Line 715: Change good to statistically significant.

Lines 716–717: Change specially to especially. Additionally, much previous work has shown how relative humidity improves phase partitioning. This is another weakness of the paper that the authors show a strong relationship between relative humidity and the rain-snow temperature threshold, but do not include in their regression (I have noted in previous comments that although relative humidity is less available than air temperature and precipitation, it can still be widely found in ground observations and gridded climate products).

**Review references**

- Dai, A., 2008. Temperature and pressure dependence of the rain-snow phase transition over land and ocean. Geophys. Res. Lett. 35.
- Ding, B., Yang, K., Qin, J., Wang, L., Chen, Y., He, X., 2014. The dependence of precipitation types on surface elevation and meteorological conditions and its parameterization. J. Hydrol. 513, 154–163.
- Feiccabrino, J., Graff, W., Lundberg, A., Sandström, N., Gustafsson, D., 2015. Meteorological Knowledge Useful for the Improvement of Snow Rain Separation in Surface Based Models. Hydrology 2, 266–288. https://doi.org/10.3390/hydrology2040266
- Froidurot, S., Zin, I., Hingray, B., Gautheron, A., 2014. Sensitivity of Precipitation Phase over the Swiss Alps to Different Meteorological Variables. J. Hydrometeorol. 15, 685–696. https://doi.org/10.1175/JHM-D-13-073.1
- Harder, P., Pomeroy, J., 2013. Estimating precipitation phase using a psychrometric energy balance method. Hydrol. Process. 27, 1901–1914. https://doi.org/10.1002/hyp.9799
- Harpold, A.A., Kaplan, M., Klos, P.Z., Link, T., McNamara, J.P., Rajagopal, S., Schumer, R., Steele, C.M., 2017a. Rain or snow: hydrologic processes, observations, prediction, and research needs. Hydrol Earth Syst Sci 21, 1–22.
- Harpold, A.A., Crews, J.B., Rajagopal, S., Winchell, T., Schumer, R., 2017b. Relative Humidity Has Uneven Effects on Shifts From Snow to Rain Over the Western U.S. Geophys. Res. Lett. 44, 2017GL075046. https://doi.org/10.1002/2017GL075046
- Jennings, K.S., Winchell, T.S., Livneh, B., Molotch, N.P., 2018. Spatial variation of the rainsnow temperature threshold across the Northern Hemisphere. Nat. Commun. 9. https://doi.org/10.1038/s41467-018-03629-7
- Kienzle, S.W., 2008. A new temperature based method to separate rain and snow. Hydrol. Process. 22, 5067–5085. https://doi.org/10.1002/hyp.7131
- Marks, D., Winstral, A., Reba, M., Pomeroy, J., Kumar, M., 2013. An evaluation of methods for determining during-storm precipitation phase and the rain/snow transition elevation at the surface in a mountain basin. Adv. Water Resour. 55, 98–110. https://doi.org/10.1016/j.advwatres.2012.11.012
- Rajagopal, S., Harpold, A.A., 2016. Testing and Improving Temperature Thresholds for Snow and Rain Prediction in the Western United States. JAWRA J. Am. Water Resour. Assoc.
- Stewart, R.E., Thériault, J.M., Henson, W., 2015. On the Characteristics of and Processes Producing Winter Precipitation Types near 0°C. Bull. Am. Meteorol. Soc. 96, 623–639. https://doi.org/10.1175/BAMS-D-14-00032.1
- Wayand, N.E., Stimberis, J., Zagrodnik, J.P., Mass, C.F., Lundquist, J.D., 2016. Improving simulations of precipitation phase and snowpack at a site subject to cold air intrusions: Snoqualmie Pass, WA. J. Geophys. Res. Atmospheres 121, 9929–9942.
- Ye, H., Cohen, J., Rawlins, M., 2013. Discrimination of Solid from Liquid Precipitation over Northern Eurasia Using Surface Atmospheric Conditions\*. J. Hydrometeorol. 14, 1345– 1355.

---

## Author Comment (AC1) · 8 Oct 2018

Reviewer #1

The authors analyzed the range and distribution of the threshold temperature of rain and snow based on the CMA daily data, and developed a statistical model to discriminate rain and snow based on the relationship between threshold temperature and latitude, elevation, and annual precipitation. This study provides a practical method to separate precipitation types. However, the introduction needs to be adjusted, and the relationship between threshold temperature and other variables needs more consideration and discussion. R: Thanks for the comments. Section Introduction has been rewritten according to the suggestion and why this work does not consider other variables has been discussed.

[Figure]

Major Revisions:

1. The title of the paper is too broad. It is suggested to be more specific. R: The title has been changed already to: "A new method for determining the single threshold temperature of precipitation phase separation".

2. The number of key words is too many. Please revise it according to the requirement of the journal (usually not more than 5-6 key words). R: Done. Thanks.

3. L120-L185: (a) These five paragraphs are all about the precipitation type discrimination schemes. The L120-l134 part introduces some precipitation type discrimination methods based on surface variables (air temperature, pressure, dewpoint, relative humidity, and wet-bulb temperature), the following L135-155 part presents the methods based on surface air temperature and upper air temperature, the L156-L162 part introduces the methods based on vertical structure of atmosphere, and the L163-L180 part presents the surface variable based methods again (single threshold temperature and double threshold temperature), and finally analyzes the difficulty of humidity based method (which is also surface variable based method). These parts of content are about various kinds of method and mixed together. The structure of these paragraphs needs to be adjusted. R: Thanks a lot for the constructive suggestions. We have almost rewritten the paragraphs, and we hope that the revised introduction reads much better than before.

(b) L120-L185: This part has reviewed the various methods of separating different precipitation types. It is surprising that the recent work about precipitation type discrimination (Ding et al., 2014) has not been cited in the paper, which finds out that the precipitation type is dependent on the temperature, elevation, and relative humidity (i.e., the threshold temperature for rain and snow increases with the increasing of elevation, and the probability of sleet increases when relative humidity increases), and develops a dynamic threshold scheme to discriminate rain, sleet, and snow. Ding, B., K. Yang, J. Qin, L. Wang, Y. Chen, and X. He, 2014: The dependence of precipitation
types on surface elevation and meteorological conditions and its parameterization, J. Hydrol., 513, 154-163, doi:10.1016/j.jhydrol.2014.03.038. R: This paper has been cited in the revised version of the manuscript. This is indeed an important publication, and we have also made a citation of it in section Discussion of the new manuscript.

(c) L153-155: This sentence is a little abrupt here. It seems not closely related to the last sentence or paragraph, and also does not have any reference cited for the sentence. R: This sentence has been deleted.

4. Section 2 "Data and methods" could be divided into two sub-sections like "2.1 Data" and "2.2 Methods (or data processing)". Further, the flow diagram of the analysis needs to be explained more clearly. R: The section has been divided into two parts according to the suggestion. Another change in the section is the remove of the flow chart.

5. L335, L338, and L339: Actually, these three values are 90 percent instead of "95 percent", since the authors gave the percentages of snowfall, rainfall, and sleet between the 5 percent and 95 percent of the event profile, respectively. R: The three values are 95%. They are consistent with the contents of the related figures and tables. Thanks for the comments.

6. (a) In L349, the minimum daily mean temperature of rainfall at Zhaozhou station was 3.4âŮęC; however, it was -3.4âŮęC in Table 2. (b) The maximum daily mean temperature of snowfall at Balikun station was -5.1âŮęC in L352, but 5.1âŮęC in Table 2. (c) The minimum daily mean temperature of sleet at Shiquanhe station was -3.3âŮęC in L357, but -5.3âŮęC in Table 2. Please correct them, and also check the other numbers in the paper carefully. R: Many thanks for the careful reading of the manuscript. These problems should not escape from our eyes. We have corrected the errors, and checked again through the text.

7. Fig. 4d, if reversing the positive and negative, the spatial distribution of (Trn-Tsn) would be also similar to those of maximum daily mean temperature of snowfall (Fig. 4a), minimum daily mean temperature of rainfall (Fig. 4b), and the average daily mean

temperature of sleet (Fig. 4c). R: It seems similar to one another. Thanks for pointing this out.

8. Fig.5, about the positive deviation of the snowfall amount by the threshold temperature (Tt-d), is there any possible that because there is occasionally snowfall occurred at high temperature in these areas, which may pull up the threshold temperature by the few snowfall events? If the temperature range to settle the threshold is enlarged (larger than Tsn âĹij Trn), for example, ranking from (Trn-2)âŮęC to (Tsn+2)âŮęC, and settling the threshold temperature from this ranking profile, maybe this deviation could be reduced. R: It is possible that the occasionally occurred events in higher temperature environment cause uncertainty in some extent. However, to enlarge the temperature range will leads to other problems. Anyway this is a good question, and we would make some experiments in future.

9. (a) L452: the authors stated that "and it decreased from east to west in areas west of 90âŮęE", however, from Fig. 6, it seems the threshold temperature decreased from south to north in the areas west of 90âŮęE. (b) Actually, the L451-L452 sentence has the totally same meaning with the L453-L456 sentence, only with the different demarcations of longitude values (i.e., 90âŮęE and 105âŮęE, respectively). R: Yes, you are right. Actually it decreased from southeast to northwest. We have revised the description of the paragraph in the new version of the manuscript.

10. In Section 3.3, the authors stated that the threshold temperature (Tt-d) is related to longitude, latitude, altitude, annual precipitation, annual mean air temperature, and annual relative humidity. However, from R2 in Figure 7, altitude is most closely related to the threshold temperature (Tt-d), and then the relative humidity is. Although latitude and longitude showed some negative relationship with the threshold temperature, actually, from Fig. 6 "the distribution of the threshold temperature", the relationship between the threshold temperature with longitude and latitude is also related to that with altitude. Since in China, the mean altitude decreases from west to east and from south to north, as well as the decreasing trend of threshold temperature with decrease of altitude, that

is why threshold temperature decreases with the increase of the longitude and the increase of the latitude. R: We agree to the comments. It is possible that the relationship of the threshold temperature with longitude and latitude is also related to the variations of altitude and relative humidity in the study region. The altitude and relative humidity generally also decrease from west to east and from south to north, and the altitude and relative humidity have better correlations with the threshold temperature. These may be the reason why threshold temperature decreases with the increase of the longitude and latitude. If so, altitude and relative humidity may be the more important factors in determining the threshold temperature. We have added a paragraph to discuss about this. Thanks a lot.

11. Also in Section 3.3, I think another explanation is more reasonable for the negative relationship of threshold temperature with the relative humidity. In the arid area, where the relative humidity is lower, when the snow particle is falling from the same condensation height, since the relative humidity of atmosphere is lower, the difference of specific humidity between the particle surface and the air is larger, and therefore the heat absorbed by the snow particle is easier to be released in the form of latent heat flux. In that case, it needs more energy, i.e., higher surface air temperature, to melt the snow particles. That is why the threshold temperature is relative higher in the arid region (i.e., lower relative humidity region). R: This is an interesting alternative explanation. However, the further examination of the mechanism could be done in future studies. We added a sentence in the end of the paragraph to embody this possibility.

12. From the above Question 10, the threshold temperature has close relationship with altitude, but not latitude or longitude. Beside, from R2 in Fig. 7c, the relationship between the threshold temperature and the annual precipitation is also relative low. Therefore, the statistical model of threshold temperature derived from latitude, altitude, and annual precipitation (Equation 1) needs more consideration. Actually, from Equation 1, the impact from R (annual precipitation) on Tt-p (threshold) is relative low comparing with N (latitude) and H (altitude). And the impact of N could be also combined to H. R: We agree with you at this point. It should be good to directly use altitude and relative humidity as predictors. However, as we discussed above, in a larger scale, the relative humidity data are unavailable for use in studies, and we have to apply precipitation data instead if the method developed in this study is to be applicable for following works.

13. L575: What is MSRE short for? Do you mean RMSE? R: Yes. It is for RMSE. Corrected already.

14. L584-L585: from Figure 9, it seems the deviation is generally positive, no matter the low threshold temperature area or the high threshold temperature area. R: Yes. More areas have positive deviation. However, the deviation is generally larger in the low threshold temperature area, and smaller in high threshold temperature area.

Minor Revisions:

1. Please check the references. The reference "Stefan et al. (2008)" (L123) has not been included in the reference list. R: This was wrongly cited for Kienzle er al. (2008). Corrected in the revised manuscript.

2. L146: phrases => phases R: Corrected.

3. L193: was = > were R: changed already.

4. L211-L214: daily data of 839 national stations' air pressure, surface air temperature (daily mean, daily maximum and daily minimum), precipitation, evaporation, relative humidity, wind speed, sunshine hours, and 0-cm ground temperature => daily data of air pressure, surface air temperature (daily mean, daily maximum and daily minimum), precipitation, evaporation, relative humidity, wind speed, sunshine hours, and 0-cm ground temperature from 839 stations R: Thanks. Changed.

5. L228-L229: greater than or equal to 1 mm => no less than 1 mm R: Corrected already.

6. L233: sleet, snow => sleet, and snow R: Done.

7. L234: registered => recorded R: Corrected.

8. L247: isotherm of in 3âŮęC => isotherm of 3âŮęC R: Corrected.

9. L336: less => lower R: Corrected.

10. L361: than instantaneous air temperature => than the instantaneous air temperature R: Corrected.

11. Fig 4: (a) the symbol "Tsm" and "Trn" is suggested to be added in the figure caption and in the figures to be more clearer; R: We have added it to the caption and in the figures, according to the suggestion.

(b) the caption for Figure 4d could be revised as "difference between the maximum daily mean temperature on snowfall day and the minimum daily mean temperature on rainfall day" or "difference between Tsm and Trn"; R: Changed already.

(c) L397: occur => occurs R: Corrected.

12. L384: Trm-Trn => Tsm-Trn R: corrected.

13. There are several different spellings as "Qinghai-Xizang Plateau" (L557), "Qinghai Tibet Plateau" (L557-L558), and "Qinghai-Tibetan Plateau" (L539). They need to be unified. R: These have been checked, and changed to "Qinghai-Tibetan Plateau".

14. L660 (two places): phrases => phases R: Corrected.

Please also note the supplement to this comment:
https://www.hydrol-earth-syst-sci-discuss.net/hess-2018-307/hess-2018-307-AC1-supplement.pdf

---

## Author Comment (AC2) · 8 Oct 2018

Reviewer 2

Overview The authors present a detailed analysis of rain-snow partitioning over northern China using meteorological measurements and precipitation phase observations from several hundred research stations. They show marked spatial variability in the rain-snow air temperature threshold, which is generally highest near the Tibetan Plateau and lowest in the northeast part of the country. They found that the threshold was correlated with latitude, longitude, relative humidity, and precipitation across their study domain. The authors then used those variables, minus longitude and relative humidity, in a stepwise multiple linear regression to predict the rain-snow air temperature threshold. The regression performed well relative to observations. R: Thanks for the

brief review and the following comments.

Major issues

Overall, this work adequately presents spatial variation in rain-snow partitioning using a robust meteorological dataset. However, I would not accept this paper in its current form. First, the paper is titled "A new method to separate precipitation phases." My contention with this is that the method is a multiple linear regression that cannot be transferred to other regions. The authors use latitude as a predictor variable instead of other meaningful physical quantities that might be transferrable in space. Because of this shortcoming, their method could not be used in other geographic regions (Europe, North America), thus limiting its utility. Froidurot et al. (2014) presents regression methods that use more meaningful independent variables. R: Thanks for the comments. In our analysis, latitude and longitude are indeed not so good as altitude and relative humidity in indicating the threshold temperature. Latitude is actually representing temperature in a large extent, however, and the threshold temperature would be dependent on thermo-environment or temperature field. It should be a better predictor than say longitude. We rechecked the relationship of the threshold temperature with climatic variables, and we found that annual precipitation days were not as good as annual total precipitation amount. However, annual mean (winter mean) temperature was almost as good as latitude. Thus, we did modify the model to use annual mean temperature rather than latitude, and also revised the section of the manuscript. We believed that the model is robust for use in separating snow and rain on a large scale.

Additionally, there is not much novelty in the work the authors present. For one, spatial variability in rain-snow partitioning has already been described at local (Wayand et al., 2016), regional (Rajagopal and Harpold, 2016), continental (Ye et al., 2013), hemispherical (Jennings et al., 2018), and global (Dai, 2008) scales. In this context, the finding that rain-snow air temperature thresholds vary over large distances is unsurprising. Secondly, the authors relate this variability to relative humidity and altitude, which has been covered in depth by previous authors. I feel that this work may be

considered more of a case study than a significant contribution to hydrologic science. There is novel research that can be done with the datasets the authors have at their disposal. For example, figure 3 presents interesting differences across the regions in rain-snow partitioning and mixed-phase events. However, most of the work presented is not currently suited to the high standards of HESS. I would therefore either recommend major revisions or rejection depending on the opinions of the Associate Editor and other reviewers. R: We presented a novel single temperature threshold method to objectively partition the snow and rain of precipitation data, based on the latest and highest observational dataset, in a subcontinent characterized by the most complicated topography and climatic condition in the world. We believed that the method had potential to be applicable in other regions outside of mainland China, especially in studies of long-term change in snowfall and extreme snow events in mid- to high latitude.

Throughout

Given HESS's large international audience, I would recommend working with a translating and copyediting service to clean up the English for clarity. I have noted in my specific comments below certain sentences and paragraphs that need particular attention, but the writing needs improvement before resubmission, if resubmission is suggested by the Associate Editor. R: Thanks for the suggestion. We have invited a copyediting service company to polish the English.

Specific comments

Line 1: I would change the title as multiple linear regression is not a new method and the regression only applies to northern China (i.e., it cannot be used in other geographic regions). R: Thanks. The title has been changed to "A new method for determining the single threshold temperature of precipitation phase separation". The study area almost includes the whole mainland China with snowfall each year, because south of the Yangtze river usually experiences little or no snowfall.

Lines 45–48: The motivation could be clearer (i.e., change from snow to rain). R: The

introduction has been rewritten, and the motivation of the study has been made clearer than the original version of the manuscript.

Lines 48–52: Break into two sentences. Clearly define the rain-snow temperature threshold. Additionally, the study mentions other types of temperature (dew point and wet bulb), so always note when it is air temperature. R: Revised already. Dew point and wet bulb temperature data have not been used in the analysis, and the temperature in the text is always the surface air temperature if not mentioned.

Line 49: Based on the rest of the article, I'm assuming that weather phenomenon records are visual observations of precipitation phase as in Ding et al. (2014) and Dai (2008). This line should be changed to reflect this (along with all other mentions of "weather phenomenon"). R: Yes. "weather phenomenon records" has been changed to "visual observations of precipitation phase" throughout the text.

Lines 52–53: The first and second clauses are describing the same thing. R: changed already.

Line 56: Although the en dash is correct here, it might be mistaken for a negative symbol. Perhaps write out the ranges (i.e., -1.2°C to 6.3°C). R: Corrected, and the ranges given also.

Line 57: Do the temperature data go to two significant figures after the decimal point (2.81)? If not, please correct this and all other instances. R: All temperature values have been changed to one significant figure after the decimal point.

Lines 57–58: What is the "actual threshold?" Use consistent terminology throughout. R: Corrected, and "actual" deleted.

Line 59: Low or lowest? R: Thanks. It is lower relatively rather than the lowest.

Line 62: Do you mean more variable instead of dispersed? R: We mean more dispersed, rather than more variable. Changed already.

Line 64: Remove semi-colon and split into two sentences. R: Changed.

Lines 66–69: Annual precipitation does not control precipitation phase partitioning. I would assume this was likely an effect of relative humidity based on the other data presented in the work and the research done by other authors. R: Thanks for the comment. Indeed the relative humidity has a better correlation with the threshold temperature. However, relative humidity data are generally unavailable in most regions outside of China, and thus we chose to use annual total precipitation which has a good positive correlation with relative humidity, and also is more easily available anywhere. We also built a model with relative humidity as predictor, and added the formula for reference in section Discussion.

Lines 74–76: This seems like an important finding, but I'm unsure of what it is. What is a relative deviation of snowfall days? Is that the number of days misidentified as rain when snow was actually occurring? Please use more precise language to clearly convey the findings. R: Thanks for the comments. We have changed the term of relative deviation to relative error, which is defined as percent ratio of absolute error of modeling value to observational (true) value.

Lines 76–78: You must note that this is only for northern China as the regression would not apply to other areas. R: The study area almost includes the whole mainland China with snowfall each year, because south of the Yangtze river usually experiences little or no snowfall. This, together with the fact that mainland China is characterized by the most complicated physical geographical and climatic conditions in the world, may make the model developed in this study potentially applicable in other regions. We made a further discussion about the applicability of the method in section Discussion.

Lines 91–94: I'm not sure what this means. Blizzards? Heavy rain events? Freezing rain? Please clarify. R: Modified as "…as more than 50

Line 95: What is a precipitation condition? Depth? R: Yes, precipitation amount or depth. Changed to "Under the similar precipitation amount…"

Lines 102–109: What do these studies actually say? Be specific. R: We have revised this paragraph. A sentence has been added to explain the main findings of the studies related to climate change.

Lines 112–116: Break up into two sentences and rewrite for clarity (i.e., there are few direct observations of precipitation phase at the global scale). Also, researchers can partition precipitation phase, but there are difficulties in doing so at air temperatures near freezing (Ding et al., 2014; Jennings et al., 2018; Stewart et al., 2015). R: Done. The publications recommended have been cited in the revised manuscript.

Lines 120–134: Please give more information on the studies you cite. The readers need to know the relevant conclusions of the papers, not just what was studied. For example, Harder and Pomeroy (2013) showed the psychrometric energy balance was more effective at predicting precipitation phase than air temperature alone. R: Thanks for the suggestion. We have revised the section according to the comment, and the unrelated citations have been deleted in the new version of the manuscript.

Line 123: Stefan et al. (2008) should be Kienzle (2008). Please double-check all citations to match HESS's style, which is to give the author's last name (I know this can be tricky as in most western countries the family name comes last, which is not the case in China). R: Double-check has been done, and the similar problems have been solved.

Lines 132–134: The Jennings et al. (2018) paper showed that including humidity improved the predictive capacity of precipitation phase methods over air temperature alone. This was also shown by other authors (e.g., Ding et al., 2014; Harder and Pomeroy, 2013; Marks et al., 2013). R: The phrasing has been changed according to the comment.

Lines 135–137: International readers will need more info on this event. R: The paragraph has been deleted, and the similar contents show more information on studies of snowfall in China.

Lines 135–155: This information would be best combined with the previous paragraph in a discussion of how precipitation phase can be best predicted. There are air temperature methods and then those that use other meteorological and physiographic quantities. The conclusions presented in lines 139–147 are much more specific than any other introductory material and seem out of place. R: We modified the contents of this part. The more specific information was deleted.

Line 148: What is a discriminant index? R: This phrase has been changed to "discriminant condition".

Line 150: Dew temperature should be dew point temperature. Correct this throughout paper. R: Corrected already. This section has been deleted. Thanks.

Lines 156–162: You could add a separate paragraph about precipitation phase methods that use atmospheric (i.e., not just surface) quantities to predict rain and snow. Much of this is covered in detail in two review papers (Feiccabrino et al., 2015; Harpold et al., 2017a). R: In revision, we have deleted most of the unrelated citations, but explained why we had to use temperature data to make investigation in this work.

Lines 163–164: Be specific on how rain and snow have differing effects on the land surface (e.g., rapid runoff from rainfall versus winter storage and spring release for snowfall). R: This has been simplified in the revision.

Line 164: "Therefore" is not correct here. R: It has been deleted.

Lines 164–180: As with my previous comments, this section should be shortened and cleaned up. Here the authors list different ways of partitioning precipitation phase, but the paragraph is introduced as if it is providing different information. Double and single thresholds are covered in depth in the aforementioned review åŇŰpapers (Feiccabrino et al., 2015; Harpold et al., 2017a). R: These have been simplified and shortened largely in the revision, and the section has been restructured.

Line 177: What is an auxiliary indicator? R: This sentence has been deleted.

Line 179: Many gridded climate datasets provide humidity information that can be used to estimate dew point temperature with reasonable accuracy. R: Thanks for the information. Compared temperature and precipitation, relative humidity data or other humidity data are very difficult to obtain, especially for continents and the global land. The data quality of relative humidity is also problematic. However, we built a model with RH as predictor and included it in section Discussion. By the way, we do not think that the reanalysis data of atmospheric humidity could be used in this kind of researches.

Line 180: There are several mentions of larger-scale analyses and how current precipitation phase methods struggle over broad spatial extents. While this is correct, the method the authors provide in this paper is specific to northern China and can also not be applied to large scales outside of the country. R: The study area almost includes the whole mainland China with snowfall each year, because south of the Yangtze river usually experiences little or no snowfall. This, together with the fact that mainland China is characterized by the most complicated physical geographical and climatic conditions in the world, may make the model developed in this study potentially applicable in other regions. Of course, we have to do more in future to reach the goal, including the verification of the model in the neighboring regions and selection of more suitable predictors in a larger region.

Lines 181–185: This paragraph provides redundant information. It should be edited and combined with previous information on phase partitioning methods. R: Thanks. This paragraph has been rewritten in the revision.

Lines 181–182: Again, many authors have shown that humidity improves phase prediction over air temperature only methods. R: Yes. Our work also shows that. As we discussed before, relative humidity data are not easy to be obtained for other regions, and this would prevent the model from being applied in regions outside of China. Please see the replies above.

Line 182: Arpold should be Harpold and Keith should be changed to Jennings. Please

doublecheck all citations. R: Thanks for pointing them out. These have been corrected.

Lines 186–189: Yes, China does have diverse climatic and physiographic characteristics. No, the method the authors introduce cannot be applied to other areas in the world because it uses latitude, which is not physically meaningful in the context of precipitation phase. R: Please see the discussion above. Latitude is generally related to temperature, and in most areas also to precipitation, due to the latitudinal distribution of solar radiation and heat on the earth surface. Our analysis also finds that latitude and the threshold temperature had a significant correlation in China. However, we have already changed it to temperature as a predictor in building model referring to this comment, as mentioned before. The results are almost the same.

Line 191: What type of observational data? Be specific. R: Thanks. Done already.

Lines 193–196: Again, the method can only be applied to northern China where the regression was developed. R: May be so, and may be not so. We will make more verification in future. Thanks.

Lines 202–204: Many spatially extensive gridded climate and reanalysis products include surface pressure and humidity information. R: This is a good idea. It is interesting to apply reanalysis data in future work. However, a problem is that reanalysis data are relatively poor in estimating long-term trends of precipitation and extreme precipitation events.

Lines 207–209: Data availability should be provided in a section at the end of the manuscript. R: OK. We will make it clear once the manuscript is accepted for publication in HESS.

Lines 208–209: As noted above, weather phenomenon should be changed to precipitation phase observations if that is what is included in the dataset. It should be clear what the dataset contains. R: The term has been changed as suggested.

Lines 211–214: For the quantities besides air temperature, please note whether they

are daily averages or totals. R: Thanks. Clarified. Temperature is daily maximum, minimum and mean temperature, precipitation is the accumulated amount of 24 hour from 0800 to 0800 Beijing Time, and all others are daily means.

Lines 216–218: Were the stations removed or the data removed? It is unclear. R: The stations were removed from the dataset.

Lines 218–219: Why and how were the latitude and longitude corrected? R: A few of errors exist in the two datasets. The sentences have been deleted in the revision because it is unnecessary to mention here.

Lines 219–220: Are the meteorological stations in the same location as the precipitation phase observations? R: Yes. They are at the same stations.

Lines 229–232. Remove and combine the first part of the paragraph with the next paragraph (lines 233–239). R: Thanks. Simplified, and combined with the next paragraph.

Lines 233–239: Sleet is not technically the same thing as a rain-snow mix. I.e., a rain-snow mix could be sleet but there could also be rain and snow in a day without sleet occurring. Change this terminology to mixed-phase events to be more accurate. R: Thanks for the suggestion. Accepted, and all "sleets" changed to "mixed-phase events" throughout the text.

Lines 237–239: What is the reasoning behind this? R: Mixed-phase events occur in a day when there might be rain or snow, or sleets during different hours. The threshold temperature must be lower than the daily maximum temperature and higher than the daily minimum temperature of the mixed-phase events.

Lines 240–251: This paragraph is confusing. Are there 324 stations in the analysis or 623 as previously mentioned? I feel like the authors are trying to stay that only stations with a minimum of 100 snowfall days were analyzed. R: We analyzed only 324 stations with snowfall in the study area. All these stations had at least 100 days of snowfall during 1981-2010 to guarantee the samples and the representativeness of

the statistics.

Line 241: Random is not correct here. R: It has been changed to "arbitrary cases of snowfall".

Line 252: What are extreme rain and snow records? R: It indicates extremely or abnormally large values of rainfall and snowfall at the stations.

Lines 255–257: This is very confusing. Are you saying the mean temperature during sleet was considered to be the rain-snow air temperature threshold? R: It was taken as only a reference. The threshold temperature value must be between daily maximum and minimum temperature of the mixed-phase events.

Lines 261–270: The choice of representative stations seems arbitrary unless I am missing something. Why not provide summary statistics for the stations in each region? Or, at least provide reasoning for representative station selection. Additionally, the Monsoon Region (I) is compared to the others throughout the paper despite the fact that it has an order of magnitude more stations than II and III. This needs to be addressed. Finally, throughout the paper, please note in which geographic region each representative station is located (i.e., Zhaozhou becomes Zhaozhou-I). R: The reasoning for selecting the three representative stations has been given. Table 1 gives the information of the three stations. We have also made an explanation of the uneven distribution of the stations. The sparse observations in the Qinghai-Tibetan Plateau will affect the analysis in a large extent.

Lines 272–279: The station colors should be included as a legend. R: Revised already.

Lines 282–285: I might be misreading this but the "percent deviation of snow days" seems like a poor way of quantifying method success. For example, let's say Station X has 50 snow days and 50 rain days in a year. I could still get a 0R: Relative deviation is not designed to indicate success rate. It shows a relative bias of the prediction from the true value or observation. In the case that you mentioned, the relative deviation will

be -100

Lines 286–297: This can be shortened significantly. Stepwise regression is not a new/novel technique. R: Thanks for the suggestion. We have largely simplified the description of the method in the revised manuscript.

Lines 300–301: What is the snow day mean temperature method? This methods section should be rewritten to provide much more clarity. It should be obvious how the rain-snow air temperature threshold is calculated. R: "snow day mean temperature method" should be termed as "mean temperature method for snow day". This section has been written in the revision, and the term has been changed to Snowday-Direct-Definition-Mothod (SDDM). Thanks.

Line 312: Figure 2 confused me more than it helped me. This information, when clearly and logically presented, should be easy to understand by reading the methods section. R: This figure has been removed in our revision.

Lines 324–326: Remove. R: Thanks. The sentence has been deleted.

Lines 343–346 (Fig. 3): These plots are great (they present a lot of useful information), but could be improved slightly. First, I would flip the axes as it is generally customary to have the cumulative distribution on the y-axis and the measured variable on the x-axis (air temperature in this case). Second, I would normalize the precipitation events so the scale goes from 0-1 (fractional) or 0-100 (percentage). Third, you only need one legend for the whole figure, not one for each subplot. R: Thanks for the comments. We revised the figure according to the suggestion, as you can see in the revised manuscript.

Lines 350–351: What is the threshold (same comment applies to line 358)? R: It is the calculated threshold temperature, and we added the values for Zhaozhou and Shiquanhe stations.

Lines 359–370: This paragraph is mostly discussion material and should be moved. Citations should be provided that support the suggestions made by the authors. R:

This paragraph has been deleted in the new version of the manuscript.

Line 372: This table should be flipped so that the stations are listed in the first column and the metrics are in the following columns. I provide an unformatted example below:

R: Already revised according to the suggestion.

Lines 374–377: This may be the case, but how much precipitation is falling at these extreme values? R: Thanks for the question. It is unclear to address this issue. Further analysis is needed to answer the question based on the method developed in this work.

Lines 378–384: This reads like a figure caption and should be removed or combined with the Fig. 4 caption. R: It has been simplified already.

Lines 385–386: Use the abbreviation once it is introduced (e.g., Tsm, Trn, etc.). R: Thanks. Already changed.

Line 395 (Fig. 4): How were the spatial interpolations performed from the point data? The same color ramp should be used for each figure in this case and only one legend is needed. Also, there is no need to include the parts of China that were not analyzed (i.e., limit the plot to what is shown in the red box). R: Accepted. Ordinary Kriging method was used to interpolate the point data. The figures have been redrawn according to the suggestion.

Lines 403–416: These are methods and should be moved. Additionally, I do not agree that this is the way the rain-snow threshold should be calculated as it seems unnecessarily confusing and arbitrary. The Dai (2008) method would be a preferable easy-to-understand and well-validated way of calculating the threshold. R: These have been moved to section Data and Methods. This method is good, if not better than that of Dai (2008), and it is a novel and simple procedure.

Lines 417–441: I would remove these paragraphs and figures for three reasons: 1) A different method for calculating the rain-snow threshold should be used (see my comment above); 2) Relative deviation is not the best method for calculating method error

(see my comment on lines 282–285); and 3) The findings are not central to the authors' main story. R: Thanks for the comments. However, we would keep as it was. The reasons are: 1) this method is novel, simple and verifiable; 2) relative deviation, or relative error (as we have revised in the new manuscript), is a good indicator of the method performance. Combined with other indicators, it is able to tell the effectiveness of calculation; 3) the findings are not central, but they are the basis to verify the effectiveness and applicability of the model developed below.

Line 452: West of 90°E, the threshold decreased from south to north, not from east to west. R: Yes, thanks. We have modified the sentences: "The threshold temperature west of 100°E showed an approximately zonal distribution, and the threshold temperature decreased with the increase of latitude; the east of 100°E had a meridional distribution, and the threshold temperature decreased with increasing longitude."

Lines 458–460: Please note how the spatial interpolations were calculated. There are large spatial extents with no station data. It may be misleading to present the threshold information in this way. R: Thanks for the comments. This will be somehow more obvious in the Qinghai-Tibetan Plateau and northwestern desert areas. We have made a brief discussion about this in the end of the same paragraph.

Lines 461–465: Remove. The threshold was calculated from the observations, so of course it should reflect them. R: Accepted. This paragraph has been removed.

Lines 467–472: Remove. This is introductory material and the regression cannot be used outside China as it uses latitude as a predictor variable. R: Thanks. This has been removed.

Lines 473–477: Combine with following paragraph and remove redundancies. R: We have largely simplified the sentences according to the suggestion.

Line 480: Change disperse to variable. R: Done. Thanks.

Line 482: Change centralized to less variable. R: Changed.

Lines 490–492 (Fig. 7): Add legend for figure colors. Latitude needs a degree symbol. Did you check how the regression in c was affected by the extreme precipitation outlier? Why was longitude not plotted? This was given as one of the significant correlates of the threshold temperature. R: The figure already modified referring to the suggestion. The extreme precipitation outlier would have an effect, but the effect is small. The new figure no longer contains both latitude and longitude, because latitude has not been used in building model.

Lines 495–515: Remove. This is discussion material. Citations are needed if this material is kept. R: This explanation would be somehow useful to readers. However, we endure pains to have deleted it, referring to the suggestion by the reviewer.

Line 517: Again, I do not agree that humidity is difficult to obtain. Yes, it is less common than air temperature and precipitation, but it is available from many meteorological stations and gridded climate products. Additionally, figure 7 shows a very weak relationship between precipitation and the threshold, especially considering the extreme outlier I noted above. R: Observed relative humidity data are less available in most regions outside of China. Reanalysis data are more easily available, but they have some intrinsic problems. One problem would be the bad usefulness in showing long-term trend over the last decades. The method we are to develop in this work was intended to be used in separating rain and snow in a large scale in forming a long-term daily snowfall dataset. This is the main reason why we did not choose to use reanalysis data in developing the method.

Line 520: As I have mentioned throughout, the use of latitude is a large limitation of this method as it means it can only be applied to northern China. R: Thanks for the comments once again. Please see the discussion above. Latitude is generally related to temperature, and in most areas also to precipitation, due to the latitudinal distribution of solar radiation and heat on the earth surface. Our analysis also found that latitude and the threshold temperature had a significant correlation in China. However, we have revised the model by including annual mean temperature rather than latitude as

a predictor.

Lines 526–527: The coefficient of variation (r2), root mean squared error, and mean bias would all be more appropriate error metrics to provide. Additionally, were any of the stations removed from the data when computing the regression? Error statistics cannot be calculated reliably if all stations were used. Ideally, the model output should be cross-validated on stations that were removed before calculating the regression coefficients. R: A good suggestion. Thanks a lot. We have added a table and the corresponding explanations in this section of the new manuscript, with relative error being given more attention but also other metrics mentioned.

Lines 534–535: This is confusingly written. Please rewrite for clarity. R: Thanks. The sentence has been changed.

Lines 541–546 (Fig. 8): What is the x-axis on these figures? The region lines should not be the same color as the data. R: Modified. X-axis is stations of the three regions.

Lines 547–549: Again, please use more robust error metrics. In this case, the mean bias would be helpful as it would indicate whether snowfall was being over or under-predicted. (It looks like this is shown in the following lines and Fig. 9, so please make this information more clear to the reader.) R: We used more error metrics to indicate the error in the revised manuscript. Thanks for the suggestion. The mean absolute errors of threshold temperature of the simulated snowfall are 0.476°C for region one, 0.560°C for region 2, and 0.435°C for region 3, for example. We have also added a figure to show the spatial distributions of the absolute errors, and a table to show the mean errors of other metrics for the whole study area and the three sub-regions.

Lines 565–568 (Fig. 9): If this is the mean bias of snowfall days and snowfall, please make that clear. R: This is for relative errors, rather than for absolute errors. However, we have already added analysis of absolute errors, including a new figure showing the distributions of the absolute errors in the study area.

Lines 569–574: I do not understand what this paragraph is trying to say and how it relates to the previous information. R: We have revised the paragraph to include the illustration of other errors except for relative errors. It is intended to explain the reason why the relative errors in the study area are so larger as compared to other regions at the same latitudes of the world.

Lines 575–586: Rewrite for clarity. I think the main point of this paragraph is that the method underpredicts snow in high threshold areas and overpredicts in low threshold areas, but I am not certain. R: We have rewritten the paragraph. The revised paragraph includes more illustrations of the absolute errors, relative errors, correlation coefficients, and RMSE of the calculation method, so that readers could get a more systematic picture of the performance of the model.

Line 594–641: There are many, many methods for predicting precipitation phase. I am unclear as to why the authors devote nearly 50 lines for comparing the method to one other (there is too much space given here to the Han method). If the authors wish to include this information in their results, they should include other methods and provide a robust comparison. As it stands, I would remove this section or shorten it and add to discussion along with other phase method comparison papers. Particularly relevant is Ding et al. (2014), who also used China Meteorological Administration data in their work. R: Thanks for the comments. This subsection has been moved to section Discussion, and we also added the comparison with that of Ding et al. (2014).

Lines 654–691: This is more of a summary of the paper than a discussion and should be rewritten. There were parts of the results section that should be moved here (noted in my previous comments). Additionally, the discussion should clearly note major limitations and assumptions, which this does not except for the last paragraph. Finally, the authors should compare their work to that of other researchers. There is a lot of literature on the subject of rainsnow partitioning, none of which is discussed here. R: According to the suggestion, this section has been enhanced largely, and the comparisons with other methods have been moved here.

Lines 708–710: Remove. R: Removed already.

Lines 711–713: To reiterate, latitude is not a physically meaningful quantity in terms of precipitation phase partitioning. Figure 6 shows that at 30°N the threshold temperature decreases from 5°C in the west to 0°C in the east. This pattern appears fairly consistently in the data as one moves northward with the only exception being northwest China, where the threshold is low. The authors are likely seeing the effect of altitude and humidity, which is, in some cases, cross-correlated with latitude. The use of latitude throughout this paper is a major weakness that must be addressed. R: Thanks for the comments. As discussed before, latitude is representing temperature in a large extent, and the threshold temperature would be dependent on thermo-environment or temperature field. However, we found the correlations of the threshold temperature with annual mean (winter mean) temperature was as good as latitude. Thus, we have modified the model, and also revised the section of the manuscript.

Lines 713–714: Rewrite to remove redundancies. R: This has been revised.

Line 715: Change good to statistically significant. R: Changed already.

Lines 716–717: Change specially to especially. Additionally, much previous work has shown how relative humidity improves phase partitioning. This is another weakness of the paper that the authors show a strong relationship between relative humidity and the rain-snow temperature threshold, but do not include in their regression (I have noted in previous comments that although relative humidity is less available than air temperature and precipitation, it can still be widely found in ground observations and gridded climate products). R: Thanks. We discussed about this before, and would you please refer to the above replies to the interesting question and comments.

Review references Dai, A., 2008. Temperature and pressure dependence of the rain-snow phase transition over land and ocean. Geophys. Res. Lett. 35. Ding, B., Yang, K., Qin, J., Wang, L., Chen, Y., He, X., 2014. The dependence of precipitation types on surface elevation and meteorological conditions and its parameterization.

J. Hydrol. 513, 154–163. Feiccabrino, J., Graff, W., Lundberg, A., Sandström, N., Gustafsson, D., 2015. Meteorological Knowledge Useful for the Improvement of Snow Rain Separation in Surface Based Models. Hydrology 2, 266–288. https://doi.org/10.3390/hydrology2040266 Froidurot, S., Zin, I., Hingray, B., Gautheron, A., 2014. Sensitivity of Precipitation Phase over the Swiss Alps to Different Meteorological Variables. J. Hydrometeorol. 15, 685–696. https://doi.org/10.1175/JHM-D-13-073.1 Harder, P., Pomeroy, J., 2013. Estimating precipitation phase using a psychrometric energy balance method. Hydrol. Process. 27, 1901–1914. https://doi.org/10.1002/hyp.9799 Harpold, A.A., Kaplan, M., Klos, P.Z., Link, T., McNamara, J.P., Rajagopal, S., Schumer, R., Steele, C.M., 2017a. Rain or snow: hydrologic processes, observations, prediction, and research needs. Hydrol Earth Syst Sci 21, 1–22. Harpold, A.A., Crews, J.B., Rajagopal, S., Winchell, T., Schumer, R., 2017b. Relative Humidity Has Uneven Effects on Shifts From Snow to Rain Over the Western U.S. Geophys. Res. Lett. 44, 2017GL075046. https://doi.org/ 10.1002/2017GL075046 Jennings, K.S., Winchell, T.S., Livneh, B., Molotch, N.P., 2018. Spatial variation of the rainsnow temperature threshold across the Northern Hemisphere. Nat. Commun. 9. https://doi.org/10.1038/s41467-018-03629-7 Kienzle, S.W., 2008. A new temperature based method to separate rain and snow. Hydrol. Process. 22, 5067–5085. https://doi.org/10.1002/hyp.7131 Marks, D., Winstral, A., Reba, M., Pomeroy, J., Kumar, M., 2013. An evaluation of methods for determining during-storm precipitation phase and the rain/snow transition elevation at the surface in a mountain basin. Adv. Water Resour. 55, 98–110. https://doi.org/10.1016/j.advwatres.2012.11.012 Rajagopal, S., Harpold, A.A., 2016. Testing and Improving Temperature Thresholds for Snow and Rain Prediction in the Western United States. JAWRA J. Am. Water Resour. Assoc. Stewart, R.E., Thériault, J.M., Henson, W., 2015. On the Characteristics of and Processes Producing Winter Precipitation Types near 0°C. Bull. Am. Meteorol. Soc. 96, 623–639. https://doi.org/10.1175/BAMS-D-14-00032.1 Wayand, N.E., Stimberis, J., Zagrodnik, J.P., Mass, C.F., Lundquist, J.D., 2016. Improving simulations of precipitation phase and snowpack at a site subject to cold air intrusions: Snoqualmie Pass,

WA. J. Geophys. Res. Atmospheres 121, 9929–9942. Ye, H., Cohen, J., Rawlins, M., 2013. Discrimination of Solid from Liquid Precipitation over Northern Eurasia Using Surface Atmospheric Conditions*. J. Hydrometeorol. 14, 1345–1355.

Please also note the supplement to this comment:
https://www.hydrol-earth-syst-sci-discuss.net/hess-2018-307/hess-2018-307-AC2-supplement.pdf

---

## Author Comment (AC3) · 8 Oct 2018

Dear colleague, This is the revised manuscript with the changes kept. I will also submit the revised one with all traces cleaned. Thanks. Guoyu Ren

Please also note the supplement to this comment:
https://www.hydrol-earth-syst-sci-discuss.net/hess-2018-307/hess-2018-307-AC3-supplement.pdf

---

## Author Comment (AC4) · 8 Oct 2018

**A new method for determining the single threshold temperature of precipitation phase separation**

Yulian Liu[a,b]   Guoyu Ren[a,c]   Xiubao Sun[a,c]   Xiufen Li[d,e]   Hengyuan Kang[f,]

[a] Department of Atmospheric Science, School of Environmental Science, China University of Geosciences, Wuhan 430074, China;
[b] Heilongjiang Climate Center, Harbin 150030,China;
[c] Laboratory for Climate Studies, National Climate Center, CMA, Beijing100081,China;
[d] Innovation and Opening laboratory of Regional Eco-Meteorology in Northeast, China Meteorological Administration
[e] Heilongjiang Provincial Institute of Meteorological Sciences, Harbin 150030,China.
[f] Harbin Meteorological Bureau, Harbin 150028,China.

Corresponding author: G. Ren (guoyoo@cma.gov.cn)

Submitted to *hydrology and earth system sciences* for possible publication

**Abstract**

Separating the solid precipitation from liquid precipitation in existing historical precipitation observation data is a key problem in the monitoring and study of climate anomaly and long-term change of extreme precipitation events in difference phases. Based on the comprehensive analysis of the historical daily air temperature, precipitation data, and visual observations of precipitation phase in the northern areas of Mainland China (north of 30°N), this paper proposes a Snowday-Direct-Definition-Mothod (SDDM) to determine the threshold temperature of rainfall and snowfall for a complex and diverse geographical and climatic region. A statistical model of separating solid from liquid precipitation was established. The main conclusions include: (1) in northern China, the threshold temperature range of the daily mean temperature of rain and snow determined based on weather phenomenon records was from -1.2°C to 6.3°C, with a difference of 7.5°C among areas, and a mean threshold value of 2.8°C for the whole region. The temperature in the northern Tibetan Plateau was the highest (generally higher than 4°C). The low threshold temperature values appeared in eastern Northeast China, North China, and northern Xinjiang Autonomous Region, which were less than 2°C. (2) The threshold temperature decreased with increase in longitude east of 100°E, but it was more dispersed in the areas west of 100°E. The threshold temperature was generally higher and more variable in the low-latitude, and lower and more concentrated in the high-latitude. The threshold temperature generally increased with altitude. (3) There was a negative correlation between the threshold temperature and the annual precipitation. It was also negatively correlated with the annual average relative humidity. (4) The multivariate regression fitting model developed based on the annual mean temperature, altitude, and annual precipitation was able to simulate the threshold temperature of the precipitation phase in northern China well. The calculated threshold temperature based on the model has a smaller relative error for snow days and snowfall, and the stations with less than 10% of relative error reached

97% and 92%, respectively. The results of this study can therefore be applied for the separation of solid and liquid precipitation events in the areas without sufficient weather phenomenon records.

**Key words:** Northern China; Precipitation; Phase; Separation; Statistical model;

Regional differences

**1. Introduction**

Precipitation is an important parameter used to characterize climate characteristics and climate change, and it is one of the key components of the Earth's water and energy cycles (Loth et al., 1993). The influence of different phases of precipitation on the surface water and energy cycles is enormous (Vavrus, 2007; Wu et al., 2009), as more than 50% of the global meteorological disasters are closely related to abnormal precipitation, including extreme intense rainfall, heavy snowfall or blizzards, freezing rain and droughts (WMO, 2013; Wang et al., 2005). Under the similar precipitation amount, the effect of different phases of precipitation on the Earth's surface system and the social and economic system is clearly different, thus it is important to distinguish and understand the characteristics and anomalies of snowfall or mixed-phase events and their causes. In addition, in monitoring long-term changes in extreme precipitation events on sub-continental to global scales, it is also necessary to distinguish rainfall and snowfall events from historical precipitation data.

To date, many studies have been published on the characteristics and multi-decadal variation of snowfall in China (e.g. Jiang et al., 2003; Yang et al., 2005; Qin et al., 2006; Liu et al., 2012, 2013; Zhang et al., 2015). Also, many studies on both the global and Asian regional total precipitation and extreme precipitation events and their long-term change have been reported (Becker et al., 2012; Noake et al., 2012; Polson et al., 2013; Blanchet et al., 2009; O'Hara et al., 2009; Kunkel et al., 2009; Ren, 2007, 2015, 2016; Liu et al., 2011; Fang et al., 2011; Zhong et al., 2013; Wan et al., 2013; Yu et al., 2014; Xiao et al., 2015; Dang et al., 2015). The analyses of precipitation change generally showed a detectable trend toward more precipitation amount and more frequent extreme rainfall events over the last decades. All of these studies have greatly enriched the understanding of global precipitation and snowfall climatology and the climate change and variability in different regions and varied scales.

There are few direct observations of precipitation phase at the global scale.

Researchers can partition precipitation phase, but there are difficulties in doing so at air temperatures near freezing (Ding et al., 2014; Jennings et al., 2018; Stewart et al.,

2015). Even in the case of relatively abundant meteorological observational data in

China, some works often need to use certain methods to separate the different phases of precipitation in historical precipitation data.

Previous works have discussed the phase identification of precipitations.

Bourgouin (2000) introduced the area-method in separating different precipitation phases, which is based on the vertical thermal structure of the atmosphere, the distribution of condensation nuclei of water vapor, and the descent velocity to predict the precipitation phase. Dai (2008) analyzed the temperature range of precipitation phase change on the continent and the ocean, and discussed the relationship between the phase change temperature and the pressure. Kienzle et al. (2008) proposed to use two input variables (temperature and range) to estimate daily snowfall from precipitation data. Ye et al. (2013) suggested the site-specific threshold values of air temperature and dewpoint to discriminate between solid and liquid precipitation for improving snow and hydrological modeling. Froidurot et al. (2014) pointed out that surface air temperature and relative humidity show the greatest explanatory power.

Sims and Liu (2015) proposed that atmospheric moisture impact precipitation phase and that wet-bulb temperature, rather than ambient air temperature, be used to separate solid and liquid precipitation. Harpold et al. (2017) and Jennings et al. (2018)

pointed out that a humidity phase prediction method had similar or more effective accuracy compared to temperature phase prediction method in separating snowfall from precipitation data. This was also shown by other authors (e.g., Ding et al., 2014;

Harder and Pomeroy, 2013, 2014; Marks et al., 2013; Feiccabrino et al., 2015).

However, in a larger scale study, it is usually difficult to obtain the observational records in the global dataset. To study the separation methods of precipitation phrase on the continental and global scales, only the surface air temperature data are more easily available. Dew point temperature and relative humidity data, for example, can be used only in regional scale investigation, despite their good suitability as indicators of precipitation phrase separation (Harpold et al., 2017a,b; Jennings et al., 2018).

In some hydrological models, the solid-liquid precipitation separation used the double threshold temperature method (Wigmosta et al., 1994; Kang et al., 1999, 2001;

Chen et al., 2008) and the single threshold temperature method (Arnold et al., 1998;

Wang et al., 2004). The customized threshold temperature method has a larger error (Marks et al., 2013; Han et al., 2010). Han et al. (2010) developed an insurance probability method to determine the single threshold temperature and a fitted model in

China.

In this work, we used the daily observational data of the national stations for years 1961–2013 in mainland China, including the long-term records of air temperature, precipitation, relative humidity and visual observations of precipitation phase. We applied the Snowday Direct Definition Method (SDDM) to determine the single threshold temperature values of rainfall and snowfall in northern China (north of 30 °N). A statistical model of the threshold temperature was established to provide a tool for use in studies of large-scale snowfall climatology and climate change, weather forecasting, and hydrological model parameterization. It is believed that

China has sub-continental scale characteristics of lands and natural conditions, and a diversity of climates and topographic types, and the phase separating methods developed in mainland China should have a better universality in continents and the world.

**2. Data and methods**

2.1 Data

The main purpose of this study was to develop an easy and convenient method for separating solid and liquid precipitation, so that the objective separation of solid and liquid parts of precipitation can be achieved without exhaustive reference of observational data. International exchange data generally only contain the daily temperature and precipitation, with no other reference data, so we have only used the indicators related to temperature and precipitation to develop a method of separation.

The data was obtained from the National Meteorological Information Center of

China Meteorological Administration (CMA). The air temperature, precipitation and relative humidity data were derived from the "China Land Daily Climatic Dataset (V3.0)''. The precipitation phase observation was derived from "China Land Climatic

Data Daily Weather Phenomena Dataset''. All the data have been quality controlled.

Collected since January 1951, the "China Land Daily Climatic Dataset (V3.0)''

contains the daily data of air pressure, surface air temperature (daily mean, daily maximum and daily minimum), precipitation, pan-evaporation, relative humidity, wind speed, sunshine hours, and 0-cm ground temperature from 839 stations The

"China Land Climatic Data Daily Weather Phenomena Dataset'' is the daily records encoded by the 752 national stations in mainland China since 1951. Cross comparison of the two datasets and the examination of station information was performed, and any incomplete temperature, precipitation, relative humidity and weather phenomena data were removed. There are total 623 stations selected for use in the study, all of which meet the demand to have information integrity, sequential continuity, and records of more than 20 years in climate reference period (1981–2010). The data may contain inhomogeneities caused by the relocation and other factors, but they would exert little influence on the analysis results, so the data are not adjusted for homogeneity.

First, the precipitation caused by fog, dew, and frost as well as the trace precipitation was removed, and daily precipitation greater than or equal to 1 mm was taken as the effective precipitation. In this regard, the main consideration is that the international exchange precipitation data only contains no less than 1 mm of daily precipitation. In the separation of daily rainfall (pure rain), mixed-phase events, and snow (pure snow) events, 'pure rain' was recorded when the weather phenomenon data indicate that only rain occurred on that day without snow and mixed-phase events; it was registered as 'pure snow' when only snowfall occurred without rain and mixed-phase events, and 'mixed-phase events' when there is rain and snow in the same day, in the records of weather phenomenon data. The daily maximum and minimum temperature during an occurrence of mixed-phase events at each station were recorded as the reference thresholds for the snow and rain temperature threshold values.

When there is less snowfall at the station in lower latitude zone or more arid regions, there may be arbitrary cases of snowfall. An example is from Lijiang station,

Yunnan, located in 26°N, at which pure snow occurred only six times in the 30 years from 1981 to 2010. The representation of the threshold temperature would be poor in these cases. In order to ensure that the snowfall frequency is great enough and the threshold temperature is representative, we took 324 stations (Fig. 1) in northern

China for use in this study. They are generally located north of the Yangtze River, approximately consistent with the January mean temperature isotherm of 3°C or the

30°N parallel. The days with the snowfall records during 1981-2010 were greater than or equal to 100d for each of the stations. In order to avoid the influence of extreme values on the determination of threshold temperature, the maximum and minimum daily mean temperature in each of the precipitation phases were not counted.

For the cases of extreme large rain and snow records at the stations, comparison was made to ensure that the minimum and maximum temperature was correct by examining the weather phenomena, surface air temperature and precipitation on the same day. When mixed-phase events occurred, the range of daily mean temperature was generally large. Threshold temperature was determined only for pure rain and pure snow; the daily mean temperature on a mixed-phase event day was only taken as the reference temperature threshold value.

2.2 Methods

According to the method of China's physical geographical regionalization, mainland China is divided into three natural geographical regions: Eastern Monsoon Region (I, 231 stations), Northwest Arid Region (II, 67 stations), and Qinghai-Tibetan Plateau Region (III, 26 stations) (Fig. 1). More stations are distributed in Eastern Monsoon Region, and there air only 26 stations in Qinghai-Tibetan Plateau Region. A vast region of western part of the Qinghai-Tibetan Plateau is the well-known no-man land without climatic observations, and this would affect the analysis in some extents. The representative station of the Eastern Monsoon Region is Zhaozhou station (Zhaozhou-I hereafter) in Heilongjiang province, which has the lowest threshold temperature of snowfall and rainfall in the country. The representative station of the Qinghai-Tibet Plateau Region is Shiquanhe station (Shiquanhe-II hereafter) in Tibet Autonomous Region, which has the highest threshold temperature of snowfall and rainfall in the country. There are relatively fewer precipitation events in the Northwest Arid Region, and Balikun station (Balikun-III hereafter) in Xinjiang Autonomous Region was selected as the representative station because it observed relatively more precipitation events, and the rain, mixed-phase events, and snow events were evenly distributed. The station is also far from the two other regions (Table 1).

[Figure]

FIG.1. Regionalization and distribution of 324 national stations north of 30 ˚N in mainland China (I: East Monsoon Region; II: Northwest Arid Region; III: Qinghai-Tibetan Plateau;

Blue triangle: stations in the East Monsoon Region; Green diamond: stations in the Northwest

Arid Region; Red circle: stations in the Qinghai-Tibetan Plateau.

The purple diamond denotes the representative stations in different regions: Zhaozhou of Region

I; Balikun of Region II; Shiquanhe of Region III)

Table 1 Information of representative stations in the three regions

| Station name | Zhaozhou | Balikun | Shiquanhe |
| --- | --- | --- | --- |
| Province | Heilongjiang | Xinjiang | Tibet |
| Climate zone | I | II | III |
| Elevation(m) | 148.7 | 1679.4 | 4278.6 |
| Latitude(N) | 45° 42′ | 43° 36′ | 32° 30′ |
| Longitude(E) | 125° 15′ | 93° 03′ | 80° 05′ |

The relative error (RE) of snow days (snowfall) was defined as the percentage (%)

of the difference between simulated snow days (snowfall) and observational (true)

snow days (snowfall) to the observational (true) snow days (snowfall), which could be used to indicate the effectiveness of simulated results.

The establishment of model was realized using the stepwise regression analysis method included with the SPSS Statistics 17.0. The advantage of stepwise regression is that the number of the arguments contained in the regression equation is fewer, it is easy to apply, the root mean squared error (RMSE) is small, and the model created is more stable.

The maximum daily mean temperature at the occurrence of snowfall at the weather station is Tsm, the minimum daily mean temperature at the time of rainfall is

Trn; the number of snowfall days between Trn and Tsm is Sn, the number of rain days is Rn, and the total number of rain and snow days between Trn and Tsm is Nsr = Sn +

Rn; the single critical temperature of rain and snow days is Tt-d, that is, the precipitation event that occurs when the daily mean temperature is lower than Tt-d is considered to be a snowfall event, otherwise it is considered as a rainfall event; the single critical temperature estimated by the statistical model is Tt-p.

Using the Snowday-Direct-Definition-Mothod (SDDM) to define the threshold temperature of precipitation phase, the calculation steps were as follows:

First, find the Trn and Tsm in the dataset of the 623 stations, and count Sn, Rn, and Nsr. Second, calculate the daily average temperature of Nsr and sort it in ascending order. Last, the average of daily mean temperature of the $Sn^{th}$ day and the

$(Sn+1)^{th}$ day was calculated, and it was taken as the threshold temperature (Tt-d) of the rain and snow days. For the area where pure rain and snow events did not overlap (Tsm<Trn, that is, the snowfall and rainfall events did not intersect in the sorted daily average temperature series), the average of Tsm and Trn was taken as the Tt-d. The average of Tt-d and the daily mean temperature of mixed-phase events day was taken as the Tt-d when Tt-d was not in the range of mixed-phase events day daily mean temperature. The Tt-ds values in this study were all within the daily mean temperature of mixed-phase events day, however, and this operation was not required.

**3. Threshold temperature**

3.1 Daily mean temperature corresponding to precipitation in different phases

Figure 2 and Table 2 show phase temperature distribution of precipitation events at the stations. The total precipitation events at 324 stations were included in the statistical calculations, and their corresponding daily mean temperature values (Fig.

2a) were examined: only snowfall occurred when the daily mean temperature was below -12.9 ℃; only rainfall occurred when the daily mean temperature was higher than 22.1 ℃; and the three phases of snow, rain, and mixed-phase events occurred when the temperature was between -12.9 ℃ and 22.1 ℃.

In northern China, pure snow (snowfall) events occurred when the daily mean temperature was below 8.5 ℃, and 95% of the snowfall events occurred when the daily mean temperature was lower than 2.7 ℃ and higher than -16.6 ℃ (Fig. 2a). All pure rain events (rainfall) occurred when the daily mean temperature was higher than

-4.9 ℃, and 95% occurred when the temperature was lower than 26.0 ℃ and higher than 6.4 ℃. All mixed-phase events appeared in the temperature range of -12.9–

22.1 ℃, with 95% occurring when the daily mean temperature was lower than 8.3 ℃

and higher than -1.6 ℃.

[Figure]

**FIG.2. Precipitation phase temperature distribution of regional average and representative stations (a-324 stations; b-Zhaozhou-I; c-Balikun-II; d-Shiquanhe-III)**

At Zhaozhou-I station (Fig. 2b), the pure snow events all occurred when the daily mean temperature was lower than -0.9 ℃, pure rainfall occurred when the daily mean temperature was higher than -3.4 ℃, and mixed-phase events occurred in case of -4.5– 6.5 ℃. Zhaozhou-I station had the lowest threshold temperature (-1.2 ℃) of snowfall and rainfall in the study region. At Balikun-II station (Fig. 2c), the pure snow events all occurred when the daily mean temperature was lower than 5.1 ℃, pure rain events occurred when the daily mean temperature was higher than 4.1 ℃, and mixed-phase events occurred within a temperature range of -7.8–12.3 ℃. At Shiquanhe-III station (Fig. 2d), the pure snow events all occurred when the daily mean temperature was lower than 6.4 ℃, pure rainfall occurred when the daily mean temperature was higher than 6.1 ℃, and mixed-phase events occurred when the temperature was from -5.3 ℃ to 16.0 ℃. Shiquanhe-III station had the highest threshold temperature (6.3 ℃) of snowfall and rainfall in the whole region.

**Table 2 The distribution range of daily mean temperature under different phases of precipitation**
**at stations**

| Station | Snow day mean temperature (℃) | | | | | Mixed-phase events day mean temperature (℃) | | | | | Rain day mean temperature (℃) | | | | |
|---|---|---|---|---|---|---|---|---|---|---|---|---|---|---|---|
| | Max | Min | Ave | 5% value | 95% value | Max | Min | Ave | 5% value | 95% value | Max | Min | Ave | 5% value | 95% value |
| All | 8.5 | -35.4 | -5.2 | -16.6 | 2.7 | 22.1 | -12.9 | 3.6 | -1.6 | 8.3 | 33.3 | -4.9 | 16.3 | 6.4 | |
| Zhaozhou-I | -0.9 | -20.5 | -10.2 | -18.6 | -3.3 | 6.5 | -4.5 | 1.6 | -4.5 | 5.5 | 27.5 | -3.4 | 17.8 | 6.1 | 25 |
| Balikun-II | 5.1 | -22.2 | -8.2 | -17.6 | 0.8 | 12.3 | -7.8 | 4.1 | -2.5 | 9.5 | 22.1 | 4.1 | 14.3 | 7.3 | 19.4 |
| Shiquanhe-III | 6.4 | -18.1 | -4.4 | -14.3 | 4.8 | 16 | -5.3 | 4.3 | -5 | 13.1 | 18.7 | 6.1 | 12.6 | 8.7 | 15.7 |

It can be seen from Fig. 2 and Table 2 that there is a larger difference of the maximum daily mean temperature of snowfall (extreme threshold temperature of snowfall) and the minimum daily mean temperature of rainfall (extreme threshold temperature of rainfall) among the stations.

Figure 3 shows that there is a common spatial distribution feature in the Tsm, Trn and the average daily mean temperature of mixed-phase events in northern China, with the high values generally in the Tibetan Plateau and southern Xinjiang, while the low values mostly in eastern and northern Xinjiang. At the stations analyzed, most have a relationship of Trn<Tsm, that is, the minimum daily mean temperature at the time of a rain event is lower than the maximum daily mean temperature at the time of a snowfall event. Only in a few of places in Northwest Arid Region, is the maximum daily mean temperature of a snow day lower than the minimum daily mean temperature of a rain day, indicating that the pure rain and snow events do not overlap.

[Figure]

**FIG.3. The distribution of daily mean temperatures when precipitation occurs (a. Tsm; b. Trn; c.**
**average daily mean temperature of mixed-phase events; and d. difference between Tsm and Trn)**
**(Red thick line represents 0℃ isotherms)**

3.2 Threshold temperature determination

Figure 4 shows the distribution of the relative error of the snow days and snowfall in northern China, determined by the threshold temperature as mentioned in section

Data and Methods, to the actual snow days and snowfall counted by using weather phenomenon records. The relative error of snow day was smaller. This is due to the definition of threshold temperature being directly determined by snow-day mean temperature. Since the daily mean temperature of the $Sn^{th}$ day and the $(Sn+1)^{th}$ (or more) day is the same under this definition, however, there will be a slight positive bias in the threshold temperature of the same temperature day, with a range of relative errors (0, 2.3%).

The spatial distribution of the relative error of the snowfall was mainly positive, which is due to the systematic deviation of the method. Larger deviation appeared in eastern part of the Qinghai-Tibetan Plateau and the Yangtze-Huaihe River Basins. These areas have more precipitation and sufficient water vapor. Under the same water vapor condition, the observed rainfall was greater than the observed snowfall, and the amount of snowfall determined by the threshold temperature was slightly large, with the certain sites even larger. Small values occurred in the southeastern Northeast China, the border zone between Inner Mongolia and Xinjiang, and western Xinjiang, with the main reason related to the less precipitation and insufficient water vapor. Overall, the relative error of snowfall is between -5% and 20%. There were 312 stations (more than 96%) with deviation less than or equal to 10%, and the absolute value of the relative error was less than 5% in most areas.

[Figure]

**FIG.4.    The spatial distribution of the relative error of the days (a) and amount (b) of snowfall determined by the threshold temperature (Tt-d) in northern China**

The spatial distribution of the threshold temperature (Tt-d) of rain and snow at the stations north of 30 °N are shown in Fig. 5. The average Tt-d is 2.3 °C for Eastern Monsoon Region, 3.4 °C for Northwest Arid Region, and 5.2 °C for the Qinghai-Tibetan Plateau. The highest threshold temperature of the study region is 6.3 °C (Shiquanhe-III, Fig. 2d), the lowest is -1.2 °C (Zhaozhou-I, Fig. 2b), the threshold temperature range was 7.5 °C, and the average threshold temperature for the whole region was 2.8 °C. The high-values were in the northern Qinghai-Tibetan

Plateau, with a threshold temperature of more than 4℃, and the low-values were generally in eastern Northeast China, North China, and northern Xinjiang with the threshold temperature mostly less than 2℃. The threshold temperature west of 100℉

showed an approximately zonal distribution, and it decreased with the increase of latitude; the east of 100℉ had a meridional distribution, and the threshold temperature decreased with increasing longitude. There are some uncertainties on the distribution of the threshold temperature in the Qinghai-Tibetan Plateau and northwestern deserts mainly due to the interpolation in the regions with sparser observations.

[Figure]

**FIG.5. Spatial distribution of threshold temperature of precipitation phases in northern China (I:**
**East Monsoon Region; II: Northwest Arid Region; III: Qinghai-Tibetan Plateau. Unit: ℃)**

3.3 Correlation between threshold temperature and geographical/climatic factors

The threshold temperature (Tt-d) is related to the longitude, latitude, altitude, annual precipitation, annual mean air temperature, and annual relative humidity of the observational sites, with a positive correlation with altitude and a negative correlation with the other factors. All the correlations passed the significant test ($p$=0.05) (Fig 6).

In areas where the annual mean temperature is lower, the threshold temperature was generally higher and more variable, while in areas with higher annual mean temperature, it was generally slightly lower and relatively less variable. The threshold temperature had a decreasing trend with increase of annual mean air temperature (Fig. 6a). In lower altitude area, the threshold temperature was lower, while it was higher in mountains and plateaus, and a highly significant increasing trend of threshold temperature with altitude can be seen (Fig. 6b). There was a negative correlation between the threshold temperature and the annual precipitation, and a more significant negative correlation with the annual relative humidity (Fig. 6c, d).

[Figure]

**FIG.6. Relationship of the threshold temperature (Tt-d) with temperature (a), altitude (b), annual precipitation (c) and annual mean relative humidity (d) in northern China**
**(Blue triangle: East Monsoon Region; Green diamond: Northwest Arid Region; Red circle: Qinghai-Tibetan Plateau)**

It is possible that the relationship of the threshold temperature with longitude and latitude is also related to the variations of altitude and relative humidity in the study region. The altitude and relative humidity generally decrease from west to east and from south to north, and the altitude and relative humidity have better correlations with the threshold temperature, which may be the reason why threshold temperature decreases with the increase of the longitude and latitude. Therefore, altitude and relative humidity may be the more important factors in determining the threshold temperature.

The threshold temperature was positively correlated with altitude, which may mainly be because the ground surface receives stronger solar radiation, causing the boundary-layer atmosphere to heat rapidly in the high altitude areas during daytime. However, the upper air temperature is low, the temperature lapse rate is larger, the cloud bottom-height is low, and the path of snowflakes is short, so the snowfall phenomenon can also be more frequently observed when the daytime surface air temperature is high.

The threshold temperature was negatively correlated with annual precipitation in particular with relative humidity, which may be related to the low latent heat flux and high sensible heat flux in arid area. When the sensible heat flux is high, the ground surface air temperature is high, and the temperature lapse rate is large. In the case of the same condensation height or cloud bottom-height, snowfall is more likely to occur under the condition of higher surface air temperature. It is also possible that the higher the threshold temperature in arid area than in humid area is caused by a difference of the more complicated microphysical processes around the snowflakes between the two climatic conditions.

3.4 Establishment of the threshold temperature model

Considering that the relative humidity data of some areas is difficult to obtain, the precipitation factor was selected as the independent variable. Using the SPSS software stepwise regression analysis method, a statistical model of threshold temperature was established with annual mean air temperature, altitude, and annual precipitation as influential factors. The model, which passed the significant test ($p$=0.05), can be expressed as follow:

Tt-p = 1.69147 + (.09585) * T + (.001311) * H + (-0.00172) * R        (1)

where Tt-p is the simulated threshold temperature (℃), T is the annual mean air temperature (℃) of the station, H is the altitude of the station (m), and R is the annual precipitation of the station (mm).

The correlation coefficient between Tt-p and Tt-d (threshold temperature determined by using the synoptic phenomena) is 0.88. The median and standard deviation of the simulated threshold temperature (Tt-p) were 2.54℃ and 1.17℃, which were close to the median (2.64℃) and standard deviation (1.33℃) of the Tt-d.

The maximum simulated threshold temperature was 5.9 ℃, minimum was -0.4 ℃, temperature range was 5.5 ℃, and average simulated threshold temperature was

2.8 ℃ for the whole region. The maximum positive deviation of the Tt-p to the Tt-d was 2.9 ℃, and the minimum negative deviation was -1.8 ℃. The numbers of stations with relative error less than 10% for snow day and snowfall reached 97% and

92% respectively.

In the East Monsoon Region (Region I), the simulated threshold temperature was generally lower than the Tt-d (0.026 ℃ lower in Region I on average). However, it was higher in the Northwest Arid Region (Region II) (0.063℃ higher on average) and the Qinghai-Tibetan Plateau Region (0.065℃ higher on average) (Fig. 7).

[Figure]

**FIG.7. Simulated threshold temperature (Tt-p), threshold temperature (Tt-d) and their difference**

**for observational stations in different regions of northern China (Region 1: East Monsoon Region;**

**Region 2: Northwest Arid Region; Region 3: Qinghai-Tibetan Plateau Region)**

The mean absolute errors (MAE) of threshold temperature of the simulated snowfall are 0.476℃ for East Monsoon Region, 0.560℃ for Northwest Arid Region, and 0.435℃ for Qinghai-Tibetan Plateau Region. Fig. 8 (and also Table 3) shows the spatial distribution of the absolute errors of threshold temperature of the simulated snowfall and snow days. Larger positive errors can be seen in the Northwest Arid

Region and western Qinghai-Tibetan Plateau Region for snow days.

[Figure]

[Figure]

**FIG. 8. Absolute error distribution of snowfall days (a) and snowfall (b) for the simulated threshold temperature Tt-p**

**Table 3 Comparison of statistics and the errors resulting from the simulated threshold temperature Tt-p**

|          |                 | North of China | Region I | Region II | Region III |
|----------|-----------------|----------------|----------|-----------|------------|
|          | Correlation     | 0.999          | 0.998    | 0.999     | 0.999      |
| Snowday  | MAE (d)         | 7.77           | 7.52     | 5.21      | 16.69      |
|          | MRE (%)         | 2.71           | 2.98     | 1.68      | 2.98       |
|          | RMSE (d)        | 3.9            | 4.3      | 2.3       | 3.6        |
|          | Correlation     | 0.997          | 0.996    | 0.999     | 0.998      |
| Snowfall | MAE (mm)        | 37.18          | 37.7     | 19.67     | 77.6       |
|          | MRE (%)         | 3.71           | 4.09     | 2.1       | 4.44       |
|          | RMSE (mm)       | 73.78          | 76.13    | 43.81     | 95.24      |

Figure 9 shows spatial distribution of the relative error of the simulated snow days (Fig. 9a) and snowfall (Fig. 9b) relative to the actual snow days and snowfall at the stations. The relative error range of snowfall days in northern China was between -16.8% and 17.0%, with an average of -0.1%; the relative error was smaller in mid-southern parts of the study region, and larger in the coastal areas and the northern Qinghai-Tibetan Plateau. In the Qinghai-Tibetan Plateau Region, the medians of the simulated snow days were smaller than those of the actual snow days, and the relative errors were larger. This may be related to the fact that the snowfall days in northern

Tibetan Plateau fluctuated greatly, with some years with larger numbers of snowfall days. The relative error range of snowfall in the whole region was between -15.5%

and 29.0% with an average of 1.1%, and the spatial distribution was basically the same as that of the relative errors of snow days.

[Figure]

**FIG.9. Relative error distribution of snowfall days (a) and snowfall (b) defined by the simulated**
**threshold temperature Tt-p**

Affected by the extremely low air temperature and the abnormally deficient water vapor due to the East Asian winter monsoon, the snow days (snowfall) with only snowfall weather were relatively less frequent (low) in northern China, as compared to other regions of the same latitude; therefore, it is more likely that the relative error is large in the study region. However, the relative error range shown here is acceptable, and the fitting effect is generally good.

The RMSE of the relative error of snow days was 3.9, and the RMSE of the relative error of snowfall was 5.6. The annual snow days and the amount of snowfall were less in the mid-southern parts of the study region which had negative relative errors of the simulated snow events; however, snow days and snowfall were slightly more numerous in the northern part of the Sichuan Basin. The number of snow days and snowfall was less in the coastal area which had positive relative errors of the simulated snow events, while there were more snow days and snowfall in the northern

Qinghai-Tibetan Plateau. The relative error of snow days (snowfall) and the threshold temperature had a correlation coefficient of -0.40 (-0.32); both passed the significant test ($p$=0.05). It can be seen that the relative error in the area with low threshold temperature tends to be positive, and that relative error in the area with high threshold temperature is generally negative.

**4. Discussion**

China has a vast territory. The study region across the latitude range 30–54 N, and a longitude range of 73–136 E, with various climate types of temperate monsoon zone, continental arid zone and alpine including the highest mountainous system of the Qinghai-Tibetan Plateau. The complex and diverse geophysical and climatic condition makes the region ideal for understanding the transition of precipitation phrase and developing a method to separate the different precipitation phrase.

We made an attempt to develop such a method to separate the precipitation phases by using a high-quality daily observational dataset in this paper. Our study not only determined the threshold temperature with more reliable results, but also tested the statistical model of threshold temperature, provided the results of the model and the relative error range for different regions, and confirmed the applicability of the method in the complex geographic area with diverse climate types.

With the method of determining threshold temperature developed in this paper, the relative error of snow days and snowfall calculated for most of the stations was very small, and the stations with less than 10% relative errors accounted for 97% and 92%, respectively. This method could be used to better determine the snow days than the snowfall, with the relative error of snowfall was slightly larger in the Huaihe River basin. This is mainly because, when using the threshold temperature to calculate the amount of snowfall, rain days with a daily mean temperature below the threshold temperature could be identified as the snow day, and also some snow days with a daily mean temperature above the threshold temperature could be classified as rain days. In the frequent transformation of the precipitation phases (early spring and early winter), precipitation on a rain day is often greater than that on a snow day, so the priority to ensure the determination of a snow day, the estimated relative error of snowfall would be a little larger.

Han et al. (2010) used the insurance probability method to determine the single threshold temperature of rain and snow, taking the daily mean temperature of the 98.5% insurance probability of rainfall and snowfall, 50% insurance probability of mixed-phase events as the threshold temperature. For comparison of SDDM and the insurance probability method, the number of snow days (Sn) and rain days (Rn) between $T_{rn}$ and $T_{sm}$ was calculated, respectively, using the two methods. The corresponding daily mean temperature at the insurance probability of the snow and rain days between [Trn, Tsm], X (x∈ (0–99%)) (at 1% intervals), was estimated. For example, the number of rain days and snow days between Trn and TSM is 100d respectively; when x = 90% is taken, the rain day temperature Tr90 corresponds to the insurance probability of 90%, that is, to ensure the minimum daily mean temperature in the event of 90% rain days between Trn and TSM, while Ts90 is to guarantee that the maximum daily mean temperature in the event of 90% snowfall days is between

Trn and TSM. The arithmetic mean of each station's Trx and Tsx is defined as the threshold temperature Tt-x at the station's insurance probability x.

The threshold temperature (Tt-x) was calculated according to the insurance probability method, and the threshold temperature (Tt-d) was obtained based on the definition in this paper; the relative error comparison is presented in Table 4. For simplicity, the insurance probability interval in the table was taken as 10%. The maximum, minimum, and range of the threshold temperature (Tt-x) under different insurance probability, and of the (Tt-d), in northern China, are given in the table; at the same time, the maximum, minimum, and range of the relative error of the snow days and snowfall, as well as the number of stations with a relative error less than or equal to 10%, are also given.

**Table 4 Comparison of statistics and the relative errors resulting from threshold temperature Tt-x**
**and Tt-d**

| | Threshold temperature (℃) | | | Relative error of snow days (%) | | | | Relative error of snowfall (%) | | | |
|---|---|---|---|---|---|---|---|---|---|---|---|
| | max | min | max−min | max | min | max−min | Stations <10% | max | min | max−min | Stations <10% |
| Tt-0 | 6.4 | −2.3 | 8.7 | 30.2 | −11.1 | 41.3 | 311 | 36.6 | −15.3 | 51.9 | 280 |
| Tt-10 | 6.4 | −2.3 | 8.7 | 25.1 | −11.1 | 36.3 | 313 | 29 | −11.8 | 40.8 | 284 |
| Tt-20 | 6.5 | −2.3 | 8.8 | 25.1 | −9.5 | 34.6 | 316 | 29 | −9.7 | 38.7 | 287 |

| | | | | | | | | | | |
|---|---|---|---|---|---|---|---|---|---|---|
| Tt-30 | 6.5 | −2.2 | 8.7 | 23.6 | −7.1 | 30.7 | 314 | 31.5 | −15.3 | 46.8 | 287 |
| Tt-40 | 6.4 | −2.2 | 8.6 | 23.6 | −5.8 | 29.4 | 316 | 31.5 | −8.4 | 39.9 | 289 |
| Tt-50 | 6.5 | −2 | 8.5 | 21.1 | −5.7 | 26.8 | 312 | 32.2 | −9.7 | 41.9 | 286 |
| Tt-60 | 6.4 | −1.5 | 7.9 | 19.1 | −6.5 | 25.6 | 313 | 32.2 | −9.7 | 41.9 | 289 |
| Tt-70 | 6.4 | −1.4 | 7.8 | 15.6 | −6.5 | 22.1 | 314 | 30.2 | −6.2 | 36.4 | 283 |
| Tt-80 | 6.7 | −1.4 | 8.1 | 18.3 | −5.8 | 24 | 307 | 45.2 | −8.4 | 53.6 | 282 |
| Tt-90 | 6.5 | −1.2 | 7.7 | 23 | −7 | 29.9 | 306 | 33.4 | −9.7 | 43.1 | 276 |
| Tt-d | 6.3 | −1.2 | 7.5 | 2.6 | 0 | 2.6 | 323 | 20.2 | −4.3 | 24.5 | 312 |

Table 4 shows that, using the insurance probability method, the test results of the threshold temperature (Tt-70), obtained when the insurance probability x = 70% was taken, represented the best values, as the difference between the minimum and maximum values of the threshold temperature was small, and the relative errors were small, with the relative error of the snow days at 314 stations ≤10%, and that of the snowfall at 283 stations ≤ 10%.

The range of threshold temperature Tt-d of snow days determined in this paper was less than that of the Tt-70. The relative error of snow days was obviously small, and the relative error of snowfall was much less than that of the Tt-70, with more stations having the relative errors ≤10% for both snow days and snowfall. Therefore, the method developed in this paper has an advantage over the insurance probability method developed in the previously works.

Ding (2014) used a similar method as that in this paper to determine the precipitation types with daily mean wet-bulb temperature, relative humidity and surface elevation as predictors. Compared to the other nine schemes used in hydrological and land surface models, their method showed a better accuracy. Our analysis also finds the better correlations of relative humidity and surface elevation with the real thresholds in the study region, but wet-bulb temperature has not been assessed in our work.

In order to compare to the two works, the accuracy of Tt-d in our work and that of Ding et al. (2014) is tested over the same air temperature range. The results are shown in Table 5. The Tp-CH in our work is the Tt-p of all stations in the study area.

The Tp-R1 is the Tt-p of stations in East Monsoon Region, the Tp-R2 is the Tt-p of

Northwest Arid Region, and the Tp-R3 is the Tt-p of the Qinghai-Tibetan Plateau.

**Table 5   Accuracy (%) of methods by Ding et al. (2014) and this work (SDDM) for the air**

**temperature range [0℃, 4℃]**

|  | Ding et al. (2014) | Tp-CH | Tp-R1 | Tp-R2 | Tp-R3 |
|---|---|---|---|---|---|
| China | 59.3 | 87 | 87 | 84 | 84 |
| R1 | 53.1 | 83 | 83 | 78 | 80 |
| R2 | 60.1 | 89 | 89 | 90 | 88 |
| R3 | 66.1 | 99 | 99 | 99 | 99 |

Ding et al. (2014) separated rain, snow and sleet or mixed-phase events, but our work only separates rain and snow. This may be the reason that the accuracy of our method is higher. However, the test result of Tt-p for each region is relatively stable.

Therefore, the SDDM as developed in this paper is stable and reliable for the separation of rain and snow. More importantly, the SDDM has potential to be applied in a broader area where the daily wet-bulb temperature and relative humidity data are unavailable. Liu and Ren et al. (2018) recently tested the possibility of extrapolating the statistical model of separating solid precipitation from liquid precipitation.

Considering less snowfall in the lower latitudes and less rainfall in the higher latitudes, the proposed area for use of the method is between 30 N to 60 N in the latitude range, and areas outside this range may have large deviation.

When building the statistical model of threshold temperature, we actually also built a model containing the relative-humidity as predictor for possible use in the region where the daily mean relative-humidity data can be obtained.

$$T_{t-p} = 2.572889 + (.089936) * T + (-.01646)*U + (.000931) * H + (-0.00089) * R \qquad (2)$$

where U is daily mean relative humidity (%) of the observational station, and the other terms are the same as Formula (1).

The test of this model showed that the Formula (2) is indeed relatively better than the formula (1), but the advantage is not so obvious. Modeled thresholds using the formula including relative humidity have a very similar spatial distribution pattern with that determined by the formula without the term. In most areas where the relative-humidity data cannot be obtained, therefore, Formula (1) can be confidently used, which has little effect on the separation results.

In this paper, only the two phases of pure snowfall and pure rainfall were determined, however, and the mixed-phase events were not analyzed. In the case of mixed-phase events, the surface air temperature changed greatly during a day; there was probably mixed-phase events, pure rain and pure snow in the same day, the threshold temperature fluctuations were large, and it would be difficult to accurately determine and simulate. Because the method used in this paper did not quantify the mixed-phase events, when precipitation was separated into solid and liquid state, the mixed-phase events will be classified as snow when the daily mean temperature is lower than the threshold temperature, and as rain when the daily mean temperature is higher than the threshold temperature, causing a certain error. However, for the study of large-scale snowfall climatology, especially for studies of the larger than sub-continental scale snowfall climate change, the snow and rain separation method presented in this paper could well meet the needs.

5. **Conclusions**

Based on the analysis of the historical daily temperature, precipitation, and weather phenomenon observation data in northern China, the threshold temperature model for determining the phase of rain and snow was established and tested. The main conclusions are as follows:

(1) The threshold temperature value of rain and snow determined based on weather phenomenon data is between -1.2–6.3 ℃, with a temperature range of 7.5 ℃

and an average value of 2.81 ℃. The high values were in the northern

Qinghai-Tibetan Plateau, reaching more than 4 ℃, and the low values were found in

Northeast China, North China, and northern Xinjiang Autonomous Region, generally less than 2 ℃. The west of 100°E showed an approximately zonal distribution, and the threshold temperature decreased with latitude; the east of 100°E had a meridional distribution, and the threshold temperature decreased with increasing longitude.

(2) The threshold temperature was more variable in the low latitude areas, while it was slightly lower and relatively centralized in the high latitudes, with a clear decreasing trend with increase of latitude. The threshold temperature was lower at low altitudes, higher in the high altitude areas, and increase with altitude. There was a statistically significant negative correlation between the threshold temperature and annual total precipitation and annual mean relative humidity, with the negative correlation with relative humidity especially significant.

(3) A statistical model based on latitude, elevation, and annual precipitation can be used to simulate the threshold temperature of the precipitation phase in northern

China, with less relative error in simulated snow days and snowfall. The stations with relative error less than 10% reached 97% and 92% for the snow days and snowfall respectively.

**Acknowledgements:** This study is financed by the Ministry of Science and Technology of China (Fund No: 2018YFA0605603) and the Natural Science Foundation of China (NSFC) (Fund No: 41575003 and 41771067).

**References**

Arnold JG, Srinivasan R, Muttiah RS, et al . 1998: Large area hydrologic modeling and assessment; part I: model development. Journal of the American Water Resources Association, 34(1): 73-89. DOI: 10.1111/j.1752-1688.1998.tb05961.x

Becker A, Finger P, Meyer-Christoffer A, et al. 2012: A description of the global land-surface precipitation data products of the Global Precipitation Climatology Centre with sample applications including centennial (trend) analysis from 1901-present, Earth Syst. Sci. Data Discuss., 5, 921–998.

Blanchet J, Marty C, Lehning M. 2009: Extreme value statistics of snowfall in the Swiss Alpine region. Water Resour. Res., 45. DOI:10.1029/2009WR007916

Bourgouin P. 2000: A method to determine precipitation types. Weather and Forecasting, 15(10): 583-592.

Chen RS, Kang ES, Lu SH, et al. 2008: A distributed water-heat coupled model for mountainous watershed of an inland river basin in Northwest China (I) model structure and equations. Environmental Geology, 53: 1299- 1309.

Dai AG.2008. Temperature and pressure dependence of the rain-snow phase transition over land and ocean. Geophysical Research Letters, 2008, 35: 62-77

Dang HF, Zhao LN, Gong YF. 2015: Temporal and Spatial Distributions of Summer Precipitation Analysis in China, Journal of Chengdu University of Information Technology, 30(6): 609-615.

Ding B, Yang K, Qin J, et al. 2014. The dependence of precipitation types on surface elevation and meteorological conditions and its parameterization. J. Hydrol. 513, 154–163.

Fang XQ, Liu CH, Hou GL. 2011: Reconstruction of Precipitation Pattern of China in the Holocene Megathermal, Scientia Geographica Sinica, 31(11): 1287-1292.

Feiccabrino, J., Graff, W., Lundberg, A., Sandström, N., Gustafsson, D., 2015: Meteorological Knowledge Useful for the Improvement of Snow Rain Separation in Surface Based Models. Hydrology 2, 266–288. https:// doi.org/10.3390/hydrology2040266.

Froidurot S, Zin I, Hingray B, et al.2014: Sensitivity of precipitation phase over the Swiss Alps to different meteorological variables. Journal of Hydrometeorology, 15: 685-696

Han CT, Chen RS, Liu JF, et al. 2010: A Discuss of the Separating Solid and Liquid Precipitations, Journal of Glaciology and Geocryology, 32(2):249-256.

Harder P, Pomeroy J W. 2013: Estimating Precipitation Phase Using a Psychrometric Energy Balance Method. Hydrological Processes.DOI: 10.1002/hyp.9799.

Harder P, Pomeroy J. 2014: Hydrological model uncertainty due to precipitation-phase partitioning methods. Hydrological Processes. DOI: 10.1002/hyp.10214.

Harpold AA, Rajagopal S, Crews JB, et al. 2017: Relative humidity has uneven effects on shifts from snow to rain over the western U.S. American Geophysical Union. DOI: 10.1002/2017/GL075046.

Harpold, A.A., Kaplan, M., Klos, P.Z., Link, T., McNamara, J.P., Rajagopal, S., Schumer, R., Steele, C.M., 2017a: Rain or snow: hydrologic processes, observations, prediction, and research needs. Hydrol Earth Syst Sci, 21: 1–22.

Jennings KS, Winchell TS, Livneh B, et al. 2018. Spatial variation of the rainsnow temperature threshold across the Northern Hemisphere. Nat. Commun. 9. DOI:10.1038/s41467-018- 03629-7.

Jiang Y, Qian WH. 2003: Regional characteristics of Heavy Snowfall or Snowstorm in Inner Mongolia. ACTA GEOGRAPHICA SINICA. 58 (Suppl.): 38-48.

Kang ES, Cheng GD, Lan YC, et al . 2001: Alpine-runoff simulation of the Yalong River for the South-North Water Diversion. Journ al of Glaciology and Geocryology, 23(2) : 139- 148.

Kang ES, Cheng GD, Lan YC, et al . 1999: A model for simulating the response of run off from the mountainous watershed of inland river basins in the arid area of northwest China to climatic changes. Science in China, Series D, 42 (Suppl.): 52- 63.

Kienzle S, 2008: A new temperature based method to separate rain and snow. Hydrol. Processes, **22**, 5067-5085. DOI: 10.1002/hyp.7131.

Kunkel KE, Palecki MA, Ensor L, et al. 2009: Trends in twentieth-century U.S. extreme snowfall seasons. Journal of Climate, 22(23): 6204-6216.

Loth B, Graf H, Oberhuber JM. 1993: Snow cover model for global climate simulations. Journal of Geophysical Research: Atmospheres. (1984–2012) 98(D6): 10451-10464.

Liu JH, Zhang PZ, Meng CH, et al. 2011: A Comprehensive Study on Precipitation Change Over Monsoon Marginal Areas During Last Five Centuries. SCIENTIA GEOGRAPHICA SINICA, 31(4):401-411.

Liu YL, Ren GY, Yu HM. 2012: Climatology of Snow in China, Scientia Geographica Sinica, 32(10):1176-1185.

Liu YL, Ren GY, Yu HM, et al. 2013: Climatic Characteristics of Intense Snowfall in China with Its Variation. Journal of Applied Meteorological Science, 24(3):304-313.

Liu YL, Ren GY,Sun XB. 2018. Establishment and Verification of Single Threshold Temperature Model for Partition Precipitation Phase Separation. Journal of Applied Meteorological Science. 29(4):449-459.

Marks D,Winstral A,Reba M,et al. 2013. An evaluation of methods for determining during-storm precipitation phase and the rain/snow transition elevation at the surface in a mountain basin, Advanced in Water Resourced,55:98-110.   DOI: 10.1016/j.advwatres.2012.11.012.

Noake K, Polson D, Hegerl G, et al. 2012: Changes in seasonal land precipitation during the latter twentieth-century, Geophysical Research Letters, 39 (3): 136-141. DOI:10.1029/2011GL050405.

O'Hara BF, Kaplan ML, Underwood SJ. 2009: Synoptic climatological analyses of extreme snowfalls in the Sierra Nevada. Weather & Forecasting, 24(6): 1610-1624.

Polson D, Hegerl GC, Zhang X, et al. 2013: Causes of robust seasonal land precipitation changes. Journal of Climate, 26 (17): 6679-6697.

Qin D, Liu S, Li P. 2006:   Snow cover distribution, variability and response to climate change in western China. J. Clim.. 19, (9): 1820 –1833.

Ren GY. 2007.Climate change and China water resources. Beijing: Meteorological press. 314.

Ren GY, Ren YY, Zhan YJ,et al. 2015: Spatial and temporal patterns of precipitation variability over mainland China: II: recent trends. Advances in Water Science, 26(4): 451-465.

Ren GY,Liu YJ,SUN XB, et al. 2016: Spatial and temporal patterns of precipitation variability over mainland China: Ⅲ: causes for recent trends,. Advances in Water Science, 27(3): 327-348.

Sims EM, Liu G.2015. A parameterization of the probability of snow-rain transition. Journal of Hydrometeorology, 16: 1466-1477.

Stewart RE, Thériault JM, Henson W, 2015. On the Characteristics of and Processes Producing Winter Precipitation Types near 0°C. Bull. Am. Meteorol. Soc. 96, 623–639. DOI:10.1175

/BAMS-D-14-00032.1

Wan H, Zhang XB, Francis WZ, et al. 2013: Effect of data coverage on the estimation of mean and variability of precipitation at global and regional scales. Journal of Geophysical Research:

Atmospheres, 118:534-546. DOI:10.1002/jgrd.50118

Wang SG, Kand ES, Li X. 2004: Progress and Perspective of Distributed Hydrological Models. Journal of

Glaciology & Geocryology, 26 (1) :61-65.

Wang CY, Wang SL, Huo ZG, et al. 2005: Progress in research of agro-meteorological disasters in China in recent decade. Acta Meteorologica Sinica, 5:659-671.

Wigmosta MS, Vail LW, Lettenmaier DP. 1994: A distributed hydrology-vegetation model for complex terrain [J] . Water Resources Research, 30(6): 1665- 1679.

WMO. 2013: Reducing and managing risks of disasters in a changing climate. WMO Bulletin, 62(Special

Issue): 23-31.

Wu BY, Yang K, Zhang RH. 2009: Eurasian snow cover variability and its association with summer rainfall in China. Advances in Atmospheric Sciences, 26:31-44.

Vavrus S. 2007: The role of terrestrial snow cover in the climate system. Climate Dynamics, 29(1):73-88.

Xiao C, Yu RC, Yuan WH, et al. 2015: Spatial and temporal differences characteristics of the rainy season of Mainland China. Acta Meteotologica Sinica, 73 (1): 84-92.

Yang LM, Yang T, Jia LH, et al. 2005: Analyses of the Climate Characteristics and Water Vapor of Heavy

Snow in Xinjiang Region, Journal of Glaciology and Geocryology, 27 (3): 389-396.

Ye H, Cohen J, Rawlins M. 2013: Discrimination of solid from liquid precipitation over northern Eurasia using surface atmospheric conditions. Journal of Hydrometeorology, 14 (4): 1345-1355.

Yu RC, Li J, Chen HM, et al. 2014: Progress in studies of the precipitation diurnal variation over contiguous China. Acta Meteorologica Sinica, 72 (5): 948-968.

Zhang ZF, Xi S, Liu N, et al. 2015: Snowfall change characteristics in China from 1961 to 2012,

Resources Science, 37(9): 1765-1773.

Zhong J, Su BD, Zhai JQ, et al. 2013: Distribution Characteristics and Future Trends of Daily

Precipitation in China, Progressus Inquisitiones De Mutatione Climatis, 9 (2): 89-95.

---

## Author Comment (AC5) · 8 Oct 2018

**A new method for determining the single threshold temperature of precipitation phase separation**

Yulian Liu[a,b]   Guoyu Ren[a,c]   Xiubao Sun[a,c]   Xiufen Li[d,e]   Hengyuan Kang[f,]

[a] Department of Atmospheric Science, School of Environmental Science, China University of Geosciences, Wuhan 430074, China;
[b] Heilongjiang Climate Center, Harbin 150030,China;
[c] Laboratory for Climate Studies, National Climate Center, CMA, Beijing100081,China;
[d] Innovation and Opening laboratory of Regional Eco-Meteorology in Northeast, China Meteorological Administration
[e] Heilongjiang Provincial Institute of Meteorological Sciences, Harbin 150030,China.
[f] Harbin Meteorological Bureau, Harbin 150028,China.

Corresponding author: G. Ren (guoyoo@cma.gov.cn)

Submitted to *hydrology and earth system sciences* for possible publication

**Abstract**

Separating the solid precipitation from liquid precipitation in  existing historical precipitation observation data  is a key problem in the monitoring and study of climate anomaly and long-term change of extreme precipitation events in difference phases. Based on the comprehensive analysis of the historical daily air temperature, precipitation data, and visual observations of precipitation phase in the northern areas of Mainland China (north of 30°N), this paper proposes a  to determine the threshold temperature of rainfall and snowfall  for a complex and diverse geographical and climatic region . A statistical model of separating solid  from liquid precipitation was established The main conclusions include: (1) in northern China, the threshold temperature range of the daily mean temperature of rain and snow determined based on weather phenomenon records was  from -1.2°C to 6.3°C, with a difference of 7.5°C among areas, and a mean threshold value of 2.8°C for the whole region. The

 temperature in the northern Tibetan Plateau was the highest (generally higher than 4°C). The low threshold temperature values appeared in eastern Northeast China, North China, and northern Xinjiang Autonomous Region, which were less than 2°C. (2) The  threshold temperature decreased with increase in longitude east of 100°E, but  it was more dispersed in the areas west of 100°E. The threshold temperature was generally higher and more variable in the low-latitude , and  lower and more concentrated in the high-latitude; the threshold temperature  generally increased with altitude. (3) There was a negative correlation between the  threshold temperature and the annual precipitation ; .  was also negatively correlated with the annual average relative humidity. (4) The multivariate regression fitting model developed  based on latitude, altitude, and annual precipitation was able to simulate the threshold temperature of the precipitation phase in northern China well. According to tthedeviationofare smallerdeviation~~ reached 95.1% and 90.7%,

respectively. The results of this study can therefore be applied for the separation of solid and liquid precipitation events in the areas without sufficient weather phenomenon records .

**Key words:** Northern China; Precipitation; Phase;  Separation; Statistical model;  Regional differences

**1. Introduction**

Precipitation is an important parameter used to characterize climate characteristics and climate change, and it is one of the key components of the Earth's water and energy cycles (Loth et al., 1993). The influence of different phases of precipitation on the surface water and energy cycles is enormous (Vavrus, 2007; Wu et al., 2009), as more than 50% of the global meteorological disasters are closely related to  abnormal precipitation, including extreme intense rainfall, heavy snowfall or blizzards, freezing rain and droughts (WMO, 2013; Wang et al., 2005). Under the similar  precipitation amount, the effect of different phases of precipitation on the Earth's surface system and the social and

economic system is clearly different, thus it is important to distinguish and understand the characteristics and anomalies of snowfall or mixed-phase events and their causes. In addition, in  monitoring  long-term changes in extreme precipitation events on sub-continental to global scales, it is also necessary to distinguish rainfall and snowfall events from historical precipitation data.

To date, many studies have been published on the characteristics and multi-decadal variation of snowfall in China (e.g. Jiang et al., 2003; Yang et al., 2005; Qin et al., 2006; Liu et al., 2012, 2013; Zhang et al., 2015). Also, many studies on both the global and Asian regional total precipitation and extreme precipitation events and their long-term change have been reported (Becker et al., 2012; Noake et al., 2012; Polson et al., 2013; Blanchet et al., 2009; O'Hara et al., 2009; Kunkel et al., 2009; Ren, 2007,  2015, 2016; Liu et al., 2011; Fang et al., 2011;  Zhong et al., 2013; Wan et al., 2013; Yu et al., 2014; Xiao et al., 2015; Dang et al., 2015). The analyses of precipitation change generally showed a detectable trend toward more precipitation amount and more frequent extreme rainfall events over the last decades. All of these studies have greatly enriched the understanding of global precipitation and snowfall climatology and the climate change and variability in different regions and varied scales.

There are few direct observations of precipitation phase at the global scale. Researchers can partition precipitation phase, but there are difficulties in doing so at air temperatures near freezing (Ding et al., 2014; Jennings et al., 2018; Stewart et al., 2015).

 Even in the case of relatively abundant meteorological observational data in China, some works often need to use certain methods to separate the different phases of precipitation in historical precipitation data.

Previous works have discussed the phase identification of precipitations . Bourgouin (2000) introduced the area-method in separating different precipitation phases, which is based on the vertical thermal structure of the atmosphere, the distribution of condensation nuclei of water vapor, and the descent velocity to predict the precipitation phase . Dai (2008) analyzed the temperature range of precipitation phase change on the continent and the ocean, and discussed the relationship between the phase change temperature and the pressure.  Kienzle et al. (2008) proposed to use two input variables ( temperature and range) to estimate daily snowfall from precipitation data. Ye et al. (2013) suggested the  site-specific threshold values of air temperature and dewpoint to discriminate between solid and liquid precipitation  for  improving snow and hydrological modeling . Froidurot et al. (2014) pointed out that surface air temperature and relative humidity show the greatest explanatory power. Sims and Liu (2015) proposed  that atmospheric moisture impact precipitation phase and that wet-bulb

temperature, rather than ambient air temperature., should be used to separate solid and liquid precipitation. Harpold et al. (2017) and Jennings et al. (2018) pointed out that a humidity phase prediction method had sSimilar or more effective accuracy compared to temperature phase prediction method in separating snowfall from precipitation data. This was also shown by other authors (e.g., Ding et al., 2014; Harder and Pomeroy, 2013, 2014; Marks et al., 2013).Harpold et al. (2017) and Keith et al. (2018) all point out that a humidity phase prediction method had similar accuracy to temperature phase prediction method in separating snowfall from precipitation data.

After the large-scale freezing rain and snow disaster in Central and South China in winter of 2008, domestic scholars paid more attention to the studies of the discrimination and identification of the precipitation phase, in order to meet the challenge of the disastrous weather forecast (Liu et al., 2013). The discriminant basis is generally the temperature of the surface and upper air layers. Zhang et al. (2013) studied the identification criteria of winter precipitation phase in Beijing, and pointed out that the phase transition in Beijing mainly occurred in March and November. They found six physical quantities closely related to the conversion of snow and rain (850 hPa temperature, 925 hPa temperature, 1000 hPa temperature, thickness between 1000 hPa and 700 hPa, thickness between 1000 hPa and 850 hPa, and the combination of surface air temperature and relative humidity). According to these physical quantities, the objective forecast index of the Beijing winter precipitation phrases was established, and its accuracy reached 77%. You et al. (2013) also analyzed the discriminant index of precipitation phases in Beijing, pointing out that precipitation is

considered as rainfall when the surface air temperature is greater than 2℃ and the dew temperature is greater than or equal to 0℃, and precipitation is considered as snowfall when the surface air temperature is less than 1℃ and the dew temperature is less than 0℃. It is sleet, or rain and snow, when the surface air temperature is between 1℃ and 3℃. The surface air temperature, dew temperature, upper air temperature, and relative humidity are frequently used in developing methods to discriminate precipitation phases.

However, in a larger scale study, it is usually difficult to obtain the observational records in the global dataset. To study the separation methods of precipitation phrase on the continental and global scales, only the surface air temperature data are more easily available. Dew point temperature and relative humidity data, for example, can be used only in regional scale investigation, despite their good suitability as indicators of precipitation phrase separationcan be used (Harpold et al., 2017; Jennings et al., 2018)Bourgouin (2000) introduced the area method in separating different precipitation phases, which is based on the vertical thermal structure of the atmosphere, the distribution of condensation nuclei of water vapor, and the descent velocity to predict the precipitation phase (liquid or solid). The method, however, also needs data of multiple observational variables in surface and upper atmosphere, which is difficult to obtain.

Rainfall induced runoff and snowmelt runoff are completely different hydrological processes. Therefore, in In some hydrological models, the solid-liquid precipitation separation useds the double threshold temperature method (Wigmosta et

al., 1994; Kang et al., 1999, 2001; Chen et al., 2008) and the single threshold temperature method (Arnold et al., 1998;  Wang et al., 2004) The customized threshold temperature method has a larger error (Marks et al., 2013; Han et al., 2010). Han et al. (2010) developed an  insurance probability  method to determine the  single threshold temperature and  a fitted model in China ~~They used the data of the national stations of the China Meteorological Administration (CMA) during 1961-1979 to draw a single threshold temperature contour map, and combined it with the monthly snowfall ratio method to separate the precipitation phases by determining occurrence of snowfall and the amount of snowfall in the watershed. Chen et al. (2013) improved the solid-liquid precipitation separation procedure for mainland China by supplementing the threshold of daily mean dew temperature. The data used for the previous studies were observed prior to 1979, and they used the monthly snowfall ratio method as an auxiliary indicator. When the rainfall and snowfall condition in different regions outside mainland China is not known, and at the same time there is no dew temperature data in the current international datasets, the method cannot be applied to the larger scale analysis.~~

Arpold et al., 2017; Keith  et al., 2018), it is at the same time difficult to be used in large scale due to the unavailability of humidity data. Research on the global scales can be only based on the temperature phase~~

separating method.

China has sub-continental scale characteristics of lands and natural conditions, and has a diversity of climates and topographic types, and the phase separating methods developed in mainland China should have a better universality in continents and the world.

In this work, the precipitation phase separation method was developed by using the daily observational data of the national stations for years 1961–2013 in mainland China, and the threshold temperature values of rainfall and snowfall in northern China (north of 30°N) was analyzed and tested. A statistical model of the threshold temperature was established to provide a method for use in studies of large-scale snowfall climatology and climate change, weather forecasting, and hydrological model parameterization.

In this work, we used the daily observational data of the national stations for years 1961–2013 in mainland China, including the long-term records of air temperature, precipitation, relative humidity and visual observations of precipitation phase. Based on the single threshold temperature method of of precipitation phase separation, wWe applied use the Snowday Direct Definition Method (SDDM) to determine the single threshold temperature values of rainfall and snowfall in northern China (north of 30°N). A statistical model of the threshold temperature was established to provide a toolmethod for use in studies of large-scale snowfall climatology and climate change, weather forecasting, and hydrological model parameterization. It is believed that China has sub-continental scale characteristics of

lands and natural conditions, and  a diversity of climates and topographic types, and the phase separating methods developed in mainland China should have a better universality in continents and the world.

**2. Data and methods**

**2.1 Data**

The main purpose of this study was to develop a easy and convenient  method for separating solid and liquid precipitation, so that the objective separation of solid and liquid parts of precipitation can be achieved without exhaustive reference of observational data. International exchange data generally only contain the daily temperature and precipitation, with no other reference data, so we have only used the indicators related to temperature and precipitation to develop a method of separation.

The data  was obtained from the National Meteorological Information Center of China Meteorological Administration (CMA). The air temperature, precipitation and relative humidity data were derived from the "China Land Daily Climatic Dataset (V3.0)". The precipitation phase observation  was derived from "China Land Climatic Data Daily Weather Phenomena Dataset''. All the data have been quality controlled. Collected since January 1951, the "China Land Daily Climatic Dataset (V3.0)'' contains the daily data of air pressure, surface air temperature (daily mean, daily maximum and daily minimum), precipitation, pan-evaporation, relative humidity, wind speed, sunshine hours, and 0-cm ground

temperature from 839 stations The "China Land Climatic Data Daily Weather Phenomena Dataset" is the daily records encoded by the 752 national stations in mainland China since 1951. Cross comparison of the two datasets and the examination of station information was performed, and any incomplete temperature, precipitation, relative humidity and weather phenomena data were removed.  There are total 623 stations selected for use in the study, all of which meet the demand to have information integrity, sequential continuity, and records of more than 20 years in climate reference period (1981–2010). The data may contain inhomogeneities caused by the relocation and other factors, but they would exert little influence on the analysis results, so the data are not adjusted for homogeneity.

First, the precipitation caused by fog, dew, and frost as well as the trace precipitation was removed, and daily precipitation greater than or equal to 1 mm was taken as the effective precipitation. In this regard, the main consideration is that the international exchange precipitation  data only contains no less than 1 mm of daily precipitation.

In the separation of daily rainfall (pure rain), mixed-phase events, and snow (pure snow) events, 'pure rain' was recorded when the weather phenomenon data indicate that only rain occurred on that day without snow and mixed-phase events; it was registered as 'pure snow' when only snowfall occurred without rain and mixed-phase events, and 'mixed-phase events' when there is rain and snow in the same day, in the records of weather phenomenon data. The daily maximum and minimum temperature during an occurrence of mixed-phase events at each station were recorded as the reference thresholds for the snow and rain temperature threshold values.

When there is less snowfall at the station in lower latitude zone or more arid regions, there may be arbitrary cases of snowfall. An example is from Lijiang station, Yunnan, located in 26 °N, at which pure snow occurred only six times in the 30 years from 1981 to 2010. The representation of the threshold temperature would be poor in these cases. In order to ensure that the snowfall frequency is great enough and the threshold temperature is representative, we took 324 stations (Fig. 1) in northern China for use in this study. They are generally located north of the Yangtze River, approximately consistent with the January mean temperature isotherm of 3 °C or the 30 °N parallel. The days with the snowfall records during 1981-2010 were greater than or equal to 100d for each of the stations. In order to avoid the influence of extreme values on the determination of threshold temperature, the maximum and minimum daily mean temperature in each of the precipitation phases were not counted.

For the cases of extreme large rain and snow records at the stations, comparison was made to ensure that the minimum and maximum temperature was correct by examining the weather phenomena, surface air temperature and precipitation on the same day. When mixed-phase events occurred, the range of daily mean temperature was generally large. Threshold temperature was determined only for pure rain and pure snow; The daily mean temperature on a mixed-phase events day was only taken as the reference temperature threshold value.

2.2 Methods

According to the method of China's physical geographical regionalization, mainland China is divided into three natural geographical regions: Eastern Monsoon Region (I, 231 stations), Northwest Arid Region (II, 67 stations), and Qinghai-Tibetan Plateau Region (III, 26 stations) (Fig. 1). More stations are distributed in Eastern Monsoon Region, and there air only 26 stations in Qinghai-Tibetan Plateau Region. A vast region of western part of the Qinghai-Tibetan Plateau is the well-known no-man land without climatic observations, and this would affect the analysis in some extents. The representative station of the Eastern Monsoon Region is Zhaozhou station (Zhaozhou-I hereafter) in Heilongjiang province, which has the lowest threshold temperature of snowfall and rainfall in the country. The representative station of the Qinghai-Tibet Plateau Region is Shiquanhe station (Shiquanhe-II hereafter) in Tibet Autonomous Region, which has the highest threshold temperature of snowfall and rainfall in the country. There are relatively fewer precipitation events in the Northwest Arid Region, and Balikun station

(Balikun-III hereafter) in Xinjiang Autonomous Region was selected as the representative station because it observed relatively more precipitation events, and the rain, sleetmixed-phase events, and snow events were evenly distributed. The station is also far from the two other regions (Table 1).

[Figure]

**FIG.1. Regionalization and distribution of 324 national stations north of 30 °N in mainland China**

**(I: East Monsoon Region; II: Northwest Arid Region; III: Qinghai-Tibetan Plateau;**

**Blue triangle: stations in the East Monsoon Region; Green diamond: stations in the Northwest**

**Arid Region; Red circle: stations in the Qinghai-Tibetan Plateau.**

**The purple diamond denotes the representative stations in different regions: Zhaozhou of Region**

**I; Balikun of Region II; Shiquanhe of Region III)**

**Table 1 Information of representative stations in the three regions**

| Station name | Zhaozhou |
|---|---|
|  |  |

| RegionClimate zone | I | II | III |
| --- | --- | --- | --- |
| | Heilongjiang | Xinjiang | Tibet |
| Elevation(m) | 148.7 | 1679.4 | 4278.6 |
| Latitude(N) | 45°42′ | 43°36′ | 32°30′ |
| Longitude(E) | 125°15′ | 93°03′ | 80°05′ |

The relative or percent errordeviation of snow days (snowfall) was defined as the percentage (%) of the difference between simulated snow days (snowfall) and observational (true) actual snow days (snowfall) to the observational (true)actual snow days (snowfall), which could be used to indicate the effectiveness of simulated results.

The establishment of model was realized using the stepwise regression analysis method included with the SPSS Statistics 17.0. The basic idea of stepwise regression is that the variables are introduced one by one, the condition of introducing the variable is the square of the partial regression, and the test is significant; at the same time, after the introduction of each variable, the selected variables are checked individually and the insignificant variables are eliminated to ensure that all the

 The advantage of stepwise regression is that the number of the arguments contained in the regression equation is fewer, it is easy to apply, the root mean squared error (RMSE) is small, and the model created is more stable.

The maximum daily mean temperature at the occurrence of snowfall at the weather station is $T_{sm}$, the minimum daily mean temperature at the time of rainfall is $T_{rn}$; the number of snowfall days between $T_{rn}$ and $T_{sm}$ is $S_n$, the number of rain days is $R_n$, and the total number of rain and snow days between $T_{rn}$ and $T_{sm}$ is $N_{sr} = S_n + R_n$; the single critical temperature of rain and snow days is $T_{t-d}$, that is, the precipitation event that occurs when the daily mean temperature is lower than $T_{t-d}$ is considered to be a snowfall event, otherwise it is considered as a rainfall event; the single critical temperature estimated by the statistical model is $T_{t-p}$.

Using the Snowday Direct Definition Method (SDDM) to define the threshold temperature of precipitation phase, the calculation steps are as follows:

First, find the $T_{rn}$ and $T_{sm}$ in the dataset of the 623 stations, and count $S_n$, $R_n$, and $N_{sr}$. Second, calculate the daily average temperature of $N_{sr}$ and sort it in ascending order. Last, the average of daily mean temperature of the $S_n^{th}$ day and the $(S_n+1)^{th}$ day is calculated, and it is taken as the threshold temperature ($T_{t-d}$) of the rain and snow days. For the area where pure rain and snow events do not overlap ($T_{sm} < T_{rn}$, that is, the snowfall and rainfall events do not intersect in the

sorted daily average temperature series), the average of $T_{sm}$ and $T_{rn}$ was  taken as the $T_{t-d}$. The average of $T_{t-d}$ and the daily mean temperature of mixed-phase events day was  taken as the $T_{t-d}$ when $T_{t-d}$ was  not in the range of mixed-phase events day daily mean temperature. The $T_{t-d}$s values in this study were  all within the daily mean temperature of mixed-phase events day, however, and this operation was  not required.

~~Figure 2 shows a flow diagram of the analysis of this paper. Firstly, the daily mean temperature of different precipitation phases in northern China was calculated, the threshold temperature of each station was determined by the method of 'snow-day mean temperature', and the relationships between threshold temperature and geographical and climatic factors were analyzed. Then, by using the stepwise regression analysis method in a module of the SPSS software, the main factors affecting the threshold temperature were determined, and the threshold temperature model was established. Finally, the difference of the simulated threshold temperature and the actual threshold temperature was analyzed. The spatial distribution of the relative deviation was examined, and the applicability of the model was tested and evaluated, in the last step.~~

[Figure]

FIG.2. Technical roadmap

**3. Threshold temperature**

**3.1 Daily mean temperature corresponding to precipitation in different phases**

There are three types of precipitation phases in northern China: snowfall, rainfall and sleet. Most of the time, snowfall occurs in winter, rainfall occurs in summer, and

 Fig2 and Table 2 show phase temperature distribution of precipitation events at the stations. The total precipitation events at 324 stations were included in the statistical calculations, and their corresponding daily mean temperature values (Fig. 2a) were examined: only snowfall occurred when the daily mean temperature was below -12.9 ℃; only rainfall occurred when the daily mean temperature was higher than 22.1℃; and the three phases of snow, rain, and mixed-phase events occurred when the temperature was between -12.9℃ and 22.1℃.

In northern China,2a) pure snow (snowfall) events occurred when the daily mean temperature was below 8.5℃, and 95% of the snowfall events occurred when the daily mean temperature was  lower than 2.7℃ and higher than -16.6℃ (Fig. 2a). All pure rain events (rainfall) occurred when the daily mean temperature was higher than -4.9℃, and 95% occurred when the temperature was lower than 26.0℃ and higher than 6.4℃. All mixed-phase  events appeared in the temperature range of -12.9–22.1℃, with 95% occurring when the daily mean temperature was lower than 8.3℃ and higher than -1.6℃.

[Figure]

[Figure]

**FIG.2. Precipitation phase temperature distribution of regional average and representative stations (a-324 stations; b-Zhaozhou-I; c-Balikun-II; d-Shiquanhe-III)**

At Zhaozhou-I station (Fig. 2b), the pure snow events all occurred when the daily mean temperature was lower than -0.9 ℃, pure rainfall occurred when the daily mean temperature was higher than -3.4 ℃, and mixed-phase events occurred in case of -4.5–6.5 ℃. Zhaozhou-I station had the lowest threshold temperature (-1.2 ℃) of snowfall and rainfall in the study region. At Balikun-II station (Fig. 2c), the pure snow events all occurred when the

daily mean temperature was lower than -5.1 ℃, pure rain events occurred when the daily mean temperature was higher than 4.1 ℃, and mixed-phase events occurred within a temperature range of -7.8–12.3 ℃. At Shiquanhe-III station (Fig. 2d), the pure snow events all occurred when the daily meanaverage temperature was lower than 6.4 ℃, pure rainfall occurred when the daily mean temperature was higher than 6.1 ℃, and mixed-phase events occurred when the temperature was from -5.3 ℃ to 16.0 ℃. Shiquanhe-III station had the highest threshold temperature (6.3 ℃) of snowfall and rainfall in the whole region.

~~Pure snowfall occurred when the daily mean temperature was above 0°C, and pure rainfall occurred when it was below 0°C. This may be because the daily mean temperature is higher/lower than instantaneous air temperature when snowfall/rainfall occurs, or the instantaneous air temperature is below/above 0°C with warming/cooling after snow/rain. It could also be because the snowflakes are formed in the upper atmosphere with the lower temperature, the temperature near the surface cools faster due to the intrusion of extremely cold air, and they are not fully melted when they fall and still exist in the form of snow. In the lower atmosphere layer (below 3000 m), there is a lot of super-cooling water, and the air temperature is in the range of 0 – -15°C. With a rich condensation nucleus, an abundance of moisture, and a lack of a freezing nucleus (the ice nucleation), raindrops can form below 0°C, producing glaze or rime on the ground surface.~~

464
465
466

**Table 2 The distribution range of daily mean temperature under different phases of precipitation at stations**

| Station | | All | Zhaozhou | Balikun | Shiquanhe |
|---|---|---|---|---|---|
| | Maximum | 8.5 | -0.9 | 5.1 | 6.4 |
| | Minimum | -35.4 | -20.5 | -22.2 | -18.1 |
| | Average | -5.2 | -10.2 | -8.2 | -4.4 |
| | 5% value | -16.6 | -18.6 | -17.6 | -14.3 |
| | 95% value | 2.7 | -3.3 | 0.8 | 4.8 |
| Sleet day temperature (℃) | Maximum | 22.1 | 6.5 | 12.3 | 16.0 |
| | Minimum | -12.9 | -4.5 | -7.8 | -5.3 |
| | Average | 3.6 | 1.6 | 4.1 | 4.3 |
| | 5% value | -1.6 | -4.5 | -2.5 | -5.0 |
| | 95% value | 8.3 | 5.5 | 9.5 | 13.1 |
| Rain day temperature | Maximum | 33.3 | 27.5 | 22.1 | 18.7 |
| | Minimum | -4.9 | 3.4 | 4.1 | 6.1 |


| Station | Snow day mean temperature (℃) | | | | | Mixed-phase events day mean temperature (℃) | | | | | Rain day mean temperature (℃) | | | | |
|---|---|---|---|---|---|---|---|---|---|---|---|---|---|---|---|
| | Max | Min | Ave | 5% value | 95% value | Max | Min | Ave | 5% value | 95% value | Max | Min | Ave | 5% value | 95% value |
| All | 8.5 | -35.4 | -5.2 | -16.6 | 2.7 | 22.1 | -12.9 | 3.6 | -1.6 | 8.3 | 33.3 | -4.9 | 16.3 | 6.4 | 26.0 |
| Zhaozhou-I | -0.9 | -20.5 | -10.2 | -18.6 | -3.3 | 6.5 | -4.5 | 1.6 | -4.5 | 5.5 | 27.5 | -3.4 | 17.8 | 6.1 | 25.0 |
| Balikun-II | 5.1 | -22.2 | -8.2 | -17.6 | 0.8 | 12.3 | -7.8 | 4.1 | -2.5 | 9.5 | 22.1 | 4.1 | 14.3 | 7.3 | 19.4 |
| Shiquanhe-III | 6.4 | -18.1 | -4.4 | -14.3 | 4.8 | 16 | -5.3 | 4.3 | -5 | 13.1 | 18.7 | 6.1 | 12.6 | 8.7 | 15.7 |

It can be seen from Fig. 2 and Table 2 that there is a larger difference of the maximum daily mean temperature of snowfall (extreme threshold temperature of snowfall) and the minimum daily mean temperature of rainfall (extreme threshold temperature of rainfall) among the stations.

Figure 3 shows that There is a common spatial distribution feature in the Tsm, Trn, and the

average daily mean temperature of mixed-phase events  in northern China,

with the high values generally in the Tibetan Plateau and southern Xinjiang, while the

low values mostly in eastern and northern Xinjiang. AtIn the stations analyzed, most

have a relationship of Trn<Tsm, that is, the minimum daily mean temperature at the

time of a rain event is lower than the maximum daily mean temperature at the time of

a snowfall event. Only in a few of places in Northwest Arid Region, is the maximum

daily mean temperature of a snow day lower than the minimum daily mean

temperature of a rain day, indicating that the is, pure rain and snow events do not

overlap.

[Figure]

[Figure]

493

494 **FIG.43. The distribution of daily mean temperatures when precipitation occurs (a. Tsm**
495 **; b. Trn; c.**
496 **average daily mean temperature of mixed-phase events ; and d. difference between Tsm**
497 **and Trn**
498 **) (Red thick line represents 0℃ isotherms)**

499

500 3.2 Threshold temperature determination

501

502

503

504

505

506

507

508

509

510

511

512

513

514

Figure 4 shows the distribution of the relative error of the snow days and snowfall in northern China, determined by the threshold temperature as mentioned in section Data and Methods, to the actual snow days and snowfall counted by using weather phenomenon records. The relative error of snow day was smaller. This is due to the definition of threshold temperature being directly determined by snow-day mean temperature. Since the daily mean temperature of the $Sn^{th}$ day and the $(Sn+1)^{th}$ (or more) day is the same under this definition, however, there will be a slight positive bias in the threshold temperature of the same temperature day, with a range of relative errors (0, 2.3%).

The spatial distribution of the relative error of the snowfall was mainly positive, which is due to the systematic deviation of the method. Larger deviation appeared in eastern part of the Qinghai-Tibetan Plateau and the Yangtze-Huaihe River Basins. These areas have more precipitation and sufficient water vapor. Under the same water vapor condition, the observed rainfall was greater than the observed snowfall, and the amount of snowfall determined by the threshold temperature was slightly large, with the certain sites even larger. Small values occurred in the southeastern Northeast China, the border zone between Inner Mongolia and Xinjiang, and western Xinjiang, with the main reason related to the less precipitation and insufficient water vapor. Overall, the relative error of snowfall is between -5% and 20%. There were 312 stations (more than 96%) with deviation less than or equal to 10%, and the absolute value of the relative error was less than 5% in most areas.

[Figure]

**FIG.4.** The spatial distribution of the relative error of the days (a) and amount
(b) of snowfall determined by the threshold temperature (Tt-d) in northern China

The spatial distribution of the threshold temperature (Tt-d) of rain and snow at the

stations north of 30 N are shown in Fig. 5. The average Tt-d is 2.3 ℃ for Eastern

Monsoon Region, 3.4 ℃ for Northwest Arid Region, and 5.2 ℃ for the

Qinghai-Tibetan Plateau. The highest threshold temperature of the study region is

6.3 ℃ (Shiquanhe-III, Fig. 2d), the lowest is -1.2 ℃

(Zhaozhou-I, Fig. 2b), the threshold temperature range was 7.5 ℃, and

the average threshold temperature for the whole region was 2.8 ℃. The high-values

 was in the northern Qinghai-Tibetan Plateau, with a threshold temperature of

more than 4 ℃, and the low-value s were generally in eastern Northeast China,

North China, and northern Xinjiang with the threshold temperature mostly less than

2 ℃. The threshold temperature

 west of

100 ℉ showed an approximately zonal distribution, and it  decreased with the increase of latitude; the east of 100 ℉ had a meridional distribution, and the threshold temperature decreased with increasing longitude. There are some uncertainties on the distribution of the threshold temperature in the Qinghai-Tibetan Plateau and northwestern deserts mainly due to the interpolation in the regions with sparser observations.

[Figure]

**FIG.5. Spatial distribution of threshold temperature of precipitation phases in northern China (I: East Monsoon Region; II: Northwest Arid Region; III: Qinghai-Tibetan Plateau. Unit: ℃)**

3.3 Correlation between threshold temperature and geographical/climatic factors

The threshold temperature ($T_{t-d}$) is related to the longitude, latitude, altitude, annual precipitation, annual mean air temperature, and annual relative humidity of the observational sites, with a positive correlation with altitude and a negative correlation with the other factors. All the correlations passed the significant test ($p$=0.05) (Fig 6).

 In the lower latitude area, the threshold temperature was generally higher and more variable, while in the higher latitude , it was generally slightly lower and relatively less variable. The threshold temperature had a clear decreasing trend with increase of latitude (Fig. 6a). In lower altitude area, the threshold temperature was lower, while it was higher in mountains and plateaus, and a highly significant increasing trend of threshold temperature with altitude can be seen (Fig. 6b). There was a negative correlation between the threshold temperature and the annual precipitation, and a more significant negative correlation with the annual

relative humidity (Fig. 6c, d).

$y = -0.1194x + 7.5277$
$R^2 = 0.2272$

$y = 0.0009x + 1.8455$
$R^2 = 0.6679$

■ East Monsoon Region
♦ Northwest Arid Region
● Qinghai-Tibetan Plateau

$y = -0.0019x + 3.7903$
$R^2 = 0.1364$

$y = -0.0858x + 8.0068$
$R^2 = 0.3037$

**FIG.76. Relationship of the threshold temperature (Tt-d) with latitude (a), altitude (b), annual precipitation (c) and annual mean relative humidity (d) in northern China (Blue triangle: East Monsoon Region; Green diamond: Northwest Arid Region; Red circle: Qinghai-Tibetan Plateau)**

It is possible that the relationship of the threshold temperature with longitude and latitude is also related to the variations of altitude and relative humidity in the study region. The altitude and relative humidity generally decrease from west to east and

from south to north, and the altitude and relative humidity have better correlations with the threshold temperature, which may be the reason why threshold temperature decreases with the increase of the longitude and latitude. Therefore, altitude and relative humidity may be the more important factors in determining the threshold temperature.

The threshold temperature was positively correlated with altitude, which may mainly be because the ground surface receives stronger solar radiation, causing the boundary-layer atmosphere to heat rapidly in the high altitude areas during daytime. However, the upper air temperature is low, the temperature lapse rate is larger, the cloud bottom-height is low, and the path of snowflakes is short, so the snowfall phenomenon can also be more frequently observed when the daytime surface air temperature is high.

The threshold temperature was negatively correlated with annual precipitation in particular with relative humidity, which may be related to the low latent heat flux and high sensible heat flux in arid area. When the sensible heat flux is high, the ground surface air temperature is high, and the temperature lapse rate is large. In the case of the same condensation height or cloud bottom-height, snowfall is more likely to occur under the condition of higher surface air temperature. It is also possible that the higher the threshold temperature in arid area than in humid area is caused by a difference of the more complicated microphysical processes around the snowflakes between the two climatic conditions.

3.4 Establishment of the threshold temperature model

629 Considering that the relative humidity data of some areas is difficult to obtain, the

630 precipitation factor was selected as the independent variable. Using the SPSS software

631 stepwise regression analysis method, a statistical model of threshold temperature was

632 established with latitude, altitude, and annual precipitation as influential factors. The

633 model, which passed the significant test at the 0.05 level, can be expressed as follow:

634 $Tt\text{-}p = 6.81576376 + (-.09305) * N + (.000567) * H + (-0.00182) * R$ (1)

635 where Tt-p is the simulated threshold temperature (℃), N is the latitude of the station,

636 H is the altitude of the station (m), and R is the annual precipitation of the station

637 (mm).

638 The correlation coefficient between Tt-p and Tt-d (threshold temperature

639 determined by using the synoptic phenomena) is 0.87. The median and standard

640 deviation of the simulated threshold temperature (Tt-p) were 2.53 and 1.16, which

641 were close to the median (2.64) and standard deviation (1.33) of the Tt-d. The

642 maximum simulated threshold temperature was 6.05 ℃, minimum was -0.22 ℃,

643 temperature range was 6.26 7 ℃, and average simulated threshold temperature was

644 2.81 ℃ for the whole region. The maximum positive deviation of the Tt-p to the Tt-d

645 was 3.0 ℃, and the minimum negative deviation was -1.7 ℃. The stations, at which

646 relative deviation of snow day and snowfall were less than 10%, reached 95% and 91%

647 of the total, respectively.

648 In the East Monsoon Region (Region I) and the Northwest Arid Region (Region

649 II), the simulated threshold temperature was generally lower than the Tt-d (0.005 ℃

650 lower in Region I on average, and 0.02 ℃ lower in Region II on average). However, it

was higher in the Qinghai-Tibetan Plateau Region (0.097 ℃ higher on average) (Fig. 87). Considering that the relative humidity data of some areas is difficult to obtain, the precipitation factor was selected as the independent variable. Using the SPSS software stepwise regression analysis method, a statistical model of threshold temperature was established with annual mean air temperature, altitude, and annual precipitation as influential factors. The model, which passed the significant test ($p$=0.05), can be expressed as follow:

$$Tt\text{-}p = 1.69147 + (.09585) * T + (.001311) * H + (-0.00172) * R \qquad (1)$$

where Tt-p is the simulated threshold temperature (℃), T is the annual mean air temperature(℃) of the station, H is the altitude of the station (m), and R is the annual precipitation of the station (mm).

The correlation coefficient between Tt-p and Tt-d (threshold temperature determined by using the synoptic phenomena) is 0.88. The median and standard deviation of the simulated threshold temperature (Tt-p) were 2.54℃ and 1.17℃, which were close to the median (2.64℃) and standard deviation (1.33℃) of the Tt-d. The maximum simulated threshold temperature was 5.9 ℃, minimum was -0.4 ℃, temperature range was 5.5 ℃, and average simulated threshold temperature was 2.8 ℃ for the whole region. The maximum positive deviation of the Tt-p to the Tt-d was 2.9 ℃, and the minimum negative deviation was -1.8 ℃. The numbers of stations with relative error less than 10% for snow day and snowfall reached 97% and 92% respectively.

In the East Monsoon Region (Region I), the simulated threshold temperature was

generally lower than the Tt-d (0.026 ℃ in Region I on average). However, it was higher in the Northwest Arid Region (Region II) and the Qinghai-Tibetan Plateau Region (0.063℃ lower in Region II on average, and 0.065℃ higher on average) (Fig. 7).

FIG.87. Simulated threshold temperature (Tt-p), actual thresholdthreshold temperature (Tt-d) and their difference for observational stations in different regions of northern China (1: East Monsoon Region; 2: Northwest Arid Region; 3: Qinghai-Tibetan Plateau Region)

The correlation coefficients of the standard deviation and median of the snowfall days (simulated snowfall days) with those of actual snowfall days at all the stations were 0.92 and 0.94, respectively. The differences of the standard deviation and median of the simulated snowfall days and actual snowfall days are smaller overall, and the differences of the median is slightly larger in the Qinghai Tibet Plateau where there was more snowfall. Fig. 9 8 shows spatial distribution of the relative deviation of the simulated snow days (Fig. 9a8a) and snowfall (Fig. 9b8b) relative to the actual snow days and snowfall at the stations. The relative deviation range of snowfall days in northern China was between -21.17% and 18.38%, with an average of -0.12%; the relative deviation was smaller in mid-southern parts of the study region, and larger in the coastal areas and the northern Qinghai Tibetan PlateauQinghai Xizang Plateau. In the Qinghai Tibetan PlateauQinghai Tibet Plateau Region, the medians of the simulated snow days were smaller than those of the actual snow days, and the relative deviations were larger. This may be related to the fact that the snowfall days in northern Tibetan Plateau fluctuated greatly, and there are some years with larger numbers of snowfall days. The relative deviation range of snowfall in the whole region was between -17.3% and 30.38% with an average of 1.09%, and the spatial distribution was basically the same as that of the relative deviations of snow days.

[Figure]

702

**FIG.7. Simulated threshold temperature (Tt-p), threshold temperature (Tt-d) and their difference for observational stations in different regions of northern China (Region 1: East Monsoon Region; Region 2: Northwest Arid Region; Region 3: Qinghai-Tibetan Plateau Region)**

The mean absolute errors of threshold temperature of the simulated snowfall are 0.476℃ for East Monsoon Region, 0.560℃ for Northwest Arid Region, and 0.435℃ for Qinghai-Tibetan Plateau Region. Fig. 8 (and also Table 3, Mean errors of threshold temperature of simulated snowfall and snow days for the study area and the three regions) shows the spatial distribution of the absolute errors of threshold temperature of the simulated snowfall and snow days. Larger errors can be seen in the Northwest Arid Region.

[Figure]

[Figure]

**FIG. 8. Absolute error distribution of snowfall days (a) and snowfall (b) for the simulated threshold temperature**

Fig. 9 shows spatial distribution of the relative error of the simulated snow days (Fig. 8a) and snowfall (Fig. 8b) relative to the actual snow days and snowfall at the stations. The relative error range of snowfall days in northern China was between -16.8% and 17.0%,with an average of -0.1%; the relative error was smaller in mid-southern parts of the study region, and larger in the coastal areas and the northern Qinghai-Tibetan Plateau. In the Qinghai-Tibetan Plateau Region, the medians of the simulated snow days were smaller than those of the actual snow days, and the relative errors were larger. This may be related to the fact that the snowfall days in northern Tibetan Plateau fluctuated greatly, with some years with larger numbers of snowfall days. The relative error range of snowfall in the whole region was between -15.5% and 29.0% with an average of 1.1%,and the spatial distribution was basically the same as that of the relative errors of snow days.

[Figure]

**FIG.9. Relative error  distribution of snowfall days (a) and snowfall (b) defined by the simulated threshold temperature**

Affected by the extremely low air temperature and the abnormally deficient water vapor due to the East Asian winter monsoon, the  snow days (snowfall) with only snowfall weather  were relatively less frequent (low) in northern China, as compared to other regions of the same latitude; therefore, it is more likely that the relative error is large in the study region. However, the relative error range shown here is acceptable, and the fitting effect is generally good.

The RMSE  of the relative error of snow days was 3.9, and the RMSE  of the relative error of snowfall was 5.3. The annual snow days and the amount of snowfall were less in the mid-southern parts of the study region which had negative relative errors of the simulated snow events; however, snow days and snowfall were slightly more numerous in the

northern part of the Sichuan Basin. The number of snow days and snowfall was less in the coastal area which had positive relative errors of the simulated snow events, while there were more snow days and snowfall in the northern Qinghai-Tibetan  Plateau. The relative error of snow days (snowfall) and the threshold temperature had a correlation coefficient of -0.38 (-0.31); both passed the significant test $(p=0.05)$ . It can be seen that the relative error in the area with low threshold temperature tends to be positive, and thate relative error in the area with high threshold temperature is generally negative.

4. Comparison with previous works

Previous researches used the insurance probability to obtain the threshold geophysical parameters of the snow rain separation (e.g. Han et al., 2010; Sims and Liu, 2015). Sims and Liu (2015) found that the wet bulb temperature and low layer temperature lapse rate had the most significant influence on the precipitation phase, with a lapse rate of 6℃·km$^{-1}$ resulting in an 86% insurance probability of solid

precipitation if the near-surface wet-bulb temperature was around 0as. Surface air pressure also exerted an influence on precipitation phase in some cases. However, the climatic parameters are once again less available in the major international historical climate datasets, though the finding and the method recommended are valuable in investigating into local and regional precipitation phrases.

Han et al.(2010) used the insurance probability method to determine the single threshold temperature of rain and snow,takeing the daily average temperature of the 98.5% guarantee tate of rainfall and snowfall, 50% guarantee rate of mixed-phase event as the threshold temperature.For comparison of SDDMsnow-day mean temperature method and the insurance probability method insurance probability method as reported in Han et al. (2010), the number of snow days (Sn) and rain days (Rn) between Trn and Tsm was calculated, respectively. The corresponding daily mean temperature at the insurance probability of the snow and rain days between [Trn, Tsm], X (x∈ (0-99%)) (at 1% intervals), was estimated. For example, the number of rain days and snow days between Trn and TSM is 100d respectively; when x = 90% is taken, the rain day temperature Tr90 corresponds to the insurance probability of 90%, that is, to ensure the minimum daily mean temperature in the event of 90% rain days between Trn and TSM, while Ts90 is to guarantee that the maximum daily mean temperature in the event of 90% snowfall days is between Trn and TSM. The arithmetic mean of each station's Trx and Tsx is defined as the threshold temperature Tt-x at the station's insurance probability x.

The threshold temperature (Tt-x) was calculated according to the insurance

probability method, and the threshold temperature (Tt-d) was obtained based on the definition in this paper; the relative deviationrelative error comparison is presented in Table 3. For simplicity, the insurance probability interval in the table was taken as 10%. The maximum, minimum, and range of the threshold temperature (Tt-x) under different insurance probability, and of the (Tt-d), in northern China, are given in the table; at the same time, the maximum, minimum, and range of the relative deviationrelative error of the snow days and snowfall, as well as the number of stations with a relative deviationrelative error less than or equal to 10%, are also given.

Table 3 Comparison of statistics and the relative deviationrelative errors resulting from threshold temperature Tt-x and Tt-d

[Figure]

max    min    max-min    max

6.4      -2.3      8.7      30.2      -11.1      41.3      311      36.6      -15.3      51.9


非加粗

| | | | | | | | | | |
|---|---|---|---|---|---|---|---|---|---|
| Tt-10 | 6.4 | -2.3 | 8.7 | 25.1 | -11.1 | 36.3 | 313 | 29 | -11.8 | 40.8 |
| Tt-20 | 6.5 | -2.3 | 8.8 | 25.1 | -9.5 | 34.6 | 316 | 29 | -9.7 | 38.x |
| Tt-30 | 6.5 | -2.2 | 8.7 | 23.6 | -7.1 | 30.7 | 314 | 31.5 | -15.3 | 46.8 |
| Tt-40 | 6.4 | -2.2 | 8.6 | 23.6 | -5.8 | 29.4 | 316 | 31.5 | -8.4 | 39.x |
| Tt-50 | 6.5 | -2 | 8.5 | 21.1 | -5.7 | 26.8 | 312 | 32.2 | -9.7 | 41.x |
| Tt-60 | 6.4 | -1.5 | 7.9 | 19.1 | -6.5 | 25.6 | 313 | 32.2 | -9.7 | 41.x |
| Tt-70 | 6.4 | -1.4 | 7.8 | 15.6 | -6.5 | 22.1 | 314 | 30.2 | -6.2 | 36 |
| Tt-80 | 6.7 | -1.4 | 8.1 | 18.3 | -5.8 | 24 | 307 | 45.2 | -8.4 | 53.x |
| Tt-90 | 6.5 | -1.2 | 7.7 | 23 | -7 | 29.9 | 306 | 33.4 | -9.7 | 43.x |
| Tt-d | 6.3 | -1.2 | 7.5 | 2.6 | 0 | 2.6 | 323 | 20.2 | -4.3 | 24.x |

Table 3 shows that, using the insurance probability method, the test results of the threshold temperature (Tt 70), obtained when the insurance probability x = 70% was taken, represented the best values, as the difference between the minimum and maximum values of the threshold temperature was small, and the relative errors were small, with the relative deviationrelative error of the snow days at 314 stations ≤ 10%, and that of the snowfall at 283 stations ≤ 10%.

The range of threshold temperature Tt d of snow days determined in this paper was less than that of the Tt 70. The relative deviationrelative error of snow days was obviously small, and the relative deviationrelative error of snowfall was much less than that of the Tt 70, with more stations having the relative deviationrelative errors ≤10% for both snow days and snowfall. Therefore, the method developed in this paper has an advantage over the insurance probability method developed in the previously works.

**Table 3 Comparison of statistics and the errors resulting from threshold temperature Tt-p**

**Table 3 Comparison of statistics and the errors resulting from Tt-p**

| | North of China | Region I | Region II | Region III |
|---|---|---|---|---|

|  |  |  |  |  |  |
|---|---|---|---|---|---|
|  | Correlation | 0.999 | 0.998 | 0.999 | 0.999 |
| Snowday | MAE (d) | 7.77 | 7.52 | 5.21 | 16.69 |
|  | MRE (%) | 2.71 | 2.98 | 1.68 | 2.98 |
|  | RMSE (d) | 3.9 | 4.3 | 2.3 | 3.6 |
|  | Correlation | 0.997 | 0.996 | 0.999 | 0.998 |
| Snowfall | MAE (mm) | 37.18 | 37.7 | 19.67 | 77.6 |
|  | MRE (%) | 3.71 | 4.09 | 2.1 | 4.44 |
|  | RMSE (mm) | 73.78 | 76.13 | 43.81 | 95.24 |

**45. Discussion**

[revised manuscript text omitted]

Ding(2014) used a parameterization scheme to determine the precipitation type with wet-bulb temperature ,relative-humidity,and surface elevation. In the work of Ding, each precipitation event is assigned a number, determined phase with Ding's parameterization scheme and other nine scheme in the literature. The accuracies over the air temperature range [0℃,4℃] show that Ding's method is better than other.

In order to compare with Ding's work, the accuracy of Tt-d is tested over the same air temperature range. The results are shown in Table 5, the accuracy of Ding is from Ding (2014). The Tp-CH is the Tt-p of all stations in the study area. The Tp-R1 is the Tt-p of stations independent sample modeling in East Monsoon Region . The Tp-R2 is the Tt-p of Northwest Arid Region. The Tp-R3 is the Tt-p of Qinghai-Tibetan Plateau.

Table 5    Accuracies of Ding's and SDDM over the air temperature range [0℃,4℃]

|  | Ding | Tp-CH | Tp-R1 | Tp-R2 | Tp-R3 |
|---|---|---|---|---|---|
| China | 59.3 | 87 | 87 | 84 | 84 |
| R1 | 53.1 | 83 | 83 | 78 | 80 |
| R2 | 60.1 | 89 | 89 | 90 | 88 |
| R3 | 66.1 | 99 | 99 | 99 | 99 |

Ding separated rain, snow, and sleet. The work of this paper only separates rain and snow, so the accuracy should be higher. However, the test result of Tt-p of each region is relatively stable. The SDDM is stable and reliable for the separation of rain and snow. Liu and Ren et al.(2018) tested the possibility of extrapolating the statistical model of separating solid precipitation from liquid precipitation. Considering less snowfall in the lower latitudes and less rainfall in the higher latitudes, the proposed area for use of the method is between 30 °N to 60 °N in the latitude range, and areas outside this range may have large deviation.

In this paper, only the two phases of pure snowfall and pure rainfall were determined, however, and the sleetmixed-phase events wereas not analyzed. In the case of sleetmixed-phase events, the surface air temperature changed greatly during a day; there was probably sleetmixed-phase events, pure rain and pure snow in the same day, the actual thresholdthreshold temperature fluctuations were large, and it would be difficult to accurately determine and simulate. Because the method used in this paper did not quantify the sleetmixed-phase events, when precipitation was separated into solid and liquid state, the mixed-phase eventssleets will be classified as snow when the daily mean temperature is lower than the threshold temperature, and as rain when the daily mean temperature is higher than the threshold temperature, causing a certain error. However, for the study of large-scale snowfall climatology, especially for studies of the larger than subcontinental scale snowfall climate change, the snow and rain separation method presented in this paper could well meet the needs.

Liu and Ren et al.(2018) tested the possibility of extrapolating the statistical

model of separating solid precipitation from liquid precipitation. Considering less snowfall in the lower latitudes and less rainfall in the higher latitudes, the proposed area for use of the method is between 30 °N to 60 °N in the latitude range, and areas outside this range may have large deviation.

**5. Conclusions**

Based on the analysis of the historical daily temperature, precipitation, and weather phenomenon observation data in northern China, the threshold temperature model for determining the phase of rain and snow was established and tested. The main conclusions are as follows:

(1) The threshold temperature value of rain and snow determined based on weather phenomenon data is between -1.2–6.3 °C, with a temperature range of 7.5 °C and an average value of 2.81 °C. The high values were in the northern Qinghai-Tibetan Plateau, reaching more than 4 °C, and the low values were found in Northeast China, North China, and northern Xinjiang Autonomous Region, generally less than 2 °C. The west of 100 °E showed an approximately zonal distribution,

and the threshold temperature decreased with latitude; the east of 100°E had a

[revised manuscript text omitted]

*Harpold, A.A., Kaplan, M., Klos, P.Z., Link, T., McNamara, J.P., Rajagopal, S., Schumer, R., Steele, C.M., 2017a. Rain or snow: hydrologic processes, observations, prediction, and research needs. Hydrol Earth Syst Sci 21, 1–22.*

Jennings KS, Winchell TS, Livneh B, et al. 2018. Spatial variation of the rainsnow temperature threshold across the Northern Hemisphere. Nat. Commun. 9. DOI:10.1038/s41467-018- 03629-7.

Jiang Y, Qian WH. 2003: Regional characteristics of Heavy Snowfall or Snowstorm in Inner Mongolia.

ACTA GEOGRAPHICA SINICA. 58 (Suppl.): 38-48.

Jones PD, Hulme M. 1996: Calculating regional climatic time series for temperature and precipitation: methods and illustrations. Int J Climatology, 16: 361-377.

Kang ES, Cheng GD, Lan YC, et al . 2001: Alpine-runoff simulation of the Yalong River for the South-North Water Diversion. Journ al of Glaciology and Geocryology, 23(2) : 139- 148.

Kang ES, Cheng GD, Lan YC, et al . 1999: A model for simulating the response of run off from the mountainous watershed of inland river basins in the arid area of northwest China to climatic changes. Science in China, Series D, 42 (Suppl.): 52- 63.

Keith SJ, Taylor SW, Ben L, et al. 2018: Spatial variation of the rain-snow temperature threshold across the Northern Hemisphere. Nature communications. DOI: 10.1038/s41467-018-03629-7.

Kienzle S, 2008: A new temperature based method to separate rain and snow. Hydrol. Processes, **22**, 5067-5085. DOIDOI: 10.1002/hyp.7131.

Kunkel KE, Palecki MA, Ensor L, et al. 2009: Trends in twentieth-century U.S. extreme snowfall seasons. Journal of Climate, 22(23): 6204-6216.

Loth B, Graf H, Oberhuber JM. 1993: Snow cover model for global climate simulations. Journal of Geophysical Research: Atmospheres. (1984–2012) 98(D6): 10451-10464.

Liu JH, Zhang PZ, Meng CH, et al. 2011: A Comprehensive Study on Precipitation Change Over Monsoon Marginal Areas During Last Five Centuries. SCIENTIA GEOGRAPHICA SINICA, 31(4):401-411.

Liu YL, Ren GY, Yu HM. 2012: Climatology of Snow in China, Scientia Geographica Sinica, 32(10):1176-1185.

Liu YL, Ren GY, Yu HM, et al. 2013: Climatic Characteristics of Intense Snowfall in China with Its Variation. Journal of Applied Meteorological Science, 24(3):304-313.

Liu YL, Ren GY,Sun XB. 2018. Establishment and Verification of Single Threshold Temperature Model for Partition Precipitation Phase Separation. Journal of Applied Meteorological Science. 29(4):449-459.

Liu YF, Zhu GF, Zhao J, et al. 2016: Spatial and Temporal Variation of Different Precipitation Type in the Loess Plateau Area. Scientia Geographica Sinica, 36 (8): 1227-1233.

Marks D,Winstral A,Reba M,et al. 2013. An evaluation of methods for determining during-storm precipitation phase and the rain/snow transition elevation at the surface in a mountain basin, Advanced in Water Resourced,55:98-110. DOI: 10.1016/j.advwatres.2012.11.012.

[revised manuscript text omitted]

---

## Author Comment (AC6) · 8 Oct 2018

This is the revised MS with all traces cleaned. Thabks. Guoyu

Please also note the supplement to this comment:
https://www.hydrol-earth-syst-sci-discuss.net/hess-2018-307/hess-2018-307-AC6-supplement.pdf

———————————————

---

## Referee Comment (RC3) · Anonymous Referee #3 · 22 Oct 2018

I appreciate the efforts made by the authors to share their findings from the study presented in this manuscript. I was particularly intrigued by one of the largest datasets used in this type of studies. Probably most interesting of all, was the title that attracted me to this manuscript. However, I have to admit that after reading the manuscript very carefully, I think there are significant gaps in many respects of this paper in its current form that prevented it from becoming an otherwise a very promising paper with large geographical coverage and data availability. As for the title, I feel hard to agree that it reflect what the paper actually does, nor can I see the novelty in the method applied in the first place.

I share many views of the other reviewers, however, my personal opinion is that substantial re-editing is needed for a re-submission (if re-submission is agreed by the editor responsible), hence no need asking the authors to mend it piecewise. However, I would like the authors to consider the followings when editing/improving the paper.

[Figure]

1. Title. Unless there is a real novel method developed/improved from old one, I think it is wise not to use the current title. Clearly, relying on a software SPSS and using a standard package to do a step-wise regression fitting, does not warrant a new method. I did want to see if there were any improvement to the method per se to justify the title, but unfortunately none was there. I do feel that the study can make a good case-study, feature-finding paper instead, even use existing method with careful analysis and justification. So my suggestion is to use an alternative, more accurate title to reflect the real work done.

2. The methodology. I felt particularly uneasy when I saw a method, before being carefully validated and justified, having been used to drive another analysis. As to the paper in its current form, I don't see sufficient evaluation and justification made against the regression model fitted. It immediately started talking about the analysis of the distribution of the threshold temperature based on the regression model (which has a rather low R-squared value in the first place) without recognising whether those results are reliable or not. My suggestion is to include more convincing analysis against the model instead showing at least the model is acceptable and focus on the uncertain nature. Also even within the category of those simple statistical models, there are plenty of alternatives. Please consider a more extensive view as to the choice of your models.

3. The science. It would be nice to see the results/distribution linking to any scientific findings, or at least, proper scientific explanations associated with the pattern found. This would be far more important in a 'case-study' paper than a 'method-development' one. Please consider in-depth views in this respect.

4. Last but not least, the presentation of the current paper is way below the bar. Apparently, editorial helps from native English speakers or a professional service are strongly advised. In addition to many grammar glitches, the paper was structured in a very confusing way and very hard to follow - it looks like that it has been directly translated from another document written in different language or for different purposes. I would

suggest to follow a thread of: problem statement -> literature review -> Data and study area -> Methodology and justification -> results discussion/science revelation.

---

## Short Comment (SC1) · 23 Oct 2018

Dear Colleague,

Many thanks for the comments and suggestions.

These would be help in an improvement of the manuscript.

I note that you read the original version of the manuscript. We have made a major revision recently, referring to the comments by RC #1 and RC #2. A significant improvement would be obvious, and I would attach the new version in this response.

However, I realize that most of your comments and suggestions are extra, and we intend to make a further revision accordingly. We will submit the revised manuscript in one month.

[Figure]

Best regards,

Guoyu Ren

Please also note the supplement to this comment:
https://www.hydrol-earth-syst-sci-discuss.net/hess-2018-307/hess-2018-307-SC1-supplement.pdf

[Figure]

**A new method for determining the single threshold temperature of precipitation phase separation**

Yulian Liu[a,b]   Guoyu Ren[a,c]   Xiubao Sun[a,c]   Xiufen Li[d,e]   Hengyuan Kang[f,]

[a] Department of Atmospheric Science, School of Environmental Science, China University of Geosciences, Wuhan 430074, China;

[b] Heilongjiang Climate Center, Harbin 150030,China;

[c] Laboratory for Climate Studies, National Climate Center, CMA, Beijing100081,China;

[d] Innovation and Opening laboratory of Regional Eco-Meteorology in Northeast, China Meteorological Administration

[e] Heilongjiang Provincial Institute of Meteorological Sciences, Harbin 150030,China.

[f] Harbin Meteorological Bureau, Harbin 150028,China.

Corresponding author: G. Ren (guoyoo@cma.gov.cn)

---

## Author Comment (AC7) · 20 Nov 2018

Responses to the interactive comments by anonymous Referee #3

I appreciate the efforts made by the authors to share their findings from the study presented in this manuscript. I was particularly intrigued by one of the largest datasets used in this type of studies. Probably most interesting of all, was the title that attracted me to this manuscript. However, I have to admit that after reading the manuscript very carefully, I think there are significant gaps in many respects of this paper in its current form that prevented it from becoming an otherwise a very promising paper with large geographical coverage and data availability. As for the title, I feel hard to agree that it reflect what the paper actually does, nor can I see the novelty in the method applied in the first place.

R: Thanks for the comments. A further revision has been made referring to the comments, on basis of the last revision as suggested by other reviewers. Please note that some of your suggestions were also proposed by the two previous reviewers, and we had already made changes of the manuscript and submitted the revised version and responses before receiving your comments. We hope that this version of manuscript has been further improved.

I share many views of the other reviewers, however, my personal opinion is that substantial re-editing is needed for a re-submission (if re-submission is agreed by the editor responsible), hence no need asking the authors to mend it piecewise. However, I would like the authors to consider the followings when editing/improving the paper.

R: Thanks. We have made changes, and will make further revisions, referring to the comments and suggestions.

1. Title. Unless there is a real novel method developed/improved from old one, I think it is wise not to use the current title. Clearly, relying on a software SPSS and using a standard package to do a step-wise regression fitting, does not warrant a new method. I did want to see if there were any improvement to the method per se to justify the title, but unfortunately none was there. I do feel that the study can make a good case-study, feature-finding paper instead, even use existing method with careful analysis and justification. So my suggestion is to use an alternative, more accurate title to reflect the real work done.

R: This is consistent with the suggestions by two other reviewers. The title has been changed to "A new method for determining the single threshold temperature of precipitation phase separation". This paper proposes a Snowday-Direct-Definition-Mothod (SDDM) to determine the threshold temperature of rainfall and snowfall for a complex and diverse geographical and climatic region in the world.

2. The methodology. I felt particularly uneasy when I saw a method, before being carefully validated and justified, having been used to drive another analysis. As to the paper in its current form, I don't see sufficient evaluation and justification made against

the regression model fitted. It immediately started talking about the analysis of the distribution of the threshold temperature based on the regression model (which has a rather low R-squared value in the first place) without recognising whether those results are reliable or not. My suggestion is to include more convincing analysis against the model instead showing at least the model is acceptable and focus on the uncertain nature. Also even within the category of those simple statistical models, there are plenty of alternatives. Please consider a more extensive view as to the choice of your models.

R: Thanks for the comments. This manuscript applies the Snowday-Direct-Definition-Method (SDDM) to determine the threshold temperature of snow and rain days in a large region where there are recorded weather phenomenon data of rain and snow events, and develops a statistical model with an intention that it could be used in other regions where weather phenomenon data of rain and snow events are unavailable. We will make further revision, with an emphasis given to the validation of the model we developed. For simplicity, the method to develop the model is usually stepwise regression. The physical relationships among the threshold temperature and the climatic variables are explicit. The method has been verified and validated, and it is therefore valid. We will add an additional analysis of the validation and uncertainty of the model in Section Discussion of the revised manuscript. Information on verification of the threshold temperature simulation could also be found in a Chinese paper"Liu YL, Ren GY, Sun XB. 2018. Establishment and Verification of Single Threshold Temperature Model for Partition Precipitation Phase Separation. Journal of Applied Meteorological Science. 29(4):449-459", which has been cited in section Discussion of the revised manuscript.

3. The science. It would be nice to see the results/distribution linking to any scientific findings, or at least, proper scientific explanations associated with the pattern found. This would be far more important in a 'case-study' paper than a 'method-development' one. Please consider in-depth views in this respect.

R: Many thanks. We have added explanations of the results revealed in the analysis

in the revised manuscript. The examples include, but not limited to, the follows. L391: A further explanation of relationship between snow season and snow days, and the possible effects of altitude and relative humidity on the threshold temperature: "This spatial distribution is highly consistent with the spatial distribution of the length of the Chinese snow season and the number of snowfall days (Liu and Ren, 2012). The threshold temperature of the Qinghai-Tibet Plateau with a longer snow season and more snowdays is higher. In the areas with higher altitude and lower relative humidity, the critical temperature is higher, and the critical temperature is lower in the region with lower altitude and higher relative humidity." More explanations were added to the relevant subsections. For example, in "3.3 Correlation between threshold temperature and geographical/climatic factors", we cited Ding (2014) in discussing the altitude influence. Over higher-elevation regions, snow droplets can land faster and thus absorb less heat from the ambient air. So the droplets tend to stay as snow at high elevations and a higher threshold temperature is needed for discriminating snow and rain.

4. Last but not least, the presentation of the current paper is way below the bar. Apparently, editorial helps from native English speakers or a professional service are strongly advised. In addition to many grammar glitches, the paper was structured in a very confusing way and very hard to follow - it looks like that it has been directly translated from another document written in different language or for different purposes. I would suggest to follow a thread of: problem statement -> literature review -> Data and study area -> Methodology and justification -> results discussion/science revelation.

R: The structure of manuscript has been adjusted according to the suggestion. The English has also been polished once again.

Please also note the supplement to this comment:
https://www.hydrol-earth-syst-sci-discuss.net/hess-2018-307/hess-2018-307-AC7-supplement.pdf